# CLOOB: Modern Hopfield Networks with InfoLOOB Outperform CLIP

**Andreas Fürst** [*1]    **Elisabeth Rumetshofer** [*1]    **Johannes Lehner** [1]    **Viet Tran** [1]

**Fei Tang** [3]    **Hubert Ramsauer** [1]    **David Kreil** [2]    **Michael Kopp** [2]

**Günter Klambauer** [1]    **Angela Bitto-Nemling** [1]    **Sepp Hochreiter** [1 2]

[1] ELLIS Unit Linz and LIT AI Lab, Institute for Machine Learning,
Johannes Kepler University, Linz, Austria
[2] Institute of Advanced Research in Artificial Intelligence (IARAI), Vienna, Austria
[3] HERE Technologies, Zurich, Switzerland
[*] Equal contribution

## Abstract

CLIP yielded impressive results on zero-shot transfer learning tasks and is considered as a foundation model like BERT or GPT3. CLIP vision models that have a rich representation are pre-trained using the InfoNCE objective and natural language supervision before they are fine-tuned on particular tasks. Though CLIP excels at zero-shot transfer learning, it suffers from an explaining away problem, that is, it focuses on one or few features, while neglecting other relevant features. This problem is caused by insufficiently extracting the covariance structure in the original multi-modal data. We suggest to use modern Hopfield networks to tackle the problem of explaining away. Their retrieved embeddings have an enriched covariance structure derived from co-occurrences of features in the stored embeddings. However, modern Hopfield networks increase the saturation effect of the InfoNCE objective which hampers learning. We propose to use the InfoLOOB objective to mitigate this saturation effect. We introduce the novel "Contrastive Leave One Out Boost" (CLOOB), which uses modern Hopfield networks for covariance enrichment together with the InfoLOOB objective. In experiments we compare CLOOB to CLIP after pre-training on the Conceptual Captions and the YFCC dataset with respect to their zero-shot transfer learning performance on other datasets. CLOOB consistently outperforms CLIP at zero-shot transfer learning across all considered architectures and datasets.

## 1   Introduction

Contrastive Language-Image Pre-training (CLIP) showed spectacular performance at zero-shot transfer learning (Radford et al., 2021). CLIP learns expressive image embeddings directly from raw text, thereby leverages a much richer source of supervision than just labels. The CLIP model is considered as an important foundation model (Bommasani et al., 2021), therefore a plethora of follow-up work has been published (see Appendix Section A.4). CLIP as a contrastive learning method has two simultaneous goals, namely (i) increasing the similarity of matched language-image pairs and (ii) decreasing the similarity of unmatched language-image pairs. Though CLIP yielded

---

Code is available at: https://github.com/ml-jku/cloob

36th Conference on Neural Information Processing Systems (NeurIPS 2022).

striking zero-shot transfer learning results, it still suffers from "explaining away". Explaining away is known in reasoning as the concept that the confirmation of one cause of an observed event dismisses alternative causes (Pearl, 1988; Wellman & Henrion, 1993). CLIP's explaining away problem is its focus on one or few features while neglecting other relevant features. This problem is caused by insufficiently extracting feature co-occurrences and covariance structures in the original multi-modal data. Humans extract co-occurrences and covariances by associating current perceptions with memories (Bonner & Epstein, 2021; Potter, 2012). In analogy to these human cognitive processes, we suggest to use modern Hopfield networks to amplify co-occurrences and covariance structures of the original data.

Hopfield networks are energy-based, binary associative memories, which popularized artificial neural networks in the 1980s (Amari, 1972; Hopfield, 1982, 1984). Associative memory networks have been designed to store and retrieve samples. Their storage capacity can be considerably increased by polynomial terms in the energy function (Chen et al., 1986; Psaltis & Cheol, 1986; Baldi & Venkatesh, 1987; Gardner, 1987; Abbott & Arian, 1987; Horn & Usher, 1988; Caputo & Niemann, 2002; Krotov & Hopfield, 2016). In contrast to these binary memory networks, we use continuous associative memory networks with very high storage capacity. These modern Hopfield networks for deep learning architectures have an energy function with continuous states and can retrieve samples with only one update (Ramsauer et al., 2021). Modern Hopfield networks have already been successfully applied to immune repertoire classification (Widrich et al., 2020), chemical reaction prediction (Seidl et al., 2022) and reinforcement learning (Widrich et al., 2021; Paischer et al., 2022). Modern Hopfield networks are a novel concept for contrastive learning to extract more covariance structure.

However, modern Hopfield networks lead to a higher similarity of retrieved samples. The increased similarity exacerbates the saturation of CLIP's InfoNCE objective (van den Oord et al., 2018). InfoNCE saturates because it contains terms of the form $a/(a + b)$. In analogy to Wang & Isola (2020), $a$ is called the "alignment score" that measures the similarity of matched pairs and $b$ is called the "uniformity penalty" that measures the similarity of unmatched pairs. The saturation problem becomes more severe for retrieved samples of the modern Hopfield network since the alignment score $a$ increases. Saturation of InfoNCE hampers the decrease of the uniformity penalty $b$ (see also Yeh et al. (2021)). Contrary to InfoNCE, the "InfoLOOB" (LOOB for "Leave One Out Bound") objective (Poole et al., 2019) contains only terms of the form $a/b$ which do not saturate. Thus, even for a large alignment score $a$, learning still decreases the uniformity penalty $b$ by distributing samples more uniformly.

We introduce "Contrastive Leave One Out Boost" (CLOOB) which combines modern Hopfield networks with the "InfoLOOB" objective. Our contributions are:

(a) we propose CLOOB, a new contrastive learning method,

(b) we propose modern Hopfield networks to reinforce covariance structures,

(c) we propose InfoLOOB as an objective to avoid saturation as observed with InfoNCE, and provide theoretical justifications for optimizing InfoLOOB.

## 2 CLOOB: Modern Hopfield Networks with InfoLOOB

Our novel contrastive learning method CLOOB can be seen as a replacement of CLIP and therefore be used in any method which builds upon CLIP. Figure 1 sketches the CLOOB architecture for image-text pairs. The training set consists of $N$ pairs of embeddings $\{(\boldsymbol{x}_1, \boldsymbol{y}_1), \ldots, (\boldsymbol{x}_N, \boldsymbol{y}_N)\}$ with $\boldsymbol{X} = (\boldsymbol{x}_1, \ldots, \boldsymbol{x}_N)$ and $\boldsymbol{Y} = (\boldsymbol{y}_1, \ldots, \boldsymbol{y}_N)$, $M$ stored embeddings $\boldsymbol{U} = (\boldsymbol{u}_1, \ldots, \boldsymbol{u}_M)$, and $K$ stored embeddings $\boldsymbol{V} = (\boldsymbol{v}_1, \ldots, \boldsymbol{v}_K)$. The state or query embeddings $\boldsymbol{x}_i$ and $\boldsymbol{y}_i$ retrieve $\boldsymbol{U}_{\boldsymbol{x}_i}$ and $\boldsymbol{U}_{\boldsymbol{y}_i}$, respectively, from $\boldsymbol{U}$ — analog for retrievals from $\boldsymbol{V}$. The samples are normalized: $\|\boldsymbol{x}_i\| = \|\boldsymbol{y}_i\| = \|\boldsymbol{u}_i\| = \|\boldsymbol{v}_i\| = 1$. $\boldsymbol{U}_{\boldsymbol{x}_i}$ denotes an image-retrieved image embedding, $\boldsymbol{U}_{\boldsymbol{y}_i}$ a text-retrieved image embedding, $\boldsymbol{V}_{\boldsymbol{x}_i}$ an image-retrieved text embedding and $\boldsymbol{V}_{\boldsymbol{y}_i}$ a text-retrieved text embedding. These retrievals from modern Hopfield networks are computed as follows (Ramsauer et al., 2021):

$$\boldsymbol{U}_{\boldsymbol{x}_i} = \boldsymbol{U}\,\mathrm{softmax}(\beta\,\boldsymbol{U}^T\boldsymbol{x}_i)\,, \quad (1) \qquad \boldsymbol{V}_{\boldsymbol{x}_i} = \boldsymbol{V}\,\mathrm{softmax}(\beta\,\boldsymbol{V}^T\boldsymbol{x}_i)\,, \quad (3)$$

$$\boldsymbol{U}_{\boldsymbol{y}_i} = \boldsymbol{U}\,\mathrm{softmax}(\beta\,\boldsymbol{U}^T\boldsymbol{y}_i)\,, \quad (2) \qquad \boldsymbol{V}_{\boldsymbol{y}_i} = \boldsymbol{V}\,\mathrm{softmax}(\beta\,\boldsymbol{V}^T\boldsymbol{y}_i)\,. \quad (4)$$

The hyperparameter $\beta$ corresponds to the inverse temperature: $\beta = 0$ retrieves the average of the stored pattern, while large $\beta$ retrieves the stored pattern that is most similar to the state pattern (query).

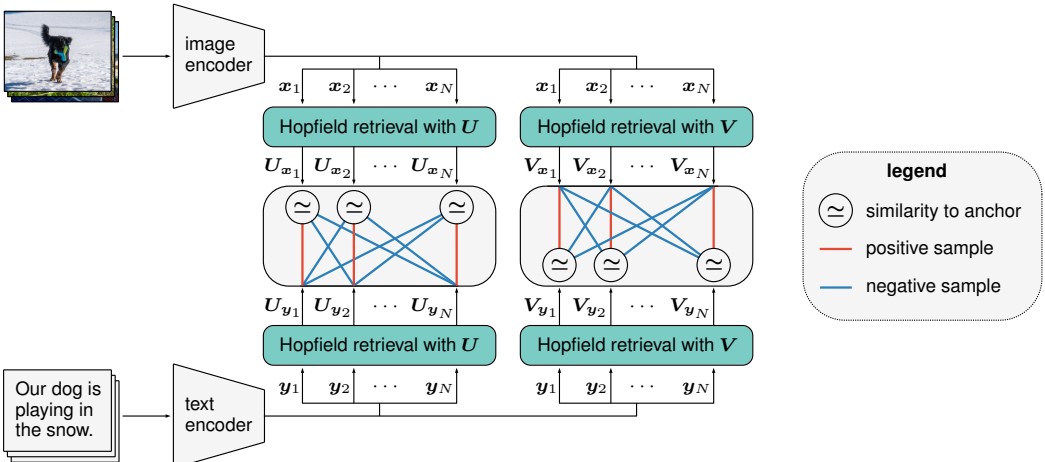

Figure 1: The CLOOB architecture for image-text pairs. The image embedding $\boldsymbol{x}_i$ and the text embedding $\boldsymbol{y}_i$ retrieve the embeddings $\boldsymbol{U_{x_i}}$ and $\boldsymbol{U_{y_i}}$, respectively, from a modern Hopfield network that stores image embeddings $\boldsymbol{U} = (\boldsymbol{u}_1, \ldots, \boldsymbol{u}_M)$ (green boxes at the left). The image-retrieved image embedding $\boldsymbol{U_{x_i}}$ serves as anchor in order to contrast the positive text-retrieved image embedding $\boldsymbol{U_{y_i}}$ with the negative text-retrieved image embedding $\boldsymbol{U_{y_j}}$ for $j \neq i$. Analogously, for the second modern Hopfield network that stores text embeddings $\boldsymbol{V} = (\boldsymbol{v}_1, \ldots, \boldsymbol{v}_K)$ (green boxes at the right).

In the InfoLOOB loss Eq. (8), CLOOB substitutes the embedded samples $\boldsymbol{x}_i$ and $\boldsymbol{y}_i$ by the normalized retrieved embedded samples. In the first term, $\boldsymbol{x}_i$ and $\boldsymbol{y}_i$ are substituted by $\boldsymbol{U_{x_i}}$ and $\boldsymbol{U_{y_i}}$, respectively, while in the second term they are substituted by $\boldsymbol{V_{x_i}}$ and $\boldsymbol{V_{y_i}}$. After retrieval, the samples are re-normalized to ensure $\|\boldsymbol{U_{x_i}}\| = \|\boldsymbol{U_{y_i}}\| = \|\boldsymbol{V_{x_i}}\| = \|\boldsymbol{V_{y_i}}\| = 1$.

We obtain the CLOOB loss function:

$$
\mathrm{L}_{\mathrm{InfoLOOB}} = -\frac{1}{N}\sum_{i=1}^{N}\ln\frac{\exp(\tau^{-1}\,\boldsymbol{U_{x_i}^T U_{y_i}})}{\sum_{j\neq i}^{N}\exp(\tau^{-1}\,\boldsymbol{U_{x_i}^T U_{y_j}})} - \frac{1}{N}\sum_{i=1}^{N}\ln\frac{\exp(\tau^{-1}\,\boldsymbol{V_{x_i}^T V_{y_i}})}{\sum_{j\neq i}^{N}\exp(\tau^{-1}\,\boldsymbol{V_{x_j}^T V_{y_i}})}\,.
\tag{5}
$$

By default, we store the minibatch in the modern Hopfield networks, that is, $\boldsymbol{U} = \boldsymbol{X}$ and $\boldsymbol{V} = \boldsymbol{Y}$. Thus, in Eq. (1) $\boldsymbol{x}_i$ can retrieve itself from $\boldsymbol{U} = \boldsymbol{X}$, but in Eq. (3) it can not retrieve itself from $\boldsymbol{V} = \boldsymbol{Y}$. Analogously, in Eq. (4) $\boldsymbol{y}_i$ can retrieve itself from $\boldsymbol{V} = \boldsymbol{Y}$, but in Eq. (2) it can not retrieve itself from $\boldsymbol{U} = \boldsymbol{X}$. By storing the embeddings of the mini-batch examples in the Hopfield memory, we do not require to compute the embeddings of additional samples via text and image encoders. However, the modern Hopfield networks can also store prototypes, templates, or proprietary samples to amplify particular embedding features via the stored samples. Either the original embeddings $\boldsymbol{x}$ and $\boldsymbol{y}$ or the retrieved embeddings $\boldsymbol{U_x}, \boldsymbol{U_y}, \boldsymbol{V_x}$, and $\boldsymbol{V_y}$ may serve for the downstream tasks, e.g. for zero-shot transfer learning.

Pseudocode 1 shows CLOOB in a PyTorch-like style. CLOOB has two major components: (i) modern Hopfield networks that alleviate CLIP's problem of insufficiently exploiting the covariance structure in the data and (ii) the InfoLOOB objective that does not suffer from InfoNCE's saturation problem. The next two sections analyze CLOOB's major components.

## 3 Modern Hopfield Networks for Enriching the Covariance Structure

We use modern Hopfield networks to amplify co-occurrences and the covariance structure. Replacing the original embeddings by retrieved embeddings reinforces features that frequently occur together in stored embeddings. Additionally, spurious co-occurrences that are peculiar to a sample are averaged out. By this means, the covariance structure is reinforced by the retrieved embeddings $\boldsymbol{U_{x_i}^T U_{y_i}}$ and $\boldsymbol{V_{x_i}^T V_{y_i}}$. The Jacobian J of the softmax $\boldsymbol{p} = \mathrm{softmax}(\beta\boldsymbol{a})$ is $\mathrm{J}(\beta\boldsymbol{a}) = \beta\left(\mathrm{diag}(\boldsymbol{p}) - \boldsymbol{pp}^T\right)$. We define the *weighted covariance* $\mathrm{Cov}(\boldsymbol{U})$, where sample $\boldsymbol{u}_i$ is drawn with probability $p_i$, as $[\mathrm{Cov}(\boldsymbol{U})]_{kl} = \left[\boldsymbol{U}\mathrm{J}(\beta\boldsymbol{a})\boldsymbol{U}^T\right]_{kl} = \beta(\sum_{i=1}^{M} p_i u_{ik} u_{il} - \sum_{i=1}^{M} p_i u_{ik} \sum_{i=1}^{M} p_i u_{il})$. The formula of the

**Pseudocode 1** CLOOB in a PyTorch-like style.

```
1   # image_encoder - ResNet                         19   # H(beta, A, B) = B.T @ softmax(beta * A @ B.T)
2   # text_encoder - Text Transformer                20   U_x = H(beta, x, x).T #[n, d_e]
3   # I[n, h, w, c] - minibatch of images            21   U_y = H(beta, y, x).T #[n, d_e]
4   # T[n, l] - minibatch of texts                   22   V_x = H(beta, x, y).T #[n, d_e]
5   # W_i[d_i, d_e] - image projection               23   V_y = H(beta, y, y).T #[n, d_e]
6   # W_t[d_t, d_e] - text projection                24
7   # beta - inverse temperature Hopfield retrieval  25   # normalize retrievals
8   # tau - temperature InfoLOOB                      26   U_x = l2_normalize(U_x) #[n, d_e]
9                                                     27   U_y = l2_normalize(U_y) #[n, d_e]
10  # extract feature representations                28   V_x = l2_normalize(V_x) #[n, d_e]
11  I_f = image_encoder(I) #[n, d_i]                 29   V_y = l2_normalize(V_y) #[n, d_e]
12  T_f = text_encoder(T) #[n, d_t]                  30
13                                                    31   # loss: info_loob(tau, anchors, samples)
14  # joint multimodal embedding                     32   # samples contain pos. and neg. embeddings
15  x = l2_normalize(I_f @ W_i) #[n, d_e]            33   loss_i = info_loob(tau, U_x, U_y)
16  y = l2_normalize(T_f @ W_t) #[n, d_e]            34   loss_t = info_loob(tau, V_y, V_x)
17                                                    35   loss = (loss_i + loss_t) * tau
18  # Hopfield retrieval H with batch stored
```

weighted covariance differs from the standard empirical covariance, since the factor $1/M$ is replaced by $p_i$. Thus, $\boldsymbol{u}_i$ is sampled with probability $p_i$ instead with probability $1/M$ (uniformly).

We apply the mean value theorem to the softmax function with mean Jacobian matrix $\mathrm{J}^{\mathrm{m}}(\beta\boldsymbol{a}) = \int_0^1 \mathrm{J}(\lambda\beta\boldsymbol{a})\ \mathrm{d}\lambda$. The mean Jacobian $\mathrm{J}^{\mathrm{m}}(\beta\boldsymbol{a})$ is a symmetric, diagonally dominant, positive semi-definite matrix with one eigenvalue of zero for eigenvector $\mathbf{1}$ and spectral norm bounded by $\|\mathrm{J}^{\mathrm{m}}\|_2 \leqslant 0.5\beta$ (see Appendix Lemma A1). According to Appendix Theorem A3, we can express $\boldsymbol{U}_{\boldsymbol{x}_i}^T\boldsymbol{U}_{\boldsymbol{y}_i}$ as:

$$\left(\bar{\boldsymbol{u}}\ +\ \mathrm{Cov}(\boldsymbol{U},\boldsymbol{x}_i)\ \boldsymbol{x}_i\right)^T\ \left(\bar{\boldsymbol{u}}\ +\ \mathrm{Cov}(\boldsymbol{U},\boldsymbol{y}_i)\ \boldsymbol{y}_i\right)\ , \tag{6}$$

where the mean is $\bar{\boldsymbol{u}} = 1/M\boldsymbol{U}\mathbf{1}$ and the weighted covariances are $\mathrm{Cov}(\boldsymbol{U},\boldsymbol{x}_i) = \boldsymbol{U}\mathrm{J}^{\mathrm{m}}(\beta\boldsymbol{U}^T\boldsymbol{x}_i)\boldsymbol{U}^T$ and $\mathrm{Cov}(\boldsymbol{U},\boldsymbol{y}_i) = \boldsymbol{U}\mathrm{J}^{\mathrm{m}}(\beta\boldsymbol{U}^T\boldsymbol{y}_i)\boldsymbol{U}^T$. The weighted covariance $\mathrm{Cov}(\boldsymbol{U},.)$ is the covariance if the stored pattern $\boldsymbol{u}_i$ is drawn according to an averaged $p_i$ given by $\mathrm{J}^{\mathrm{m}}(.)$. Maximizing the dot product $\boldsymbol{U}_{\boldsymbol{x}_i}^T\boldsymbol{U}_{\boldsymbol{y}_i}$ forces the normalized vectors $\boldsymbol{x}_i$ and $\boldsymbol{y}_i$ to agree on drawing the patterns $\boldsymbol{u}_i$ with the same probability $p_i$ in order to generate similar weighted covariance matrices $\mathrm{Cov}(\boldsymbol{U},.)$. If subsets of $\boldsymbol{U}$ have a strong covariance structure, then it can be exploited to produce large weighted covariances and, in turn, large dot products of $\boldsymbol{U}_{\boldsymbol{x}_i}^T\boldsymbol{U}_{\boldsymbol{y}_i}$. Furthermore, for a large dot product $\boldsymbol{U}_{\boldsymbol{x}_i}^T\boldsymbol{U}_{\boldsymbol{y}_i}$, $\boldsymbol{x}_i$ and $\boldsymbol{y}_i$ have to be similar to each other to extract the same direction from the covariance matrices. The above considerations for $\boldsymbol{U}_{\boldsymbol{x}_i}^T\boldsymbol{U}_{\boldsymbol{y}_i}$ analogously apply to $\boldsymbol{V}_{\boldsymbol{x}_i}^T\boldsymbol{V}_{\boldsymbol{y}_i}$.

We did not use a loss function that contains dot products like $\boldsymbol{U}_{\boldsymbol{x}_i}^T\boldsymbol{V}_{\boldsymbol{y}_i}$, because they have higher variance than the ones we have used. The dot product $\boldsymbol{U}_{\boldsymbol{x}_i}^T\boldsymbol{V}_{\boldsymbol{y}_i}$ has higher variance, since it uses $M + K$ stored patterns, whereas $\boldsymbol{U}_{\boldsymbol{x}_i}^T\boldsymbol{U}_{\boldsymbol{y}_i}$ and $\boldsymbol{V}_{\boldsymbol{x}_i}^T\boldsymbol{V}_{\boldsymbol{y}_i}$ use $M$ and $K$ stored patterns, respectively.

**Modern Hopfield networks enable to extract more covariance structure.** To demonstrate the effect of modern Hopfield networks, we computed the eigenvalues of the covariance matrix of the image and text embeddings. We counted the number of effective eigenvalues, that is, the number of eigenvalues needed to obtain 99% of the total sum of eigenvalues. Figure 2 shows the relative change of the number of effective eigenvalues compared to the respective reference epoch (the epoch before the first learning rate restart). Modern Hopfield networks consistently increase the number of effective eigenvalues during learning. Consequently, modern Hopfield networks enable to extract more covariance structure during learning, i.e. enrich the embeddings by covariances that are already in the raw multi-modal data. This enrichment of embeddings mitigates explaining away. Further details can be found in Appendix Section A.2.7.

## 4   InfoLOOB for Contrastive Learning

Modern Hopfield networks lead to a higher similarity of retrieved samples. The increased similarity exacerbates the saturation of the InfoNCE objective. To avoid the saturation of InfoNCE, CLOOB uses the "InfoLOOB" objective. The "InfoLOOB" objective is called "Leave one out upper bound" in Poole et al. (2019) and "L1Out" in Cheng et al. (2020). InfoLOOB is not established as a contrastive

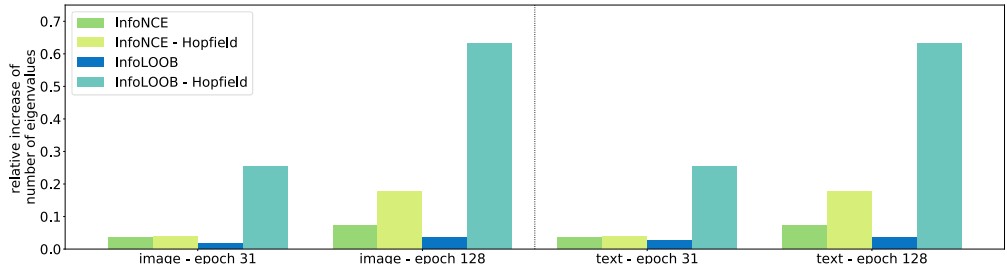

Figure 2: Relative change in the number of the effective eigenvalues of the embedding covariance matrices, which were obtained from image and text encoders at two different training points. Models with modern Hopfield networks steadily extract more covariance structure during learning.

objective, although it is a known bound. Recently, InfoLOOB was independently introduced as objective for image-to-image contrastive learning (Yeh et al., 2021).

**InfoNCE and InfoLOOB loss functions.** $N$ samples are drawn iid from $p(\boldsymbol{x}, \boldsymbol{y})$ giving the training set $\{(\boldsymbol{x}_1, \boldsymbol{y}_1), \dots, (\boldsymbol{x}_N, \boldsymbol{y}_N)\}$. For the sample $\boldsymbol{y}_1$, InfoNCE uses for the matrix of negative samples $\boldsymbol{X} = (\boldsymbol{x}_1, \dots, \boldsymbol{x}_N)$, while InfoLOOB uses $\tilde{\boldsymbol{X}} = (\boldsymbol{x}_2, \dots, \boldsymbol{x}_N)$. The matrices differ by the positive sample $\boldsymbol{x}_1$. For the score function $f(\boldsymbol{x}, \boldsymbol{y})$, we use $f(\boldsymbol{x}, \boldsymbol{y}) = \exp(\tau^{-1}\mathrm{sim}(\boldsymbol{x}, \boldsymbol{y}))$ with the similarity $\mathrm{sim}(\boldsymbol{x}, \boldsymbol{y}) = \boldsymbol{y}^T \boldsymbol{x}$ and $\tau$ as the temperature. We have the InfoNCE and InfoLOOB loss functions:

$$\mathrm{L_{InfoNCE}} = -\frac{1}{N}\sum_{i=1}^{N}\ln\frac{\exp(\tau^{-1}\,\boldsymbol{x}_i^T\boldsymbol{y}_i)}{\sum_{j=1}^{N}\exp(\tau^{-1}\,\boldsymbol{x}_i^T\boldsymbol{y}_j)} - \frac{1}{N}\sum_{i=1}^{N}\ln\frac{\exp(\tau^{-1}\,\boldsymbol{x}_i^T\boldsymbol{y}_i)}{\sum_{j=1}^{N}\exp(\tau^{-1}\,\boldsymbol{x}_j^T\boldsymbol{y}_i)}\,, \quad (7)$$

$$\mathrm{L_{InfoLOOB}} = -\frac{1}{N}\sum_{i=1}^{N}\ln\frac{\exp(\tau^{-1}\,\boldsymbol{x}_i^T\boldsymbol{y}_i)}{\sum_{j\neq i}^{N}\exp(\tau^{-1}\,\boldsymbol{x}_i^T\boldsymbol{y}_j)} - \frac{1}{N}\sum_{i=1}^{N}\ln\frac{\exp(\tau^{-1}\,\boldsymbol{x}_i^T\boldsymbol{y}_i)}{\sum_{j\neq i}^{N}\exp(\tau^{-1}\,\boldsymbol{x}_j^T\boldsymbol{y}_i)}\,. \quad (8)$$

We abbreviate $\boldsymbol{y} = \boldsymbol{y}_1$ leading to the pair $(\boldsymbol{x}_1, \boldsymbol{y})$ and the negatives $\tilde{\boldsymbol{X}} = (\boldsymbol{x}_2, \dots, \boldsymbol{x}_N)$. In the second sum of the losses in Eq. 7 and Eq. 8, we consider only the first term, respectively:

$$\mathrm{L_{InfoNCE}}(\boldsymbol{y}) = -\ln\frac{\overbrace{\exp(\tau^{-1}\,\boldsymbol{x}_1^T\boldsymbol{y})}^{a}}{\underbrace{\exp(\tau^{-1}\,\boldsymbol{x}_1^T\boldsymbol{y})}_{a} + \underbrace{\sum_{j=2}^{N}\exp(\tau^{-1}\,\boldsymbol{x}_j^T\boldsymbol{y})}_{b}}\,, \quad (9)$$

$$\mathrm{L_{InfoLOOB}}(\boldsymbol{y}) = -\ln\frac{\overbrace{\exp(\tau^{-1}\,\boldsymbol{x}_1^T\boldsymbol{y})}^{a}}{\underbrace{\sum_{j=2}^{N}\exp(\tau^{-1}\,\boldsymbol{x}_j^T\boldsymbol{y})}_{b}}\,. \quad (10)$$

In analogy to Wang & Isola (2020), $a$ is called the "alignment score" that measures the similarity of matched pairs and $b$ the "uniformity penalty" that measures the similarity of unmatched pairs.

**Gradients of InfoNCE and InfoLOOB loss functions.** Eq. (9) and Eq. (10) are equal to

$$-\tau^{-1}\boldsymbol{y}^T\boldsymbol{x}_1 + \tau^{-1}\mathrm{lse}(\tau^{-1}, \boldsymbol{X}^T\boldsymbol{y})\,, \qquad -\tau^{-1}\boldsymbol{y}^T\boldsymbol{x}_1 + \tau^{-1}\mathrm{lse}(\tau^{-1}, \tilde{\boldsymbol{X}}^T\boldsymbol{y})\,,$$

where lse is the log-sum-exp function (see Eq. (A73) in the Appendix).

The gradients of Eq. (9) and Eq. (10) with respect to $\boldsymbol{y}$ are

$$-\tau^{-1}\boldsymbol{x}_1 + \tau^{-1}\boldsymbol{X}\mathrm{softmax}(\tau^{-1}\boldsymbol{X}^T\boldsymbol{y})\,, \qquad -\tau^{-1}\boldsymbol{x}_1 + \tau^{-1}\tilde{\boldsymbol{X}}\mathrm{softmax}(\tau^{-1}\tilde{\boldsymbol{X}}^T\boldsymbol{y})\,.$$

Using $\boldsymbol{p} = (p_1, \dots, p_N)^T = \mathrm{softmax}(\tau^{-1}\boldsymbol{X}^T\boldsymbol{y})$, the gradient of InfoNCE with respect to $\boldsymbol{y}$ is

$$\frac{\partial\mathrm{L_{InfoNCE}}(\boldsymbol{y})}{\partial\boldsymbol{y}} = -\tau^{-1}(1-p_1)(\boldsymbol{x}_1 - \tilde{\boldsymbol{X}}\mathrm{softmax}(\tau^{-1}\tilde{\boldsymbol{X}}^T\boldsymbol{y})) = (1-p_1)\frac{\partial\mathrm{L_{InfoLOOB}}(\boldsymbol{y})}{\partial\boldsymbol{y}}\,.$$

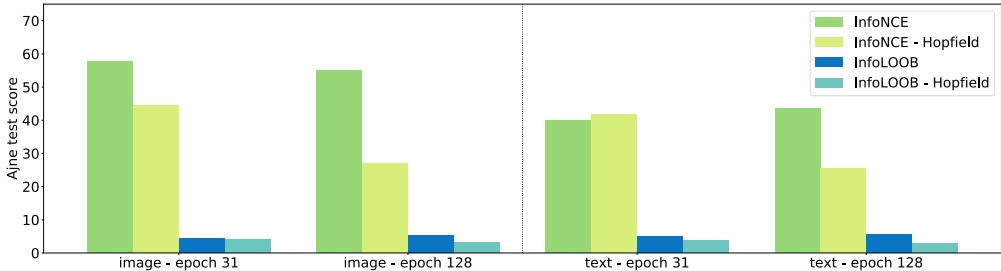

Figure 3: Ajne uniformity test statistics for image and text embeddings for two different epochs during training. A high test statistic indicates low uniformity of an embedding. Models trained with the InfoLOOB objective develop more uniform image and text embeddings on the hypersphere.

By and large, the gradient of InfoNCE is scaled by $(1 - p_1)$ compared to the gradient of InfoLOOB, where $p_1$ is the softmax similarity between the anchor $\boldsymbol{y}$ and the positive sample $\boldsymbol{x}_1$. Consequently, InfoNCE is saturating with increasing similarity between the anchor and the positive sample. For more details we refer to Appendix Section A.1.4.

This saturation of InfoNCE motivated the use of the InfoLOOB objective in order to decrease the uniformity penalty even for large alignment scores. The uniformity penalty decreases since learning does not stall and the most prominent features become down-scaled which makes negative examples less similar to the anchor sample. The InfoNCE objective Eq. 9 has the form $a/(a + b)$, while the InfoLOOB objective Eq. 10 has the form $a/b$. InfoLOOB does not saturate and keeps decreasing the uniformity penalty $b$. Figure 3 shows how InfoLOOB leads to an increase in the uniformity of image and text embeddings on the sphere, which is described by the statistics of the uniformity test of Ajne that was extended by Prentice (Ajne, 1968; Prentice, 1978). Higher uniformity on the sphere correlates with a lower uniformity penalty $b$. For more details we refer to Appendix Section A.2.7.

**Theoretical justification for optimizing InfoLOOB.** The InfoNCE information is a lower bound on the mutual information, which was proven by Poole et al. (2019). In the Appendix Section A.1, we elaborate more on theoretical properties of the bounds and properties of the objective functions. Specifically, we show that InfoLOOB with neural networks is not an upper bound on the mutual information. Thus, unlike hitherto approaches to contrastive learning we use InfoLOOB as an objective, since it does not suffer from saturation effects as InfoNCE.

## 5    Experiments

CLOOB is compared to CLIP with respect to zero-shot transfer learning performance on two pre-training datasets. The first dataset, Conceptual Captions (CC) (Sharma et al., 2018), has a very rich textual description of images but only three million image-text pairs. The second dataset, a subset of YFCC100M (Thomee et al., 2016), has 15 million image-text pairs but the textual description is less rich than for CC and often vacuous. For both pre-training datasets, the downstream zero-shot transfer learning performance is tested on seven image classification datasets.

### 5.1    Conceptual Captions Pre-training

**Pre-training dataset.** The Conceptual Captions (CC) (Sharma et al., 2018) dataset contains 2.9 million images with high-quality captions. Images and their captions have been gathered from the web via an automated process and have a wide variety of content. Raw descriptions of images are from the *alt-text* HTML attribute.

**Methods.** The CLOOB implementation is based on OpenCLIP (Ilharco et al., 2021), which achieves results equivalent to CLIP on the YFCC dataset (see Section 5.2). OpenCLIP also reports results on the CC dataset. As CLIP does not train models on CC, we report results from this reimplementation as baseline. Analogously to Radford et al. (2021, Section 2.4), we used the modified ResNet (He et al., 2016) and BERT (Devlin et al., 2018, 2019) architectures to encode image and text input. We used the ResNet encoder ResNet-50 for experiments on CC.

Table 1: Zero-shot results for models trained on CC with ResNet-50 vision encoders for two different checkpoints. Results are given as mean accuracy over 5 runs. Statistically significant results are shown in bold. CLIP and CLOOB were trained for 31 epochs while CLIP* and CLOOB* were trained for 128 epochs. In the majority of tasks CLOOB significantly outperforms CLIP.

| Dataset | CLIP RN-50 | CLOOB RN-50 | CLIP* RN-50 | CLOOB* RN-50 |
|---|---|---|---|---|
| Birdsnap | $2.26 \pm 0.20$ | $\mathbf{3.06 \pm 0.30}$ | $2.8 \pm 0.16$ | $\mathbf{3.24 \pm 0.31}$ |
| Country211 | $0.67 \pm 0.11$ | $0.67 \pm 0.05$ | $0.7 \pm 0.04$ | $0.73 \pm 0.05$ |
| Flowers102 | $12.56 \pm 0.38$ | $13.45 \pm 1.19$ | $13.32 \pm 0.43$ | $14.36 \pm 1.17$ |
| GTSRB | $7.66 \pm 1.07$ | $6.38 \pm 2.11$ | $8.96 \pm 1.70$ | $7.03 \pm 1.22$ |
| UCF101 | $20.98 \pm 1.55$ | $22.26 \pm 0.72$ | $21.63 \pm 0.65$ | $\mathbf{23.03 \pm 0.85}$ |
| Stanford Cars | $0.91 \pm 0.10$ | $\mathbf{1.23 \pm 0.10}$ | $0.99 \pm 0.16$ | $\mathbf{1.41 \pm 0.32}$ |
| ImageNet | $20.33 \pm 0.28$ | $\mathbf{23.97 \pm 0.15}$ | $21.3 \pm 0.42$ | $\mathbf{25.67 \pm 0.22}$ |
| ImageNet V2 | $20.24 \pm 0.50$ | $\mathbf{23.59 \pm 0.15}$ | $21.24 \pm 0.22$ | $\mathbf{25.49 \pm 0.11}$ |

**Hyperparameter selection and learning schedule.** The hyperparameter values of OpenCLIP were used as default, concretely, a learning rate of $1 \times 10^{-3}$ and a weight decay of $0.1$ for the Adam optimizer (Kingma et al., 2014) with decoupled weight decay regularization (Loshchilov & Hutter, 2019). Deviating from OpenCLIP, we used a batch size of $512$ due to computational restraints, which did not change the performance. The learning rate scheduler for all experiments was cosine annealing with warmup and hard restarts (Loshchilov & Hutter, 2017). We report the hyperparameter $\tau$ (default $0.07$) from CLIP as $\tau^{-1}$ of $14.3$ to be in the same regime as the hyperparameter $\beta$ for the modern Hopfield networks. The main hyperparameter search for CLOOB (also for YFCC pre-training in the next section) was done with ResNet-50 as the vision encoder. Learnable $\tau^{-1}$ in combination with the InfoLOOB loss results in undesired learning behavior (see Appendix Section A.1.4). Therefore, we set $\tau^{-1}$ to a fixed value of 30, which was determined via a hyperparameter search (see Appendix Section A.2.2). For modern Hopfield networks, the hyperparameter $\beta$ was set to $8$. Further we scaled the loss $L_{\text{InfoLOOB}}$ with $\tau$ to remove the factor $\tau^{-1}$ from the gradients (see Appendix Section A.1.4) resulting in the loss function $\tau L_{\text{InfoLOOB}}$.

**Evaluation metrics: Zero-shot transfer learning.** We evaluated and compared both CLIP and CLOOB on their zero-shot transfer learning capabilities on the following downstream image classification tasks. Birdsnap (Berg et al., 2014) contains images of 500 different North American bird species. The Country211 (Radford et al., 2021) dataset consists of photos across 211 countries and is designed to test the geolocalization capability of visual representations. Flowers102 (Nilsback & Zisserman, 2008) is a dataset containing images of 102 flower species. GTSRB (Stallkamp et al., 2011) contains images for classification of German traffic signs. UCF101 (Soomro et al., 2012) is a video dataset with short clips for action recognition. For UCF101 we followed the procedure reported in CLIP and extract the middle frame of every video to assemble the dataset. Stanford Cars (Krause et al., 2013) contains images of 196 types of cars. ImageNet (Deng et al., 2009) is a large scale image classification dataset with images across 1,000 classes. ImageNetv2 (Recht et al., 2019) consists of three new test sets with 10,000 images each for the ImageNet benchmark. For further details see Appendix Section A.2.3.

**Results.** We employed the same evaluation strategy and used the prompts as published in CLIP (see Appendix Section A.2.3). We report zero-shot results from two checkpoints in Table 1. CLIP and CLOOB were trained for a comparable number of epochs used in CLIP (see Appendix Section A.2.2) while CLIP* and CLOOB* were trained until evaluation performance plateaued (epoch 128). In both cases CLOOB significantly outperforms CLIP on the majority of tasks or matches its performance. Statistical significance of these results was assessed by an unpaired Wilcoxon test on a $5\%$ level.

## 5.2 YFCC Pre-training

**Pre-training dataset.** To be comparable to the CLIP results, we used the same subset of 15 million samples from the YFCC100M dataset (Thomee et al., 2016) as in Radford et al. (2021), which we refer to as YFCC. YFCC was created by filtering YFCC100M for images which contain natural language descriptions and/or titles in English. It was not filtered by quality of the captions, therefore the textual descriptions are less rich and contain superfluous information. The dataset with 400

million samples used to train the CLIP models in Radford et al. (2021) has not been released and, thus, is not available for comparison. Due to limited computational resources we were unable to compare CLOOB to CLIP on other datasets of this size.

**Methods.** Besides experiments with a ResNet-50 image encoder, we additionally conducted experiments with the larger ResNet variants ResNet-101 and ResNet-50x4. In addition to the comparison of CLOOB and CLIP based on the OpenCLIP reimplementation (Ilharco et al., 2021), we include the original CLIP results (Radford et al., 2021, Table 12).

**Hyperparameter selection.** Hyperparameters were the same as for the Conceptual Captions dataset, except learning rate, batch size, and $\beta$. For modern Hopfield networks, the hyperparameter $\beta$ was set to 14.3, which is default for $\tau^{-1}$ in Radford et al. (2021). Furthermore, the learning rate was set to $5 \times 10^{-4}$ and the batch size to 1024 as used in OpenCLIP of Ilharco et al. (2021). All models were trained for 28 epochs. For further details see Appendix Section A.2.2.

**Evaluation metrics.** As in the previous experiment, methods were again evaluated at their zero-shot transfer learning capabilities on downstream tasks.

Table 2: Results of CLIP and CLOOB trained on YFCC with ResNet-50 encoder. Except for one linear probing dataset, CLOOB consistently outperforms CLIP at all tasks.

| | Linear Probing | | Zero-Shot | |
| | CLIP | CLOOB | CLIP | CLOOB |
| Dataset | (OpenAI) | (ours) | (OpenAI) | (ours) |
|---|---|---|---|---|
| Birdsnap | 47.4 | **56.2** | 19.9 | **28.9** |
| Country211 | **23.1** | 20.6 | 5.2 | **7.9** |
| Flowers102 | 94.4 | **96.1** | 48.6 | **55.1** |
| GTSRB | 66.8 | **78.9** | 6.9 | **8.1** |
| UCF101 | 69.2 | **72.3** | 22.9 | **25.3** |
| Stanford Cars | 31.4 | **37.7** | 3.8 | **4.1** |
| ImageNet | 62.0 | **65.7** | 31.3 | **35.7** |
| ImageNet V2 | - | 58.7 | - | 34.6 |

Table 3: Zero-shot results for the CLIP reimplementation and CLOOB using different ResNet architectures trained on YFCC. CLOOB outperforms CLIP in 7 out of 8 tasks using ResNet-50 encoders. With larger ResNet encoders CLOOB outperforms CLIP on all tasks. The performance of CLOOB scales with increased encoder size.

| | RN-50 | | RN-101 | | RN-50x4 | |
| Dataset | CLIP | CLOOB | CLIP | CLOOB | CLIP | CLOOB |
|---|---|---|---|---|---|---|
| Birdsnap | 21.8 | **28.9** | 22.6 | **30.3** | 20.8 | **32.0** |
| Country211 | 6.9 | **7.9** | 7.8 | **8.5** | 8.1 | **9.3** |
| Flowers102 | 48.0 | **55.1** | 48.0 | **55.3** | 50.1 | **54.3** |
| GTSRB | 7.9 | **8.1** | 7.4 | **11.6** | 9.4 | **11.8** |
| UCF101 | **27.2** | 25.3 | 28.6 | **28.8** | 31.0 | **31.9** |
| Stanford Cars | 3.7 | **4.1** | 3.8 | **5.5** | 3.5 | **6.1** |
| ImageNet | 34.6 | **35.7** | 35.3 | **37.1** | 37.7 | **39.0** |
| ImageNet V2 | 33.4 | **34.6** | 34.1 | **35.6** | 35.9 | **37.3** |

**Results.** Table 2 provides results of the original CLIP and CLOOB trained on YFCC. Results on zero-shot downstream tasks show that CLOOB outperforms CLIP on all 7 tasks (ImageNet V2 results have not been reported in Radford et al. (2021)). Similarly, CLOOB outperforms CLIP on 6 out of 7 tasks for linear probing. Results of CLOOB and the CLIP reimplementation of OpenCLIP are given in Table 3. CLOOB exceeds the CLIP reimplementation in 7 out of 8 tasks for zero-shot classification using ResNet-50 encoders. With larger ResNet encoders, CLOOB outperforms CLIP on all tasks. Furthermore, the experiments with larger vision encoder networks show that CLOOB performance increases with network size. Results of CLOOB zero-shot classification on all datasets are shown in Appendix Section A.2.4.

### 5.3 Ablation studies

CLOOB has two new major components compared to CLIP: (1) modern Hopfield networks and (2) the InfoLOOB objective instead of the InfoNCE objective. To assess effects of the new major components of CLOOB, we performed ablation studies.

**Modern Hopfield networks.** Modern Hopfield networks amplify the covariance structure in the data via the retrievals. Ablation studies confirm this amplification as modern Hopfield networks consistently increase the number of effective eigenvalues of the embedding covariance matrices during learning. Figure 2 shows the relative change of the number of effective eigenvalues compared to the respective reference epoch, which is the epoch before the first learning rate restart. These results indicate that modern Hopfield networks steadily extract more covariance structure during learning. Modern Hopfield networks induce higher similarity of retrieved samples, which in turn leads to stronger saturation of the InfoNCE objective. As a result, we observe low uniformity (see Figure 3) and a small number of effective eigenvalues (see Appendix Figure A1).

**Modern Hopfield networks with InfoLOOB.** CLOOB counters the saturation of InfoNCE by using the InfoLOOB objective. The effectiveness of InfoLOOB manifests in an increased uniformity measure of image and text embeddings on the sphere, as shown in Figure 3. The ablation study verifies that modern Hopfield networks together with InfoLOOB have a strong synergistic effect.

**InfoLOOB.** However, using solely InfoLOOB results in overfitting of the alignment score. This overfitting occasionally leads to high similarities of unmatched pairs (see Figure 4), which may decreases the zero-shot downstream performance. The reason for this is that the top-1 evaluation metric is very sensitive to occasionally high similarities of prompts of the incorrect class. Yeh et al. (2021) and Zhang et al. (2022) reported similar observations of overfitting.

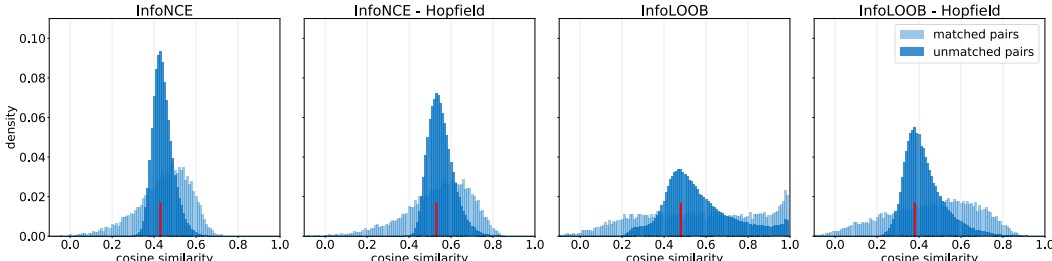

Figure 4: Distribution of the cosine similarity of matched pairs and the cosine similarity of the 10 unmatched pairs that have the highest similarity score with the anchor. Modern Hopfield networks lead to higher values of both matched and unmatched pairs. InfoLOOB without Hopfield has high similarity scores of the matched pairs which correlate with high similarity scores of the top-10 unmatched pairs. In contrast, InfoLOOB with Hopfield does not suffer from this overfitting problem.

CLOOB balances the overfitting of InfoLOOB with the underfitting of modern Hopfield networks and remains in effective learning regimes. For more details and further ablation studies see Appendix Section A.2.1.

## 6 Conclusion

We have introduced "Contrastive Leave One Out Boost" (CLOOB), which combines modern Hopfield networks with the InfoLOOB objective. Modern Hopfield networks enable CLOOB to extract additional covariance structure in the data. This allows for building more relevant features in the embedding space, mitigating the explaining away problem. We show that InfoLOOB avoids the saturation problem of InfoNCE. Additionally, we theoretically justify the use of the InfoLOOB objective for contrastive learning and suggest it as an alternative to InfoNCE. At seven zero-shot transfer learning tasks, the novel CLOOB was compared to CLIP after pre-training on the Conceptual Captions and the YFCC dataset. CLOOB consistently outperforms CLIP at zero-shot transfer learning across all considered architectures and datasets.

## Acknowledgments

The ELLIS Unit Linz, the LIT AI Lab, the Institute for Machine Learning, are supported by the Federal State Upper Austria. IARAI is supported by Here Technologies. We thank the projects AI-MOTION (LIT-2018-6-YOU-212), AI-SNN (LIT-2018-6-YOU-214), DeepFlood (LIT-2019-8-YOU-213), Medical Cognitive Computing Center (MC3), INCONTROL-RL (FFG-881064), PRIMAL (FFG-873979), S3AI (FFG-872172), DL for GranularFlow (FFG-871302), AIRI FG 9-N (FWF-36284, FWF-36235), ELISE (H2020-ICT-2019-3 ID: 951847). We thank Audi.JKU Deep Learning Center, TGW LOGISTICS GROUP GMBH, Silicon Austria Labs (SAL), FILL Gesellschaft mbH, Anyline GmbH, Google, ZF Friedrichshafen AG, Robert Bosch GmbH, UCB Biopharma SRL, Merck Healthcare KGaA, Verbund AG, Software Competence Center Hagenberg GmbH, TÜV Austria, Frauscher Sensonic and the NVIDIA Corporation.

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
