# A Appendix

This appendix consists of four sections (A.1–A.4). Section A.1 provides the theoretical properties of InfoLOOB and InfoNCE. It is shown how to derive that InfoNCE is a lower bound on mutual information. Further it is shown how to derive that InfoLOOB is an upper bound on mutual information. The proposed loss function and its gradients are discussed. Section A.2 provides details on the experiments. Section A.3 briefly reviews continuous modern Hopfield networks. Section A.4 discusses further related work.

## Contents of the appendix

## List of theorems

## List of definitions

## List of figures

## List of tables

## A.1 InfoLOOB vs. InfoNCE

### A.1.1 InfoNCE: Lower Bound on Mutual Information

We derive a lower bound on the mutual information between random variables $X$ and $Y$ distributed according to $p(\boldsymbol{x}, \boldsymbol{y})$. The mutual information $\mathrm{I}(X \; ; \; Y)$ between random variables $X$ and $Y$ is

$$\mathrm{I}(X \; ; \; Y) \; = \; \mathrm{E}_{p(\boldsymbol{x},\boldsymbol{y})} \left[ \ln \frac{p(\boldsymbol{x}, \boldsymbol{y})}{p(\boldsymbol{x}) \, p(\boldsymbol{y})} \right] \; = \; \mathrm{E}_{p(\boldsymbol{x},\boldsymbol{y})} \left[ \ln \frac{p(\boldsymbol{x} \mid \boldsymbol{y})}{p(\boldsymbol{x})} \right] \; = \; \mathrm{E}_{p(\boldsymbol{x},\boldsymbol{y})} \left[ \ln \frac{p(\boldsymbol{y} \mid \boldsymbol{x})}{p(\boldsymbol{y})} \right] \; . \tag{A1}$$

"InfoNCE" has been introduced in van den Oord et al. (2018) and is a *multi-sample bound*. In the setting introduced in van den Oord et al. (2018), we have an anchor sample $\boldsymbol{y}$ given. For the anchor sample $\boldsymbol{y}$ we draw a positive sample $\boldsymbol{x}_1$ according to $p(\boldsymbol{x}_1 \mid \boldsymbol{y})$. Next, we draw a set $\tilde{X} = \{\boldsymbol{x}_2, \ldots, \boldsymbol{x}_N\}$ according to $p(\tilde{X})$, which are $n - 1$ negative samples drawn iid according to $p(\boldsymbol{x})$. We have drawn a set $X = \{\boldsymbol{x}_1, \boldsymbol{x}_2, \ldots, \boldsymbol{x}_N\}$ according to $p(X \mid \boldsymbol{y})$, which is one positive sample $\boldsymbol{x}_1$ drawn by $p(\boldsymbol{x}_1 \mid \boldsymbol{y})$ and $N - 1$ negative samples $\{\boldsymbol{x}_2, \ldots, \boldsymbol{x}_N\}$ drawn iid according to $p(\boldsymbol{x})$.

The InfoNCE with probabilities is

$$\mathrm{I}_{\mathrm{InfoNCE}}(X_1 \; ; \; Y) \; = \; \mathrm{E}_{p(\boldsymbol{y})} \left[ \mathrm{E}_{p(X|\boldsymbol{y})} \left[ \ln \left( \frac{p(\boldsymbol{y} \mid \boldsymbol{x}_1)}{\frac{1}{N} \sum_{i=1}^{N} p(\boldsymbol{y} \mid \boldsymbol{x}_i)} \right) \right] \right] \; , \tag{A2}$$

where we inserted the factor $\frac{1}{N}$ in contrast to the original version in van den Oord et al. (2018), where we followed Poole et al. (2019); Tschannen et al. (2019); Cheng et al. (2020); Chen et al. (2021).

The InfoNCE with score function $f(\boldsymbol{x}, \boldsymbol{y})$ is

$$\mathrm{I}_{\mathrm{InfoNCE}}(X_1 \; ; \; Y) \; = \; \mathrm{E}_{p(\boldsymbol{y})} \left[ \mathrm{E}_{p(X|\boldsymbol{y})} \left[ \ln \left( \frac{f(\boldsymbol{x}_1, \boldsymbol{y})}{\frac{1}{N} \sum_{i=1}^{N} f(\boldsymbol{x}_i, \boldsymbol{y})} \right) \right] \right] \; . \tag{A3}$$

The InfoNCE with probabilities can be rewritten as:

$$\mathrm{I}_{\mathrm{InfoNCE}}(X_1 \; ; \; Y) \; = \; \mathrm{E}_{p(\boldsymbol{y})} \left[ \mathrm{E}_{p(X|\boldsymbol{y})} \left[ \ln \left( \frac{p(\boldsymbol{y} \mid \boldsymbol{x}_1)}{\frac{1}{N} \sum_{i=1}^{N} p(\boldsymbol{y} \mid \boldsymbol{x}_i)} \right) \right] \right] \tag{A4}$$

$$= \; \mathrm{E}_{p(\boldsymbol{y})} \left[ \mathrm{E}_{p(X|\boldsymbol{y})} \left[ \ln \left( \frac{\frac{p(\boldsymbol{y}|\boldsymbol{x}_1)}{p(\boldsymbol{y})}}{\frac{1}{N} \sum_{i=1}^{N} \frac{p(\boldsymbol{y}|\boldsymbol{x}_i)}{p(\boldsymbol{y})}} \right) \right] \right]$$

$$= \; \mathrm{E}_{p(\boldsymbol{y})} \left[ \mathrm{E}_{p(X|\boldsymbol{y})} \left[ \ln \left( \frac{\frac{p(\boldsymbol{x}_1|\boldsymbol{y})}{p(\boldsymbol{x}_1)}}{\frac{1}{N} \sum_{i=1}^{N} \frac{p(\boldsymbol{x}_i|\boldsymbol{y})}{p(\boldsymbol{x}_i)}} \right) \right] \right] \; .$$

This is the InfoNCE with $f(\boldsymbol{x}, \boldsymbol{y}) = p(\boldsymbol{y} \mid \boldsymbol{x})$.

**Set of pairs.** The InfoNCE can be written in a different setting Poole et al. (2019), which is used in most implementations. We sample $N$ pairs independently from $p(\boldsymbol{x}, \boldsymbol{y})$, which gives $Z = \{(\boldsymbol{x}_1, \boldsymbol{y}_1), (\boldsymbol{x}_2, \boldsymbol{y}_2), \ldots, (\boldsymbol{x}_N, \boldsymbol{y}_N)\}$. The InfoNCE is then

$$\mathrm{I}_{\mathrm{InfoNCE}}(X \; ; \; Y) \; = \; \mathrm{E}_{p(X|\boldsymbol{y})} \left[ \frac{1}{N} \sum_{i=1}^{N} \ln \left( \frac{f(\boldsymbol{x}_i, \boldsymbol{y}_i)}{\frac{1}{N} \sum_{j=1}^{N} f(\boldsymbol{x}_j, \boldsymbol{y}_i)} \right) \right] \; . \tag{A5}$$

Following van den Oord et al. (2018) we have

$$
\begin{aligned}
\mathrm{I}_{\mathrm{InfoNCE}}(X_1 \; ; \; Y) \; &= \; \mathrm{E}_{p(\boldsymbol{y})}\left[\mathrm{E}_{p(X|\boldsymbol{y})}\left[\ln\left(\frac{\frac{p(\boldsymbol{y}|\boldsymbol{x}_1)}{p(\boldsymbol{y})}}{\frac{1}{N}\sum_{i=1}^{N}\frac{p(\boldsymbol{y}|\boldsymbol{x}_i)}{p(\boldsymbol{y})}}\right)\right]\right] && \text{(A6)} \\
&= \; \mathrm{E}_{p(\boldsymbol{y})}\left[\mathrm{E}_{p(X|\boldsymbol{y})}\left[\ln\left(\frac{\frac{p(\boldsymbol{x}_1|\boldsymbol{y})}{p(\boldsymbol{x}_1)}}{\frac{1}{N}\sum_{i=1}^{N}\frac{p(\boldsymbol{x}_i|\boldsymbol{y})}{p(\boldsymbol{x}_i)}}\right)\right]\right] \\
&= \; \mathrm{E}_{p(\boldsymbol{y})}\left[\mathrm{E}_{p(X|\boldsymbol{y})}\left[\ln\left(\frac{p(\boldsymbol{x}_1\mid\boldsymbol{y})\prod_{l=2}^{N}p(\boldsymbol{x}_l)}{\sum_{i=1}^{N}p(\boldsymbol{x}_i\mid\boldsymbol{y})\prod_{l\neq i}p(\boldsymbol{x}_l)}\right)\right]\right] \; + \; \ln(N) \\
&= \; \mathrm{E}_{p(\boldsymbol{y})}\left[\mathrm{E}_{p(X|\boldsymbol{y})}\left[\ln p(i=1\mid X,\boldsymbol{y})\right]\right] \; + \; \ln(N) \,,
\end{aligned}
$$

where $p(i=1\mid X,\boldsymbol{y})$ is the probability that sample $\boldsymbol{x}_1$ is the positive sample if we know there exists exactly one positive sample in $X$.

The InfoNCE is a lower bound on the mutual information. The following inequality is from van den Oord et al. (2018):

$$
\begin{aligned}
\mathrm{I}(X_1 \; ; \; Y) \; &= \; \mathrm{E}_{p(\boldsymbol{y})}\left[\mathrm{E}_{p(\boldsymbol{x}_1|\boldsymbol{y})}\left[\ln\left(\frac{p(\boldsymbol{x}_1\mid\boldsymbol{y})}{p(\boldsymbol{x}_1)}\right)\right]\right] && \text{(A7)} \\
&= \; \mathrm{E}_{p(\boldsymbol{y})}\left[\mathrm{E}_{p(\boldsymbol{x}_1|\boldsymbol{y})}\left[-\ln\left(\frac{p(\boldsymbol{x}_1)}{p(\boldsymbol{x}_1\mid\boldsymbol{y})}\right)\right]\right] \\
&\geq \; \mathrm{E}_{p(\boldsymbol{y})}\left[\mathrm{E}_{p(\boldsymbol{x}_1|\boldsymbol{y})}\left[-\ln\left(\frac{1}{N}+\frac{p(\boldsymbol{x}_1)}{p(\boldsymbol{x}_1\mid\boldsymbol{y})}\right)\right]\right] \\
&\approx \; \mathrm{E}_{p(\boldsymbol{y})}\left[\mathrm{E}_{p(X|\boldsymbol{y})}\left[-\ln\left(\frac{1}{N}+\frac{1}{N}\frac{p(\boldsymbol{x}_1)}{p(\boldsymbol{x}_1\mid\boldsymbol{y})}\sum_{i=2}^{N}\frac{p(\boldsymbol{x}_i\mid\boldsymbol{y})}{p(\boldsymbol{x}_i)}\right)\right]\right] \\
&= \; \mathrm{E}_{p(\boldsymbol{y})}\left[\mathrm{E}_{p(X|\boldsymbol{y})}\left[\ln\left(\frac{\frac{p(\boldsymbol{x}_1|\boldsymbol{y})}{p(\boldsymbol{x}_1)}}{\frac{1}{N}\frac{p(\boldsymbol{x}_1|\boldsymbol{y})}{p(\boldsymbol{x}_1)}+\frac{1}{N}\sum_{i=2}^{N}\frac{p(\boldsymbol{x}_i|\boldsymbol{y})}{p(\boldsymbol{x}_i)}}\right)\right]\right] \\
&= \; \mathrm{I}_{\mathrm{InfoNCE}}(X_1 \; ; \; Y) \,,
\end{aligned}
$$

where the "$\geq$" is obtained by bounding $\ln(1/N+a)$ by $\ln(a)$, which gives a bound that is not very tight, since $a=\frac{p(\boldsymbol{x}_1)}{p(\boldsymbol{x}_1|\boldsymbol{y})}$ can become small. However for the "$\approx$" van den Oord et al. (2018) have to assume

$$
\frac{1}{N}\sum_{i=2}^{N}\frac{p(\boldsymbol{x}_i\mid\boldsymbol{y})}{p(\boldsymbol{x}_i)} \; = \; \frac{1}{N}\sum_{i=2}^{N}\frac{p(\boldsymbol{y}\mid\boldsymbol{x}_i)}{p(\boldsymbol{y})} \; \geq \; 1 \,, \tag{A8}
$$

which is unclear how to ensure.

For a proof of this bound see Poole et al. (2019).

We assumed that for the anchor sample $\boldsymbol{y}$ a positive sample $\boldsymbol{x}_1$ has been drawn according to $p(\boldsymbol{x}_1\mid\boldsymbol{y})$. A set $\tilde{X}=\{\boldsymbol{x}_2,\ldots,\boldsymbol{x}_N\}$ of negative samples is drawn according to $p(\boldsymbol{x})$. Therefore, we have a set $X=\{\boldsymbol{x}_1,\boldsymbol{x}_2,\ldots,\boldsymbol{x}_N\}$ that is drawn with one positive sample $\boldsymbol{x}_1$ and $N-1$ negative samples $\tilde{X}=\{\boldsymbol{x}_2,\ldots,\boldsymbol{x}_N\}$. We have

$$
p(\tilde{X}) \; = \; \prod_{i=2}^{N}p(\boldsymbol{x}_i) \,, \tag{A9}
$$

$$
p(X\mid\boldsymbol{y}) \; = \; p(\boldsymbol{x}_1\mid\boldsymbol{y})\prod_{i=2}^{N}p(\boldsymbol{x}_i) \,, \tag{A10}
$$

$$
p(X) \; = \; \prod_{i=1}^{N}p(\boldsymbol{x}_i) \,. \tag{A11}
$$

Next, we present a theorem that shows this bound, where we largely follow Poole et al. (2019) in the proof. In contrast to Poole et al. (2019), we do not use the NWJ bound Nguyen et al. (2010). The mutual information is

$$\mathrm{I}(X_1 \; ; \; Y) \; = \; \mathrm{E}_{p(\boldsymbol{x}_1, \boldsymbol{y})} \left[ \ln \left( \frac{p(\boldsymbol{x}_1 \mid \boldsymbol{y})}{p(\boldsymbol{x}_1)} \right) \right] \; . \tag{A12}$$

**Theorem A1** (InfoNCE lower bound). *InfoNCE with score function $f(\boldsymbol{x}, \boldsymbol{y})$ according to Eq. (A3) is a lower bound on the mutual information.*

$$\mathrm{I}(X_1 \; ; \; Y) \; \geq \; \mathrm{E}_{p(\boldsymbol{y})p(X|\boldsymbol{y})} \left[ \ln \left( \frac{f(\boldsymbol{x}_1, \boldsymbol{y})}{\frac{1}{N} \sum_{i=1}^{N} f(\boldsymbol{x}_i, \boldsymbol{y})} \right) \right] \; = \; \mathrm{I}_{\mathrm{InfoNCE}}(X_1 \; ; \; Y) \; . \tag{A13}$$

*InfoNCE with probabilities according to Eq. (A2) is a lower bound on the mutual information.*

$$\mathrm{I}(X_1 \; ; \; Y) \; \geq \; \mathrm{E}_{p(\boldsymbol{y})p(X|\boldsymbol{y})} \left[ \ln \left( \frac{p(\boldsymbol{y} \mid \boldsymbol{x}_1)}{\frac{1}{N} \sum_{i=1}^{N} p(\boldsymbol{y} \mid \boldsymbol{x}_i)} \right) \right] \; = \; \mathrm{I}_{\mathrm{InfoNCE}}(X_1 \; ; \; Y) \; . \tag{A14}$$

*The second bound Eq. (A14) is a special case of the first bound Eq. (A13).*

*Proof.* **Part (I)**: Lower bound with score function $f(\boldsymbol{x}, \boldsymbol{y})$.

For each set $\tilde{X} = \{\boldsymbol{x}_2, \ldots, \boldsymbol{x}_N\}$, we define as data-dependent (depending on $\tilde{X}$) score function $g(\boldsymbol{x}_1, \boldsymbol{y}, \tilde{X})$ that is based on the score function $f(\boldsymbol{x}, \boldsymbol{y})$. Therefore we have for each $\tilde{X}$ a different data-dependent score function $g$ based on $f$. We will derive a bound on the InfoNCE, which is the expectation of a lower bond on the mutual information over the score functions. For score function $g(\boldsymbol{x}_1, \boldsymbol{y}, \tilde{X})$, we define a variational distribution $q(\boldsymbol{x}_1 \mid \boldsymbol{y}, \tilde{X})$ over $\boldsymbol{x}_1$:

$$q(\boldsymbol{x}_1 \mid \boldsymbol{y}, \tilde{X}) \; = \; \frac{p(\boldsymbol{x}_1) \, g(\boldsymbol{x}_1, \boldsymbol{y}, \tilde{X})}{Z(\boldsymbol{y}, \tilde{X})} \; , \tag{A15}$$

$$Z(\boldsymbol{y}, \tilde{X}) \; = \; \mathrm{E}_{p(\boldsymbol{x}_1)} \left[ g(\boldsymbol{x}_1, \boldsymbol{y}, \tilde{X}) \right] \; , \tag{A16}$$

which ensures

$$\int q(\boldsymbol{x}_1 \mid \boldsymbol{y}, \tilde{X}) \, \mathrm{d}\boldsymbol{x}_1 \; = \; 1 \; . \tag{A17}$$

We have

$$\frac{q(\boldsymbol{x}_1 \mid \boldsymbol{y}, \tilde{X})}{p(\boldsymbol{x}_1)} \; = \; \frac{g(\boldsymbol{x}_1, \boldsymbol{y}, \tilde{X})}{Z(\boldsymbol{y}, \tilde{X})} \; . \tag{A18}$$

For the function $g$, we set

$$g(\boldsymbol{x}_1, \boldsymbol{y}, \tilde{X}) \; = \; \frac{f(\boldsymbol{x}_1, \boldsymbol{y})}{\frac{1}{N} \sum_{i=1}^{N} f(\boldsymbol{x}_i, \boldsymbol{y})} \; , \tag{A19}$$

For the function $f$ we use

$$f(\boldsymbol{x}_1, \boldsymbol{y}) \; = \; \exp(\tau^{-1} \, \mathrm{sim}(\boldsymbol{x}_1, \boldsymbol{y})) \; , \tag{A20}$$

where $\mathrm{sim}(\boldsymbol{x}, \boldsymbol{y})$ is typically the cosine similarity.

We next show that InfoNCE is a lower bound on the mutual information.

$$\mathrm{I}(X_1 \ ; \ Y) \ = \ \mathrm{E}_{p(\tilde{X})}\left[\mathrm{I}(X_1 \ ; \ Y)\right] \ = \ \mathrm{E}_{p(\tilde{X})}\left[\mathrm{E}_{p(\boldsymbol{x}_1, \boldsymbol{y})}\left[\ln \frac{p(\boldsymbol{x}_1 \mid \boldsymbol{y})}{p(\boldsymbol{x}_1)}\right]\right] \tag{A21}$$

$$= \ \mathrm{E}_{p(\tilde{X})}\left[\mathrm{E}_{p(\boldsymbol{x}_1, \boldsymbol{y})}\left[\ln \left(\frac{p(\boldsymbol{x}_1 \mid \boldsymbol{y})}{q(\boldsymbol{x}_1 \mid \boldsymbol{y}, \tilde{X})} \ \frac{q(\boldsymbol{x}_1 \mid \boldsymbol{y}, \tilde{X})}{p(\boldsymbol{x}_1)}\right)\right]\right]$$

$$= \ \mathrm{E}_{p(\tilde{X})}\left[\mathrm{E}_{p(\boldsymbol{x}_1, \boldsymbol{y})}\left[\ln \frac{q(\boldsymbol{x}_1 \mid \boldsymbol{y}, \tilde{X})}{p(\boldsymbol{x}_1)}\right] \ + \ \mathrm{E}_{p(\boldsymbol{y})}\left[\mathrm{KL}(p(\boldsymbol{x}_1 \mid \boldsymbol{y}) \parallel q(\boldsymbol{x}_1 \mid \boldsymbol{y}, \tilde{X}))\right]\right]$$

$$\geq \ \mathrm{E}_{p(\tilde{X})}\left[\mathrm{E}_{p(\boldsymbol{x}_1, \boldsymbol{y})}\left[\ln \frac{q(\boldsymbol{x}_1 \mid \boldsymbol{y}, \tilde{X})}{p(\boldsymbol{x}_1)}\right]\right] \ = \ \mathrm{E}_{p(\tilde{X})}\left[\mathrm{E}_{p(\boldsymbol{x}_1, \boldsymbol{y})}\left[\ln \frac{g(\boldsymbol{x}_1, \boldsymbol{y}, \tilde{X})}{Z(\boldsymbol{y}, \tilde{X})}\right]\right]$$

$$= \ \mathrm{E}_{p(\tilde{X})}\left[\mathrm{E}_{p(\boldsymbol{x}_1, \boldsymbol{y})}\left[\ln g(\boldsymbol{x}_1, \boldsymbol{y}, \tilde{X}) \ - \ \ln \left(\mathrm{E}_{p(\boldsymbol{x}_1)}\left[g(\boldsymbol{x}_1, \boldsymbol{y}, \tilde{X})\right]\right)\right]\right]$$

$$= \ \mathrm{E}_{p(\tilde{X})}\left[\mathrm{E}_{p(\boldsymbol{y})}\left[\mathrm{E}_{p(\boldsymbol{x}_1 \mid \boldsymbol{y})}\left[\ln g(\boldsymbol{x}_1, \boldsymbol{y}, \tilde{X})\right] \ - \ \ln \left(\mathrm{E}_{p(\boldsymbol{x}_1)}\left[g(\boldsymbol{x}_1, \boldsymbol{y}, \tilde{X})\right]\right)\right]\right]$$

$$= \ \mathrm{E}_{p(\tilde{X})}\left[\mathrm{E}_{p(\boldsymbol{y})}\left[\mathrm{E}_{p(\boldsymbol{x}_1 \mid \boldsymbol{y})}\left[\ln g(\boldsymbol{x}_1, \boldsymbol{y}, \tilde{X})\right]\right]\right] \ - \ \mathrm{E}_{p(\tilde{X})}\left[\mathrm{E}_{p(\boldsymbol{y})}\left[\ln \left(\mathrm{E}_{p(\boldsymbol{x}_1)}\left[g(\boldsymbol{x}_1, \boldsymbol{y}, \tilde{X})\right]\right)\right]\right]$$

$$\geq \ \mathrm{E}_{p(\boldsymbol{y})p(X \mid \boldsymbol{y})}\left[\ln g(\boldsymbol{x}_1, \boldsymbol{y}, \tilde{X})\right] \ - \ \mathrm{E}_{p(\tilde{X})}\left[\mathrm{E}_{p(\boldsymbol{y})}\left[\mathrm{E}_{p(\boldsymbol{x}_1)}\left[g(\boldsymbol{x}_1, \boldsymbol{y}, \tilde{X})\right] \ - \ 1\right]\right]$$

$$= \ \mathrm{E}_{p(\boldsymbol{y})p(X \mid \boldsymbol{y})}\left[\ln \frac{f(\boldsymbol{x}_1, \boldsymbol{y})}{\frac{1}{N} \sum_{i=1}^{N} f(\boldsymbol{x}_i, \boldsymbol{y})}\right] \ - \ \mathrm{E}_{p(\boldsymbol{y})}\left[\mathrm{E}_{p(X)}\left[\frac{f(\boldsymbol{x}_1, \boldsymbol{y})}{\frac{1}{N} \sum_{i=1}^{N} f(\boldsymbol{x}_i, \boldsymbol{y})}\right] \ - \ 1\right]$$

$$= \ \mathrm{E}_{p(\boldsymbol{y})p(X \mid \boldsymbol{y})}\left[\ln \frac{f(\boldsymbol{x}_1, \boldsymbol{y})}{\frac{1}{N} \sum_{i=1}^{N} f(\boldsymbol{x}_i, \boldsymbol{y})}\right] \ - \ \mathrm{E}_{p(\boldsymbol{y})}\left[\frac{1}{N} \sum_{i=1}^{N} \mathrm{E}_{p(X)}\left[\frac{f(\boldsymbol{x}_i, \boldsymbol{y})}{\frac{1}{N} \sum_{i=1}^{N} f(\boldsymbol{x}_i, \boldsymbol{y})}\right] \ - \ 1\right]$$

$$= \ \mathrm{E}_{p(\boldsymbol{y})p(X \mid \boldsymbol{y})}\left[\ln \frac{f(\boldsymbol{x}_1, \boldsymbol{y})}{\frac{1}{N} \sum_{i=1}^{N} f(\boldsymbol{x}_i, \boldsymbol{y})}\right] \ - \ \mathrm{E}_{p(\boldsymbol{y})}\left[\mathrm{E}_{p(X)}\left[\frac{\frac{1}{N} \sum_{i=1}^{N} f(\boldsymbol{x}_i, \boldsymbol{y})}{\frac{1}{N} \sum_{i=1}^{N} f(\boldsymbol{x}_i, \boldsymbol{y})}\right] \ - \ 1\right]$$

$$= \ \mathrm{E}_{p(\boldsymbol{y})p(X \mid \boldsymbol{y})}\left[\ln \frac{f(\boldsymbol{x}_1, \boldsymbol{y})}{\frac{1}{N} \sum_{i=1}^{N} f(\boldsymbol{x}_i, \boldsymbol{y})}\right]$$

$$= \ \mathrm{I}_{\mathrm{InfoNCE}}(X_1 \ ; \ Y) \, .$$

For the first "$\geq$" we used that the Kullback-Leibler divergence is non-negative. For the second "$\geq$" we used the inequality $\ln a \leqslant a - 1$ for $a > 0$.

**Part (II)**: Lower bound with probabilities.

If the score function $f$ is

$$f(\boldsymbol{x}, \boldsymbol{y}) \ = \ p(\boldsymbol{y} \mid \boldsymbol{x}) \, , \tag{A22}$$

then the bound is

$$\mathrm{I}(X_1 \ ; \ Y) \ \geq \ \mathrm{E}_{p(\boldsymbol{y})p(X \mid \boldsymbol{y})}\left[\ln \left(\frac{f(\boldsymbol{x}_1, \boldsymbol{y})}{\frac{1}{N} \sum_{i=1}^{N} f(\boldsymbol{x}_i, \boldsymbol{y})}\right)\right] \ = \ \mathrm{E}_{p(\boldsymbol{y})p(X \mid \boldsymbol{y})}\left[\ln \left(\frac{p(\boldsymbol{y} \mid \boldsymbol{x}_1)}{\frac{1}{N} \sum_{i=1}^{N} p(\boldsymbol{y} \mid \boldsymbol{x}_i)}\right)\right] \tag{A23}$$

$$= \ \mathrm{E}_{p(\boldsymbol{y})p(X \mid \boldsymbol{y})}\left[\ln \left(\frac{\frac{p(\boldsymbol{y} \mid \boldsymbol{x}_1)}{p(\boldsymbol{y})}}{\frac{1}{N} \sum_{i=1}^{N} \frac{p(\boldsymbol{y} \mid \boldsymbol{x}_i)}{p(\boldsymbol{y})}}\right)\right] \ = \ \mathrm{I}_{\mathrm{InfoNCE}}(X_1 \ ; \ Y) \, .$$

This is the bound with probabilities in the theorem. □

### A.1.2 InfoLOOB: Upper Bound on Mutual Information

We derive an upper bound on the mutual information between random variables $X$ and $Y$ distributed according to $p(\boldsymbol{x}, \boldsymbol{y})$. The mutual information $\mathrm{I}(X \ ; \ Y)$ between random variables $X$ and $Y$ is

$$\mathrm{I}(X \ ; \ Y) \ = \ \mathrm{E}_{p(\boldsymbol{x}, \boldsymbol{y})}\left[\ln \frac{p(\boldsymbol{x}, \boldsymbol{y})}{p(\boldsymbol{x}) \, p(\boldsymbol{y})}\right] \ = \ \mathrm{E}_{p(\boldsymbol{x}, \boldsymbol{y})}\left[\ln \frac{p(\boldsymbol{x} \mid \boldsymbol{y})}{p(\boldsymbol{x})}\right] \ = \ \mathrm{E}_{p(\boldsymbol{x}, \boldsymbol{y})}\left[\ln \frac{p(\boldsymbol{y} \mid \boldsymbol{x})}{p(\boldsymbol{y})}\right] \, . \tag{A24}$$

In Poole et al. (2019) Eq. (13) introduces a variational upper bound on the mutual information, which has been called "Leave one out upper bound" (called "L1Out" in Cheng et al. (2020)). For simplicity, we call this bound "InfoLOOB", where LOOB is an acronym for "Leave One Out Bound". In contrast to InfoNCE, InfoLOOB is an upper bound on the mutual information. InfoLOOB is analog to InfoNCE except that the negative samples do not contain a positive sample. Fig. 1 and Fig. 2 in Cheng et al. (2020) both show that InfoLOOB is a better estimator for the mutual information than InfoNCE (van den Oord et al., 2018), MINE (Belghazi et al., 2018), and NWJ (Nguyen et al., 2010).

The InfoLOOB with score function $f(\boldsymbol{x}, \boldsymbol{y})$ is defined as

$$\mathrm{I}_{\mathrm{InfoLOOB}}(X_1 \; ; \; Y) \;=\; \mathrm{E}_{p(\boldsymbol{y})}\left[\mathrm{E}_{p(X|\boldsymbol{y})}\left[\ln\left(\frac{f(\boldsymbol{x}_1, \boldsymbol{y})}{\frac{1}{N-1}\sum_{i=2}^{N} f(\boldsymbol{x}_i, \boldsymbol{y})}\right)\right]\right] . \tag{A25}$$

The InfoLOOB with probabilities is defined as

$$\mathrm{I}_{\mathrm{InfoLOOB}}(X_1 \; ; \; Y) \;=\; \mathrm{E}_{p(\boldsymbol{y})}\left[\mathrm{E}_{p(X|\boldsymbol{y})}\left[\ln\left(\frac{p(\boldsymbol{y} \mid \boldsymbol{x}_1)}{\frac{1}{N-1}\sum_{i=2}^{N} p(\boldsymbol{y} \mid \boldsymbol{x}_i)}\right)\right]\right] . \tag{A26}$$

This is the InfoLOOB Eq. (A25) with $f(\boldsymbol{x}, \boldsymbol{y}) = p(\boldsymbol{y} \mid \boldsymbol{x})$.

The InfoLOOB with probabilities can be written in different forms:

$$\mathrm{I}_{\mathrm{InfoLOOB}}(X_1 \; ; \; Y) \;=\; \mathrm{E}_{p(\boldsymbol{y})}\left[\mathrm{E}_{p(X|\boldsymbol{y})}\left[\ln\left(\frac{p(\boldsymbol{y} \mid \boldsymbol{x}_1)}{\frac{1}{N-1}\sum_{i=2}^{N} p(\boldsymbol{y} \mid \boldsymbol{x}_i)}\right)\right]\right] \tag{A27}$$

$$=\; \mathrm{E}_{p(\boldsymbol{y})}\left[\mathrm{E}_{p(X|\boldsymbol{y})}\left[\ln\left(\frac{\frac{p(\boldsymbol{y}|\boldsymbol{x}_1)}{p(\boldsymbol{y})}}{\frac{1}{N-1}\sum_{i=2}^{N} \frac{p(\boldsymbol{y}|\boldsymbol{x}_i)}{p(\boldsymbol{y})}}\right)\right]\right] \;=\; \mathrm{E}_{p(\boldsymbol{y})}\left[\mathrm{E}_{p(X|\boldsymbol{y})}\left[\ln\left(\frac{\frac{p(\boldsymbol{x}_1|\boldsymbol{y})}{p(\boldsymbol{x}_1)}}{\frac{1}{N-1}\sum_{i=2}^{N} \frac{p(\boldsymbol{x}_i|\boldsymbol{y})}{p(\boldsymbol{x}_i)}}\right)\right]\right] .$$

**Set of pairs.** The InfoLOOB can we written in a different setting (Poole et al., 2019), which will be used in our implementations. We sample $N$ pairs independently from $p(\boldsymbol{x}, \boldsymbol{y})$, which gives $X = \{(\boldsymbol{x}_1, \boldsymbol{y}_1), (\boldsymbol{x}_2, \boldsymbol{y}_2), \ldots, (\boldsymbol{x}_N, \boldsymbol{y}_N)\}$. The InfoLOOB is then

$$\mathrm{I}_{\mathrm{InfoLOOB}}(X \; ; \; Y) \;=\; \mathrm{E}_{p(X|\boldsymbol{y})}\left[\frac{1}{N}\sum_{i=1}^{N}\ln\left(\frac{f(\boldsymbol{x}_i, \boldsymbol{y}_i)}{\frac{1}{N-1}\sum_{j=1,j\neq i}^{N} f(\boldsymbol{x}_j, \boldsymbol{y}_i)}\right)\right] . \tag{A28}$$

We assume that an anchor sample $\boldsymbol{y}$ is given. For the anchor sample $\boldsymbol{y}$ we draw a positive sample $\boldsymbol{x}_1$ according to $p(\boldsymbol{x}_1 \mid \boldsymbol{y})$. Next, we draw a set $\tilde{X} = \{\boldsymbol{x}_2, \ldots, \boldsymbol{x}_N\}$ of negative samples according to $\tilde{p}(\boldsymbol{x} \mid \boldsymbol{y})$. **For a given $\boldsymbol{y}$, the $\boldsymbol{x}$ that have a large $p(\boldsymbol{x} \mid \boldsymbol{y})$ are drawn with a lower probability $\tilde{p}(\boldsymbol{x} \mid \boldsymbol{y})$ compared to random drawing via $p(\boldsymbol{x})$.** The negatives are indeed negatives. We have drawn first anchor sample $\boldsymbol{y}$ and then $X = \{\boldsymbol{x}_1, \ldots, \boldsymbol{x}_N\}$, where $\boldsymbol{x}_1$ is drawn according to $p(\boldsymbol{x}_1 \mid \boldsymbol{y})$ and $\tilde{X} = \{\boldsymbol{x}_2, \ldots, \boldsymbol{x}_N\}$ are drawn iid according to $\tilde{p}(\boldsymbol{x} \mid \boldsymbol{y})$. We have

$$\tilde{p}(\tilde{X} \mid \boldsymbol{y}) \;=\; \prod_{i=2}^{N} \tilde{p}(\boldsymbol{x}_i \mid \boldsymbol{y}) , \tag{A29}$$

$$\tilde{p}(X \mid \boldsymbol{y}) \;=\; p(\boldsymbol{x}_1 \mid \boldsymbol{y}) \prod_{i=2}^{N} \tilde{p}(\boldsymbol{x}_i \mid \boldsymbol{y}) , \tag{A30}$$

$$\tilde{p}(\tilde{X} \mid \boldsymbol{y}) \, p(\boldsymbol{x}_1) \;=\; p(\boldsymbol{x}_1) \prod_{i=2}^{N} \tilde{p}(\boldsymbol{x}_i \mid \boldsymbol{y}) . \tag{A31}$$

We assume for score function $f(\boldsymbol{x}, \boldsymbol{y})$

$$\forall_{\boldsymbol{y}}\forall_{\boldsymbol{x}} : \; 0 \;<\; f(\boldsymbol{x}, \boldsymbol{y}) . \tag{A32}$$

We ensure this by using for score function $f$

$$f(\boldsymbol{x}, \boldsymbol{y}) \;=\; \exp(\tau^{-1}\,\mathrm{sim}(\boldsymbol{x}, \boldsymbol{y})) , \tag{A33}$$

where $\mathrm{sim}(\boldsymbol{x}, \boldsymbol{y})$ is typically the cosine similarity.

InfoLOOB with score function $f(\boldsymbol{x}, \boldsymbol{y})$ and with undersampling via $\tilde{p}(X \mid \boldsymbol{y})$ is (compare the definition of InfoLOOB Eq. (A25) without undersampling):

$$\mathrm{I}_{\mathrm{InfoLOOB}}(X \; ; \; Y) \; = \; \mathrm{E}_{p(\boldsymbol{y})}\left[\mathrm{E}_{\tilde{p}(X|\boldsymbol{y})}\left[\ln\left(\frac{f(\boldsymbol{x}_1, \boldsymbol{y})}{\frac{1}{N-1}\sum_{i=2}^{N} f(\boldsymbol{x}_i, \boldsymbol{y})}\right)\right]\right] \; . \tag{A34}$$

The reference constant $Z(\boldsymbol{y})$ gives the average score $f(\boldsymbol{x}, \boldsymbol{y})$, if the negatives for $\boldsymbol{y}$ are selected with lower probability via $\tilde{p}(\boldsymbol{x} \mid \boldsymbol{y})$ than with random drawing according to $p(\boldsymbol{x})$.

$$Z(\boldsymbol{y}) \; = \; \mathrm{E}_{\tilde{p}(\boldsymbol{x}|\boldsymbol{y})}\left[f(\boldsymbol{x}, \boldsymbol{y})\right] \; . \tag{A35}$$

We define the variational distribution

$$q(\boldsymbol{x} \mid \boldsymbol{y}) \; = \; \frac{p(\boldsymbol{x})\, f(\boldsymbol{x}, \boldsymbol{y})}{Z^*(\boldsymbol{y})} \; , \quad Z^*(\boldsymbol{y}) \; = \; \mathrm{E}_{p(\boldsymbol{x})}\left[f(\boldsymbol{x}, \boldsymbol{y})\right] \; . \tag{A36}$$

With the variational distribution $q(\boldsymbol{x} \mid \boldsymbol{y})$, we express our main assumption. **The main assumption for the bound is:**

$$\mathrm{E}_{p(\boldsymbol{y})}\left[\mathrm{KL}(p(\boldsymbol{x} \mid \boldsymbol{y}) \parallel q(\boldsymbol{x} \mid \boldsymbol{y}))\right] \; \leqslant \; \mathrm{E}_{p(\boldsymbol{y})}\left[\ln Z^*(\boldsymbol{y}) \; - \; \ln Z(\boldsymbol{y})\right] \; . \tag{A37}$$

This assumption can be written as

$$\mathrm{E}_{p(\boldsymbol{y})}\left[\mathrm{E}_{p(\boldsymbol{x}|\boldsymbol{y})}\left[\ln\left(\frac{p(\boldsymbol{y} \mid \boldsymbol{x})\, Z(\boldsymbol{y})}{p(\boldsymbol{y})\, f(\boldsymbol{x}, \boldsymbol{y})}\right)\right]\right] \; \leqslant \; 0 \; . \tag{A38}$$

This assumption ensures that the $\boldsymbol{x}$ with large $p(\boldsymbol{x} \mid \boldsymbol{y})$) are selected with lower probability via $\tilde{p}(\boldsymbol{x} \mid \boldsymbol{y})$ than with random drawing according to $p(\boldsymbol{x})$. The negatives are ensured to be real negatives, that is, $p(\boldsymbol{x} \mid \boldsymbol{y})$ is small and so is $f(\boldsymbol{x}, \boldsymbol{y})$. Consequently, we make sure that we draw $\boldsymbol{x}$ with sufficient small $f(\boldsymbol{x}, \boldsymbol{y})$. The Kullback-Leibler gives the minimal required gap between drawing $f(\boldsymbol{x}, \boldsymbol{y})$ via $p(\boldsymbol{x})$ and drawing $f(\boldsymbol{x}, \boldsymbol{y})$ via $\tilde{p}(\boldsymbol{x} \mid \boldsymbol{y})$.

**EXAMPLE.** With $h(\boldsymbol{y}) > 0$, we consider the setting

$$f(\boldsymbol{x}, \boldsymbol{y}) \; = \; \frac{p(\boldsymbol{y} \mid \boldsymbol{x})\, h(\boldsymbol{y})}{p(\boldsymbol{y})} \; , \tag{A39}$$

$$\tilde{p}(\boldsymbol{x} \mid \boldsymbol{y}) \; = \; \frac{p(\boldsymbol{x})\, p(\boldsymbol{y})}{h(\boldsymbol{y})\, p(\boldsymbol{y} \mid \boldsymbol{x})\, C(\boldsymbol{y})} \; , \quad C(\boldsymbol{y}) \; = \; \mathrm{E}_{p(\boldsymbol{x})}\left[\left(\frac{p(\boldsymbol{y} \mid \boldsymbol{x})\, h(\boldsymbol{y})}{p(\boldsymbol{y})}\right)^{-1}\right] \; . \tag{A40}$$

The main assumption becomes

$$\mathrm{E}_{p(\boldsymbol{y})}\left[\mathrm{E}_{p(\boldsymbol{x}|\boldsymbol{y})}\left[\ln \frac{Z(\boldsymbol{y})}{h(\boldsymbol{y})}\right]\right] \; \leqslant \; 0 \; . \tag{A41}$$

The main assumption holds since

$$Z(\boldsymbol{y}) \; = \; \mathrm{E}_{\tilde{p}(\boldsymbol{x}|\boldsymbol{y})}\left[\frac{p(\boldsymbol{y} \mid \boldsymbol{x})\, h(\boldsymbol{y})}{p(\boldsymbol{y})}\right] \; = \; \int \frac{p(\boldsymbol{x})\, p(\boldsymbol{y})}{h(\boldsymbol{y})\, p(\boldsymbol{y} \mid \boldsymbol{x})\, C(\boldsymbol{y})} \, \frac{p(\boldsymbol{y} \mid \boldsymbol{x})\, h(\boldsymbol{y})}{p(\boldsymbol{y})} \, \mathrm{d}\boldsymbol{x} \tag{A42}$$

$$= \; \int p(\boldsymbol{x})\, C(\boldsymbol{y})^{-1} \, \mathrm{d}\boldsymbol{x} \; = \; C(\boldsymbol{y})^{-1} \; = \; \left(\mathrm{E}_{p(\boldsymbol{x})}\left[\left(\frac{p(\boldsymbol{y} \mid \boldsymbol{x})\, h(\boldsymbol{y})}{p(\boldsymbol{y})}\right)^{-1}\right]\right)^{-1}$$

$$\leqslant \; \left(\mathrm{E}_{p(\boldsymbol{x})}\left[\frac{p(\boldsymbol{y} \mid \boldsymbol{x})\, h(\boldsymbol{y})}{p(\boldsymbol{y})}\right]^{-1}\right)^{-1} \; = \; \mathrm{E}_{p(\boldsymbol{x})}\left[\frac{p(\boldsymbol{y} \mid \boldsymbol{x})\, h(\boldsymbol{y})}{p(\boldsymbol{y})}\right]$$

$$= \; \int \frac{p(\boldsymbol{y}, \boldsymbol{x})\, h(\boldsymbol{y})}{p(\boldsymbol{y})} \, \mathrm{d}\boldsymbol{x} \; = \; h(\boldsymbol{y}) \; ,$$

where we used for the $\leqslant$ Jensen's inequality with the function $f(a) = 1/a$, which is convex for $a > 0$.

For score function $f(\boldsymbol{x}, \boldsymbol{y})$ and distribution $\tilde{p}(\boldsymbol{x} \mid \boldsymbol{y})$ for sampling the negative samples, we have defined:

$$Z(\boldsymbol{y}) = \mathrm{E}_{\tilde{p}(\boldsymbol{x}|\boldsymbol{y})}[f(\boldsymbol{x}, \boldsymbol{y})] \ , \tag{A43}$$

$$Z^*(\boldsymbol{y}) = \mathrm{E}_{p(\boldsymbol{x})}[f(\boldsymbol{x}, \boldsymbol{y})] \ , \tag{A44}$$

$$q(\boldsymbol{x} \mid \boldsymbol{y}) = \frac{p(\boldsymbol{x}) \, f(\boldsymbol{x}, \boldsymbol{y})}{Z^*(\boldsymbol{y})} \ . \tag{A45}$$

Next theorem gives the upper bound of the InfoLOOB on the mutual information, which is

$$\mathrm{I}(X_1 \ ; \ Y) = \mathrm{E}_{p(\boldsymbol{x}_1, \boldsymbol{y})}\left[\ln \frac{p(\boldsymbol{x}_1 \mid \boldsymbol{y})}{p(\boldsymbol{x}_1)}\right] \ . \tag{A46}$$

**Theorem A2** (InfoLOOB upper bound). *If $\tilde{X} = \{\boldsymbol{x}_2, \ldots, \boldsymbol{x}_N\}$ are drawn iid according to $\tilde{p}(\boldsymbol{x} \mid \boldsymbol{y})$ and if the main assumption holds:*

$$\mathrm{E}_{p(\boldsymbol{y})}[\mathrm{KL}(p(\boldsymbol{x} \mid \boldsymbol{y}) \ \| \ q(\boldsymbol{x} \mid \boldsymbol{y}))] \ \leqslant \ \mathrm{E}_{p(\boldsymbol{y})}[\ln Z^*(\boldsymbol{y}) \ - \ \ln Z(\boldsymbol{y})] \ . \tag{A47}$$

*Then InfoLOOB with score function $f(\boldsymbol{x}, \boldsymbol{y})$ and undersampling positives by $\tilde{p}(X \mid \boldsymbol{y})$ is an upper bound on the mutual information:*

$$\mathrm{I}(X_1 \ ; \ Y) \ \leqslant \ \mathrm{E}_{p(\boldsymbol{y})}\left[\mathrm{E}_{\tilde{p}(X|\boldsymbol{y})}\left[\ln\left(\frac{f(\boldsymbol{x}_1, \boldsymbol{y})}{\frac{1}{N-1}\sum_{i=2}^{N} f(\boldsymbol{x}_i, \boldsymbol{y})}\right)\right]\right] \ = \ \mathrm{I}_{\mathrm{InfoLOOB}}(X_1 \ ; \ Y) \ . \tag{A48}$$

*If the negative samples $\tilde{X} = \{\boldsymbol{x}_2, \ldots, \boldsymbol{x}_N\}$ are drawn iid according to $p(\boldsymbol{x})$, then InfoLOOB with probabilities according to Eq. (A26) is an upper bound on the mutual information:*

$$\mathrm{I}(X_1 \ ; \ Y) \ \leqslant \ \mathrm{E}_{p(\boldsymbol{y})}\left[\mathrm{E}_{p(X|\boldsymbol{y})}\left[\ln\left(\frac{p(\boldsymbol{y} \mid \boldsymbol{x}_1)}{\frac{1}{N-1}\sum_{i=2}^{N} p(\boldsymbol{y} \mid \boldsymbol{X}_i)}\right)\right]\right] \ = \ \mathrm{I}_{\mathrm{InfoLOOB}}(X_1 \ ; \ Y) \ . \tag{A49}$$

*The second bound Eq. (A49) is a special case of the first bound Eq. (A48).*

*Proof.* **Part (I)**: Upper bound with score function $f(\boldsymbol{x}, \boldsymbol{y})$.

$$I(X_1 ; Y) = E_{p(\boldsymbol{x}_1, \boldsymbol{y})} \left[ \ln \frac{p(\boldsymbol{x}_1 \mid \boldsymbol{y})}{p(\boldsymbol{x}_1)} \right] \tag{A50}$$

$$= E_{p(\boldsymbol{x}_1, \boldsymbol{y})} \left[ \ln \left( \frac{p(\boldsymbol{x}_1 \mid \boldsymbol{y})}{q(\boldsymbol{x}_1 \mid \boldsymbol{y})} \frac{q(\boldsymbol{x}_1 \mid \boldsymbol{y})}{p(\boldsymbol{x}_1)} \right) \right]$$

$$= E_{p(\boldsymbol{x}_1, \boldsymbol{y})} \left[ \ln \frac{q(\boldsymbol{x}_1 \mid \boldsymbol{y})}{p(\boldsymbol{x}_1)} \right] + E_{p(\boldsymbol{y})} \left[ KL(p(\boldsymbol{x}_1 \mid \boldsymbol{y}) \parallel q(\boldsymbol{x}_1 \mid \boldsymbol{y})) \right]$$

$$\leqslant E_{p(\boldsymbol{x}_1, \boldsymbol{y})} \left[ \ln \frac{q(\boldsymbol{x}_1 \mid \boldsymbol{y})}{p(\boldsymbol{x}_1)} \right] + E_{p(\boldsymbol{y})} \left[ \ln E_{p(\boldsymbol{x}_1)} \left[ f(\boldsymbol{x}_1, \boldsymbol{y}) \right] - \ln Z(\boldsymbol{y}) \right]$$

$$= E_{p(\boldsymbol{x}_1, \boldsymbol{y})} \left[ \ln \frac{q(\boldsymbol{x}_1 \mid \boldsymbol{y})}{p(\boldsymbol{x}_1)} + \ln \frac{E_{p(\boldsymbol{x}_1)} \left[ f(\boldsymbol{x}_1, \boldsymbol{y}) \right]}{Z(\boldsymbol{y})} \right]$$

$$= E_{p(\boldsymbol{x}_1, \boldsymbol{y})} \left[ \ln \left( \frac{f(\boldsymbol{x}_1, \boldsymbol{y})}{E_{p(\boldsymbol{x}_1)} \left[ f(\boldsymbol{x}_1, \boldsymbol{y}) \right]} \frac{E_{p(\boldsymbol{x}_1)} \left[ f(\boldsymbol{x}_1, \boldsymbol{y}) \right]}{Z(\boldsymbol{y})} \right) \right]$$

$$= E_{p(\boldsymbol{x}_1, \boldsymbol{y})} \left[ \ln \frac{f(\boldsymbol{x}_1, \boldsymbol{y})}{Z(\boldsymbol{y})} \right]$$

$$= E_{p(\boldsymbol{x}_1, \boldsymbol{y})} \left[ \ln \left( \frac{f(\boldsymbol{x}_1, \boldsymbol{y})}{E_{\tilde{p}(X \mid \boldsymbol{y})} \left[ \frac{1}{N-1} \sum_{i=2}^{N} f(\boldsymbol{x}_i, \boldsymbol{y}) \right]} \right) \right]$$

$$= E_{p(\boldsymbol{x}_1, \boldsymbol{y})} \left[ \ln f(\boldsymbol{x}_1, \boldsymbol{y}) \right] - E_{p(\boldsymbol{y})} \left[ \ln \left( E_{\tilde{p}(X \mid \boldsymbol{y})} \left[ \frac{1}{N-1} \sum_{i=2}^{N} f(\boldsymbol{x}_i, \boldsymbol{y}) \right] \right) \right]$$

$$\leqslant E_{p(\boldsymbol{x}_1, \boldsymbol{y})} \left[ \ln f(\boldsymbol{x}_1, \boldsymbol{y}) \right] - E_{p(\boldsymbol{y})} \left[ E_{\tilde{p}(X \mid \boldsymbol{y})} \left[ \ln \left( \frac{1}{N-1} \sum_{i=2}^{N} f(\boldsymbol{x}_i, \boldsymbol{y}) \right) \right] \right]$$

$$= E_{p(\boldsymbol{y})} \left[ E_{\tilde{p}(X \mid \boldsymbol{y})} \left[ \ln \left( \frac{f(\boldsymbol{x}_1, \boldsymbol{y})}{\frac{1}{N-1} \sum_{i=2}^{N} f(\boldsymbol{x}_i, \boldsymbol{y})} \right) \right] \right]$$

$$= I_{\text{InfoLOOB}}(X_1 ; Y) \,,$$

where the first "$\leqslant$" uses assumption Eq. (A37), while Jensens's inequality was used for the second "$\leqslant$" by exchanging the expectation and the "ln". We also used

$$E_{\tilde{p}(X \mid \boldsymbol{y})} \left[ \frac{1}{N-1} \sum_{i=2}^{N} f(\boldsymbol{x}_i, \boldsymbol{y}) \right] = \frac{1}{N-1} \sum_{i=2}^{N} E_{\tilde{p}(\boldsymbol{x}_i \mid \boldsymbol{y})} \left[ f(\boldsymbol{x}_i, \boldsymbol{y}) \right] = \frac{1}{N-1} \sum_{i=2}^{N} Z(\boldsymbol{y}) = Z(\boldsymbol{y}) \,. \tag{A51}$$

**Part (II)**: Upper bound with probabilities.

If the score function $f$ is

$$f(\boldsymbol{x}, \boldsymbol{y}) = p(\boldsymbol{y} \mid \boldsymbol{x}) \tag{A52}$$

and

$$\tilde{p}(\boldsymbol{x} \mid \boldsymbol{y}) = p(\boldsymbol{x}) \,, \tag{A53}$$

then

$$\tilde{p}(X \mid \boldsymbol{y}) = p(X \mid \boldsymbol{y}) \,, \tag{A54}$$

$$Z(\boldsymbol{y}) = E_{p(\boldsymbol{x})} \left[ p(\boldsymbol{y} \mid \boldsymbol{x}) \right] = p(\boldsymbol{y}) \,, \tag{A55}$$

$$Z^*(\boldsymbol{y}) = E_{p(\boldsymbol{x})} \left[ p(\boldsymbol{y} \mid \boldsymbol{x}) \right] = p(\boldsymbol{y}) \,, \tag{A56}$$

$$q(\boldsymbol{x} \mid \boldsymbol{y}) = \frac{p(\boldsymbol{x}) \, p(\boldsymbol{y} \mid \boldsymbol{x})}{p(\boldsymbol{y})} = p(\boldsymbol{x} \mid \boldsymbol{y}) \,, \tag{A57}$$

$$KL(p(\boldsymbol{x} \mid \boldsymbol{y}) \parallel q(\boldsymbol{x} \mid \boldsymbol{y})) = KL(p(\boldsymbol{x} \mid \boldsymbol{y}) \parallel p(\boldsymbol{x} \mid \boldsymbol{y})) = 0 \,. \tag{A58}$$

Therefore, the main assumption holds, since

$$0 \;=\; \mathrm{E}_{p(\boldsymbol{y})}\left[\mathrm{KL}(p(\boldsymbol{x} \mid \boldsymbol{y}) \parallel q(\boldsymbol{x} \mid \boldsymbol{y}))\right] \;=\; \mathrm{E}_{p(\boldsymbol{y})}\left[\ln Z^*(\boldsymbol{y}) \;-\; \ln Z(\boldsymbol{y})\right] . \tag{A59}$$

The bound becomes

$$\mathrm{I}(X_1 \,;\, Y) \;\leqslant\; \mathrm{E}_{p(\boldsymbol{y})}\left[\mathrm{E}_{p(X\mid\boldsymbol{y})}\left[\ln\left(\frac{p(\boldsymbol{y} \mid \boldsymbol{x}_1)}{\frac{1}{N-1}\sum_{i=2}^{N} p(\boldsymbol{y} \mid \boldsymbol{x}_i)}\right)\right]\right] \tag{A60}$$

$$= \; \mathrm{E}_{p(\boldsymbol{y})}\left[\mathrm{E}_{p(X\mid\boldsymbol{y})}\left[\ln\left(\frac{\frac{p(\boldsymbol{y}\mid\boldsymbol{x}_1)}{p(\boldsymbol{y})}}{\frac{1}{N-1}\sum_{i=2}^{N}\frac{p(\boldsymbol{y}\mid\boldsymbol{x}_i)}{p(\boldsymbol{y})}}\right)\right]\right] \; = \; \mathrm{I}_{\mathrm{InfoLOOB}}(X_1 \,;\, Y) .$$

An alternative proof is as follows:

$$\mathrm{I}(X_1 \,;\, Y) \;=\; \mathrm{I}(X_1 \,;\, Y) \;-\; \mathrm{E}_{p(\boldsymbol{y})}\left[\ln\left(\frac{1}{N-1}\sum_{i=2}^{N}\frac{p(\boldsymbol{y})}{p(\boldsymbol{y})}\right)\right] \tag{A61}$$

$$= \; \mathrm{I}(X_1 \,;\, Y) \;-\; \mathrm{E}_{p(\boldsymbol{y})}\left[\ln\left(\mathrm{E}_{p(X\mid\boldsymbol{y})}\left[\frac{1}{N-1}\sum_{i=2}^{N}\frac{p(\boldsymbol{y} \mid \boldsymbol{x}_i)}{p(\boldsymbol{y})}\right]\right)\right]$$

$$\leqslant \; \mathrm{I}(X_1 \,;\, Y) \;-\; \mathrm{E}_{p(\boldsymbol{y})}\left[\mathrm{E}_{p(X\mid\boldsymbol{y})}\left[\ln\left(\frac{1}{N-1}\sum_{i=2}^{N}\frac{p(\boldsymbol{y} \mid \boldsymbol{x}_i)}{p(\boldsymbol{y})}\right)\right]\right]$$

$$= \; \mathrm{E}_{p(\boldsymbol{y})}\left[\mathrm{E}_{p(\boldsymbol{x}_1\mid\boldsymbol{y})}\left[\ln\left(\frac{p(\boldsymbol{x}_1 \mid \boldsymbol{y})}{p(\boldsymbol{x}_1)}\right)\right]\right] \;-\; \mathrm{E}_{p(\boldsymbol{y})}\left[\mathrm{E}_{p(X\mid\boldsymbol{y})}\left[\ln\left(\frac{1}{N-1}\sum_{i=2}^{N}\frac{p(\boldsymbol{x}_i \mid \boldsymbol{y})}{p(\boldsymbol{x}_i)}\right)\right]\right]$$

$$= \; \mathrm{E}_{p(\boldsymbol{y})}\left[\mathrm{E}_{p(X\mid\boldsymbol{y})}\left[\ln\left(\frac{\frac{p(\boldsymbol{x}_1\mid\boldsymbol{y})}{p(\boldsymbol{x}_1)}}{\frac{1}{N-1}\sum_{i=2}^{N}\frac{p(\boldsymbol{x}_i\mid\boldsymbol{y})}{p(\boldsymbol{x}_i)}}\right)\right]\right]$$

$$= \; \mathrm{I}_{\mathrm{InfoLOOB}}(X_1 \,;\, Y) .$$

where we applied Jensens's inequality for the exchanging the expectation and the "ln" to obtain the "$\leqslant$" inequality.

$\square$

Experiments that compare upper and lower bounds as mutual information estimates are provided in Cheng et al. (2020) and in Poole et al. (2019). In Fig. 2 in Cheng et al. (2020) it is shown that InfoLOOB is a good estimator of the mutual information.

### A.1.3 InfoLOOB: Analysis of the Objective

This subsection justifies the maximization of the InfoLOOB bound for contrastive learning. Maximizing the InfoLOOB bound is not intuitive as it was introduced as an upper bound on the mutual information in the previous subsection. Still maximizing the InfoLOOB bound leads to a good approximation of the mutual information, in particular for high mutual information.

InfoLOOB with a neural network as a scoring function is not an upper bound on the mutual information when not under-sampling. As we use InfoLOOB on training data for which we do not know the sampling procedure, we cannot assume under-sampling. Therefore, we elaborate more on the rationale behind the maximization of the InfoLOOB bound. (I) We show that InfoLOOB with neural networks as scoring function is bounded from above. Therefore, there exists a maximum and the optimization problem is well defined. (II) We show that InfoLOOB with neural networks as scoring function differs by two terms from the mutual information. The first term is the Kullback-Leibler divergence between the variational $q(\boldsymbol{x} \mid \boldsymbol{y})$ and the posterior $p(\boldsymbol{x} \mid \boldsymbol{y})$. This divergence is minimal for $q(\boldsymbol{x} \mid \boldsymbol{y}) = p(\boldsymbol{x} \mid \boldsymbol{y})$, which implies $f(\boldsymbol{y} \mid \boldsymbol{x}) = p(\boldsymbol{y} \mid \boldsymbol{x})$. The second term is governed by the difference between the mean $\mathrm{E}[f(\boldsymbol{x}, \boldsymbol{y})]$ and the empirical mean $1/(N-1)\sum_i f(\boldsymbol{x}_i, \boldsymbol{y})$. The Hoeffding bound can be applied to this difference. Therefore, the second term is negligible for large $N$. In contrast, the KL term is dominant and the relevant term, therefore maximizing InfoLOOB leads to $f(\boldsymbol{y} \mid \boldsymbol{x}) \approx p(\boldsymbol{y} \mid \boldsymbol{x})$.

We assume that an anchor sample $\boldsymbol{y}$ is given. For the anchor sample $\boldsymbol{y}$, we draw a positive sample $\boldsymbol{x}_1$ according to $p(\boldsymbol{x}_1 \mid \boldsymbol{y})$. We define the set $\tilde{X} = \{\boldsymbol{x}_2, \ldots, \boldsymbol{x}_N\}$ of negative samples, which are drawn iid according to $p(\boldsymbol{x})$. We define the set $X = \{\boldsymbol{x}_1, \ldots, \boldsymbol{x}_N\}$.

We have

$$p(\tilde{X}) \;=\; \prod_{i=2}^{N} p(\boldsymbol{x}_i) \,, \tag{A62}$$

$$p(X \mid \boldsymbol{y}) \;=\; p(\boldsymbol{x}_1 \mid \boldsymbol{y}) \prod_{i=2}^{N} p(\boldsymbol{x}_i) \;=\; p(\boldsymbol{x}_1 \mid \boldsymbol{y}) \, p(\tilde{X}) \,, \tag{A63}$$

$$p(X) \;=\; \prod_{i=1}^{N} p(\boldsymbol{x}_i) \;=\; p(\boldsymbol{x}_1) \, p(\tilde{X}) \,. \tag{A64}$$

We use the score function

$$f(\boldsymbol{x}, \boldsymbol{y}) \;=\; \exp(\tau^{-1} \, \mathrm{sim}(\boldsymbol{x}, \boldsymbol{y})) \,, \tag{A65}$$

where $\mathrm{sim}(\boldsymbol{x}, \boldsymbol{y})$ is typically the cosine similarity.

The InfoLOOB with score function $f(\boldsymbol{x}, \boldsymbol{y})$ is defined as

$$\mathrm{I}_{\mathrm{InfoLOOB}}(X_1 \,;\, Y) \;=\; \mathrm{E}_{p(\boldsymbol{y})} \left[ \mathrm{E}_{p(X \mid \boldsymbol{y})} \left[ \ln \left( \frac{f(\boldsymbol{x}_1, \boldsymbol{y})}{\frac{1}{N-1} \sum_{i=2}^{N} f(\boldsymbol{x}_i, \boldsymbol{y})} \right) \right] \right] \,. \tag{A66}$$

We define the variational distribution

$$q(\boldsymbol{x} \mid \boldsymbol{y}) \;=\; \frac{p(\boldsymbol{x}) \, f(\boldsymbol{x}, \boldsymbol{y})}{Z(\boldsymbol{y})} \,, \tag{A67}$$

$$Z(\boldsymbol{y}) \;=\; \mathrm{E}_{p(\boldsymbol{x})} \left[ f(\boldsymbol{x}, \boldsymbol{y}) \right] \,. \tag{A68}$$

The next inequality shows the relation between $I(X_1 ; Y)$ and $I_{\text{InfoLOOB}}(X_1 ; Y)$ for random variables $X_1$ and $Y$.

$$I(X_1 ; Y) = E_{p(\boldsymbol{x}_1, \boldsymbol{y})} \left[ \ln \frac{p(\boldsymbol{x}_1 \mid \boldsymbol{y})}{p(\boldsymbol{x}_1)} \right] \tag{A69}$$

$$= E_{p(\boldsymbol{x}_1, \boldsymbol{y})} \left[ \ln \left( \frac{p(\boldsymbol{x}_1 \mid \boldsymbol{y})}{q(\boldsymbol{x}_1 \mid \boldsymbol{y})} \frac{q(\boldsymbol{x}_1 \mid \boldsymbol{y})}{p(\boldsymbol{x}_1)} \right) \right]$$

$$= E_{p(\boldsymbol{x}_1, \boldsymbol{y})} \left[ \ln \frac{q(\boldsymbol{x}_1 \mid \boldsymbol{y})}{p(\boldsymbol{x}_1)} \right] + E_{p(\boldsymbol{y})} \left[ \text{KL}(p(\boldsymbol{x}_1 \mid \boldsymbol{y}) \, \| \, q(\boldsymbol{x}_1 \mid \boldsymbol{y})) \right]$$

$$= E_{p(\boldsymbol{x}_1, \boldsymbol{y})} \left[ \ln \frac{f(\boldsymbol{x}_1, \boldsymbol{y})}{Z(\boldsymbol{y})} \right] + E_{p(\boldsymbol{y})} \left[ \text{KL}(p(\boldsymbol{x}_1 \mid \boldsymbol{y}) \, \| \, q(\boldsymbol{x}_1 \mid \boldsymbol{y})) \right]$$

$$= E_{p(\boldsymbol{x}_1, \boldsymbol{y})} \left[ \ln \left( \frac{f(\boldsymbol{x}_1, \boldsymbol{y})}{E_{p(X \mid \boldsymbol{y})} \left[ \frac{1}{N-1} \sum_{i=2}^{N} f(\boldsymbol{x}_i, \boldsymbol{y}) \right]} \right) \right] + E_{p(\boldsymbol{y})} \left[ \text{KL}(p(\boldsymbol{x}_1 \mid \boldsymbol{y}) \, \| \, q(\boldsymbol{x}_1 \mid \boldsymbol{y})) \right]$$

$$= E_{p(\boldsymbol{x}_1, \boldsymbol{y})} \left[ \ln f(\boldsymbol{x}_1, \boldsymbol{y}) \right] - E_{p(\boldsymbol{y})} \left[ \ln \left( E_{p(X \mid \boldsymbol{y})} \left[ \frac{1}{N-1} \sum_{i=2}^{N} f(\boldsymbol{x}_i, \boldsymbol{y}) \right] \right) \right]$$
$$+ E_{p(\boldsymbol{y})} \left[ \text{KL}(p(\boldsymbol{x}_1 \mid \boldsymbol{y}) \, \| \, q(\boldsymbol{x}_1 \mid \boldsymbol{y})) \right]$$

$$= E_{p(\boldsymbol{x}_1, \boldsymbol{y})} \left[ \ln f(\boldsymbol{x}_1, \boldsymbol{y}) \right] - E_{p(\boldsymbol{y})} \left[ E_{p(X \mid \boldsymbol{y})} \left[ \ln \left( \frac{1}{N-1} \sum_{i=2}^{N} f(\boldsymbol{x}_i, \boldsymbol{y}) \right) \right] \right]$$
$$+ E_{p(\boldsymbol{y})} \left[ E_{p(X \mid \boldsymbol{y})} \left[ \ln \left( \frac{1}{N-1} \sum_{i=2}^{N} f(\boldsymbol{x}_i, \boldsymbol{y}) \right) \right] \right] - E_{p(\boldsymbol{y})} \left[ \ln \left( E_{p(X \mid \boldsymbol{y})} \left[ \frac{1}{N-1} \sum_{i=2}^{N} f(\boldsymbol{x}_i, \boldsymbol{y}) \right] \right) \right]$$
$$+ E_{p(\boldsymbol{y})} \left[ \text{KL}(p(\boldsymbol{x}_1 \mid \boldsymbol{y}) \, \| \, q(\boldsymbol{x}_1 \mid \boldsymbol{y})) \right]$$

$$= E_{p(\boldsymbol{y})} \left[ E_{p(X \mid \boldsymbol{y})} \left[ \ln \left( \frac{f(\boldsymbol{x}_1, \boldsymbol{y})}{\frac{1}{N-1} \sum_{i=2}^{N} f(\boldsymbol{x}_i, \boldsymbol{y})} \right) \right] \right] + E_{p(\boldsymbol{y})} \left[ E_{p(X \mid \boldsymbol{y})} \left[ \ln \left( \frac{1}{N-1} \sum_{i=2}^{N} f(\boldsymbol{x}_i, \boldsymbol{y}) \right) \right] \right]$$
$$- E_{p(\boldsymbol{y})} \left[ \ln \left( E_{p(X \mid \boldsymbol{y})} \left[ \frac{1}{N-1} \sum_{i=2}^{N} f(\boldsymbol{x}_i, \boldsymbol{y}) \right] \right) \right] + E_{p(\boldsymbol{y})} \left[ \text{KL}(p(\boldsymbol{x}_1 \mid \boldsymbol{y}) \, \| \, q(\boldsymbol{x}_1 \mid \boldsymbol{y})) \right]$$

$$= I_{\text{InfoLOOB}}(X_1 ; Y)$$
$$+ E_{p(\boldsymbol{y})} \left[ E_{p(X \mid \boldsymbol{y})} \left[ \ln \left( \frac{1}{N-1} \sum_{i=2}^{N} f(\boldsymbol{x}_i, \boldsymbol{y}) \right) \right] \right] - E_{p(\boldsymbol{y})} \left[ \ln \left( E_{p(X \mid \boldsymbol{y})} \left[ \frac{1}{N-1} \sum_{i=2}^{N} f(\boldsymbol{x}_i, \boldsymbol{y}) \right] \right) \right]$$
$$+ E_{p(\boldsymbol{y})} \left[ \text{KL}(p(\boldsymbol{x}_1 \mid \boldsymbol{y}) \, \| \, q(\boldsymbol{x}_1 \mid \boldsymbol{y})) \right]$$
$$= I_{\text{InfoLOOB}}(X_1 ; Y)$$
$$+ E_{p(\boldsymbol{y})} \left[ E_{p(X \mid \boldsymbol{y})} \left[ \ln \left( \frac{1}{N-1} \sum_{i=2}^{N} f(\boldsymbol{x}_i, \boldsymbol{y}) \right) \right] \right] - E_{p(\boldsymbol{y})} \left[ \ln \left( E_{p(\boldsymbol{x}_1)} \left[ f(\boldsymbol{x}_1, \boldsymbol{y}) \right] \right) \right]$$
$$+ E_{p(\boldsymbol{y})} \left[ \text{KL}(p(\boldsymbol{x}_1 \mid \boldsymbol{y}) \, \| \, q(\boldsymbol{x}_1 \mid \boldsymbol{y})) \right]$$
$$= I_{\text{InfoLOOB}}(X_1 ; Y)$$
$$- E_{p(\boldsymbol{y})} \left[ E_{p(\tilde{X})} \left[ \ln \left( \frac{E_{p(\boldsymbol{x}_1)} \left[ f(\boldsymbol{x}_1, \boldsymbol{y}) \right]}{\frac{1}{N-1} \sum_{i=2}^{N} f(\boldsymbol{x}_i, \boldsymbol{y})} \right) \right] \right]$$
$$+ E_{p(\boldsymbol{y})} \left[ \text{KL}(p(\boldsymbol{x}_1 \mid \boldsymbol{y}) \, \| \, q(\boldsymbol{x}_1 \mid \boldsymbol{y})) \right]$$
$$= I_{\text{InfoLOOB}}(X_1 ; Y) - \text{DE} + E_{p(\boldsymbol{y})} \left[ \text{KL}(p(\boldsymbol{x}_1 \mid \boldsymbol{y}) \, \| \, q(\boldsymbol{x}_1 \mid \boldsymbol{y})) \right] ,$$

where we used

$$\text{DE} = E_{p(\boldsymbol{y})} \left[ E_{p(\tilde{X})} \left[ \ln \left( \frac{E_{p(\boldsymbol{x}_1)} \left[ f(\boldsymbol{x}_1, \boldsymbol{y}) \right]}{\frac{1}{N-1} \sum_{i=2}^{N} f(\boldsymbol{x}_i, \boldsymbol{y})} \right) \right] \right] = E_{p(\boldsymbol{y})} \left[ E_{p(\tilde{X})} \left[ \ln \left( \frac{Z(\boldsymbol{y})}{\frac{1}{N-1} \sum_{i=2}^{N} f(\boldsymbol{x}_i, \boldsymbol{y})} \right) \right] \right]$$
$$\tag{A70}$$

(DE for difference of expectations) and

$$Z(\boldsymbol{y}) \;=\; \mathrm{E}_{p(\boldsymbol{x}_1)}\left[f(\boldsymbol{x}_1, \boldsymbol{y})\right] \;=\; \mathrm{E}_{p(\tilde{X})}\left[\frac{1}{N-1}\sum_{i=2}^{N} f(\boldsymbol{x}_i, \boldsymbol{y})\right] \tag{A71}$$

$$=\; \mathrm{E}_{p(X|\boldsymbol{y})}\left[\frac{1}{N-1}\sum_{i=2}^{N} f(\boldsymbol{x}_i, \boldsymbol{y})\right] \;.$$

Since both KL and DE are non-negative (for DE see below), to increase InfoLOOB we have either to decrease KL or to increase DE.

**Bounding** DE. Next we bound DE. We define

$$\mathrm{L} \;=\; \boldsymbol{z}^T\boldsymbol{x} \;-\; \beta^{-1}\sum_{i=1}^{N} z_i \ln z_i \;. \tag{A72}$$

The *log-sum-exp function* (lse) is

$$\mathrm{lse}(\beta, \boldsymbol{a}) \;=\; \beta^{-1}\log\left(\sum_{i=1}^{N}\exp(\beta a_i)\right) \;, \tag{A73}$$

for $\beta > 0$ and vector $\boldsymbol{a} = (a_1, \ldots, a_N)$.

The lse is a convex function (Lemma 4 in (Gao & Pavel, 2017)). We obtain via Jensen's inequality and the convex lse:

$$\mathrm{E}_{p(\boldsymbol{y})}\left[\mathrm{E}_{p(\tilde{X})}\left[\ln\left(\mathrm{E}_{p(\boldsymbol{x}_1)}\left[\frac{\exp(\tau^{-1}\,\mathrm{sim}(\boldsymbol{x}_1, \boldsymbol{y}))}{\frac{1}{N-1}\sum_{i=2}^{N}\exp(\tau^{-1}\,\mathrm{sim}(\boldsymbol{x}_i, \boldsymbol{y}))}\right]\right)\right]\right] \tag{A74}$$
$$\leqslant\; \mathrm{E}_{p(\boldsymbol{y})}\left[\ln\mathrm{E}_{p(\boldsymbol{x}_1)}\left[\exp(\tau^{-1}\,\mathrm{sim}(\boldsymbol{x}_1, \boldsymbol{y}))\right] \;-\; \tau^{-1}\,\mathrm{E}_{p(\boldsymbol{x}_1)}\left[\mathrm{sim}(\boldsymbol{x}_1, \boldsymbol{y})\right]\right] \;.$$

Again using Jensen's inequality and the concave ln, we get

$$\mathrm{E}_{p(\boldsymbol{y})}\left[\mathrm{E}_{p(\tilde{X})}\left[\ln\left(\mathrm{E}_{p(\boldsymbol{x}_1)}\left[\frac{\exp(\tau^{-1}\,\mathrm{sim}(\boldsymbol{x}_1, \boldsymbol{y}))}{\frac{1}{N-1}\sum_{i=2}^{N}\exp(\tau^{-1}\,\mathrm{sim}(\boldsymbol{x}_i, \boldsymbol{y}))}\right]\right)\right]\right] \tag{A75}$$
$$\geq\; \mathrm{E}_{p(\boldsymbol{y})}\left[\ln\mathrm{E}_{p(\boldsymbol{x}_1)}\left[\exp(\tau^{-1}\,\mathrm{sim}(\boldsymbol{x}_1, \boldsymbol{y}))\right] \;-\; \ln\left(\frac{1}{N-1}\sum_{i=2}^{N}\mathrm{E}_{p(\boldsymbol{x}_i)}\left[\exp(\tau^{-1}\,\mathrm{sim}(\boldsymbol{x}_1, \boldsymbol{y}))\right]\right)\right]$$
$$=\; 0 \;.$$

If we combine both previous inequalities, we obtain

$$0 \;\leqslant\; \mathrm{DE} \;\leqslant\; \mathrm{E}_{p(\boldsymbol{y})}\left[\ln\mathrm{E}_{p(\boldsymbol{x}_1)}\left[\exp(\tau^{-1}\,\mathrm{sim}(\boldsymbol{x}_1, \boldsymbol{y}))\right] \;-\; \tau^{-1}\,\mathrm{E}_{p(\boldsymbol{x}_1)}\left[\mathrm{sim}(\boldsymbol{x}_1, \boldsymbol{y})\right]\right] \;. \tag{A76}$$

In particular, for bounded $\mathrm{sim}(\boldsymbol{x}_1, \boldsymbol{y})$, we get

$$0 \;\leqslant\; \mathrm{DE} \;\leqslant\; \tau^{-1}\left(\max_{\boldsymbol{y}, \boldsymbol{x}_1}\mathrm{sim}(\boldsymbol{x}_1, \boldsymbol{y}) \;-\; \min_{\boldsymbol{y}, \boldsymbol{x}_1}\mathrm{sim}(\boldsymbol{x}_1, \boldsymbol{y})\right) \;, \tag{A77}$$

while Hoeffding's lemma gives

$$0 \;\leqslant\; \mathrm{DE} \;\leqslant\; \frac{1}{8}\tau^{-2}\left(\max_{\boldsymbol{y}, \boldsymbol{x}_1}\mathrm{sim}(\boldsymbol{x}_1, \boldsymbol{y}) \;-\; \min_{\boldsymbol{y}, \boldsymbol{x}_1}\mathrm{sim}(\boldsymbol{x}_1, \boldsymbol{y})\right)^2 \;. \tag{A78}$$

Thus, for bounded $\mathrm{sim}(\boldsymbol{x}_1, \boldsymbol{y})$, DE is bounded, therefore also InfoLOOB.

Next, we show that DE is small. The Hoeffding bound (Proposition 2.5 in Wainwright (2019)) states that if the random variable $S_f = f(X_1, \boldsymbol{y})$ with $X_1 \sim p(\boldsymbol{x}_1)$ is $\sigma_f^2$-sub-Gaussian then with random variables $S_f^i$ iid distributed like $S_f$

$$p\left( \left| \mathrm{E}\left[S_f\right] - \frac{1}{N-1} \sum_{i=2}^{N} S_f^i) \right| \geq \epsilon \right) \tag{A79}$$

$$= p\left( \left| \mathrm{E}_{p(\boldsymbol{x}_1)}\left[f(\boldsymbol{x}_1, \boldsymbol{y})\right] - \frac{1}{N-1} \sum_{i=2}^{N} f(\boldsymbol{x}_i, \boldsymbol{y}) \right| \geq \epsilon \right) \leq 2 \exp\left( -\frac{(N-1)\,\epsilon^2}{2\,\sigma_f^2} \right) . \tag{A80}$$

If $S_f \in [a, b]$ (e.g. if $f(\boldsymbol{x}, \boldsymbol{y}) \in [a, b]$) then we can set $\sigma_f = (b-a)/2$.

For

$$\mathrm{E}_{p(\boldsymbol{x}_1)}\left[f(\boldsymbol{x}_1, \boldsymbol{y})\right] - \frac{1}{N-1} \sum_{i=2}^{N} f(\boldsymbol{x}_i, \boldsymbol{y}) \leq \epsilon \tag{A81}$$

we have

$$\ln\left( \frac{\mathrm{E}_{p(\boldsymbol{x}_1)}\left[f(\boldsymbol{x}_1, \boldsymbol{y})\right]}{\frac{1}{N-1}\sum_{i=2}^{N} f(\boldsymbol{x}_i, \boldsymbol{y})} \right) \leq \ln\left( \frac{\frac{1}{N-1}\sum_{i=2}^{N} f(\boldsymbol{x}_i, \boldsymbol{y}) + \epsilon}{\frac{1}{N-1}\sum_{i=2}^{N} f(\boldsymbol{x}_i, \boldsymbol{y})} \right) \tag{A82}$$

$$\leq \frac{\epsilon}{\frac{1}{N-1}\sum_{i=2}^{N} f(\boldsymbol{x}_i, \boldsymbol{y})} \leq \frac{\epsilon}{Z - \epsilon} ,$$

where we used $\ln a \leq a - 1$ for $0 < a$. Analog for

$$\frac{1}{N-1} \sum_{i=2}^{N} f(\boldsymbol{x}_i, \boldsymbol{y}) - \mathrm{E}_{p(\boldsymbol{x}_1)}\left[f(\boldsymbol{x}_1, \boldsymbol{y})\right] \leq \epsilon \tag{A83}$$

we have

$$\ln\left( \frac{\mathrm{E}_{p(\boldsymbol{x}_1)}\left[f(\boldsymbol{x}_1, \boldsymbol{y})\right]}{\frac{1}{N-1}\sum_{i=2}^{N} f(\boldsymbol{x}_i, \boldsymbol{y})} \right) \geq \ln\left( \frac{\mathrm{E}_{p(\boldsymbol{x}_1)}\left[f(\boldsymbol{x}_1, \boldsymbol{y})\right]}{\mathrm{E}_{p(\boldsymbol{x}_1)}\left[f(\boldsymbol{x}_1, \boldsymbol{y})\right] + \epsilon} \right) \tag{A84}$$

$$= - \ln\left( \frac{\mathrm{E}_{p(\boldsymbol{x}_1)}\left[f(\boldsymbol{x}_1, \boldsymbol{y})\right] + \epsilon}{\mathrm{E}_{p(\boldsymbol{x}_1)}\left[f(\boldsymbol{x}_1, \boldsymbol{y})\right]} \right) \geq - \frac{\epsilon}{\mathrm{E}_{p(\boldsymbol{x}_1)}\left[f(\boldsymbol{x}_1, \boldsymbol{y})\right]} = - \frac{\epsilon}{Z} ,$$

where we used $-\ln a \geq 1 - a$ for $0 < a$.

In summary, if for all $\boldsymbol{y}$

$$\left| \mathrm{E}_{p(\boldsymbol{x}_1)}\left[f(\boldsymbol{x}_1, \boldsymbol{y})\right] - \frac{1}{N-1} \sum_{i=2}^{N} f(\boldsymbol{x}_i, \boldsymbol{y}) \right| \leq \epsilon , \tag{A85}$$

then we have

$$- \frac{\epsilon}{Z} \leq \ln\left( \frac{\mathrm{E}_{p(\boldsymbol{x}_1)}\left[f(\boldsymbol{x}_1, \boldsymbol{y})\right]}{\frac{1}{N-1}\sum_{i=2}^{N} f(\boldsymbol{x}_i, \boldsymbol{y})} \right) \leq \frac{\epsilon}{Z - \epsilon} . \tag{A86}$$

It follows that

$$- \epsilon\, \mathrm{E}_{p(\boldsymbol{y})}\left[Z(\boldsymbol{y})^{-1}\right] \leq \mathrm{DE} \leq \epsilon\, \mathrm{E}_{p(\boldsymbol{y})}\left[(Z(\boldsymbol{y}) - \epsilon)^{-1}\right] . \tag{A87}$$

Consequently, for large $N$, the Hoeffding bound Eq. (A79) holds with high probability, if $\epsilon$ is chosen reasonably. Therefore, with high probability the term DE is small.

Next, we show that DE is governed by the variance of $\mathrm{sim}(\boldsymbol{x}, \boldsymbol{y})$ for unmatched pairs.

In Eq. (A76) the term DE is upper bounded:

$$\text{DE} \leqslant \text{E}_{p(\boldsymbol{y})}\left[\ln \text{E}_{p(\boldsymbol{x}_1)}\left[\exp(\tau^{-1}\,\text{sim}(\boldsymbol{x}_1,\boldsymbol{y}))\right] - \tau^{-1}\,\text{E}_{p(\boldsymbol{x}_1)}\left[\text{sim}(\boldsymbol{x}_1,\boldsymbol{y})\right]\right] \tag{A88}$$

We express these equations via the random variable $S = \text{sim}(X_1, \boldsymbol{y})$ with $s = \text{sim}(\boldsymbol{x}_1, \boldsymbol{y})$, which replaces the random variable $X_1$ and its realization $\boldsymbol{x}_1$.

$$\text{E}_{p(\boldsymbol{x}_1)}\left[g\left(\text{sim}(\boldsymbol{x}_1,\boldsymbol{y})\right)\right] = \int p(\boldsymbol{x}_1)\, g\left(\text{sim}(\boldsymbol{x}_1,\boldsymbol{y})\right)\,\mathrm{d}\boldsymbol{x}_1 \tag{A89}$$

$$= \int p(\boldsymbol{x}_1)\int \delta\left(s - \text{sim}(\boldsymbol{x}_1,\boldsymbol{y})\right)\,g(s)\,\mathrm{d}s\,\mathrm{d}\boldsymbol{x}_1 = \int\int p(\boldsymbol{x}_1)\,\delta\left(s - \text{sim}(\boldsymbol{x}_1,\boldsymbol{y})\right)\,\mathrm{d}\boldsymbol{x}_1\,g(s)\,\mathrm{d}s$$

$$= \int p(s)\,g(s)\,\mathrm{d}s = \text{E}_{p(s)}\left[g(s)\right]\,,$$

where we used the Dirac delta-distribution $\delta$ and for $s = \text{sim}(\boldsymbol{x}_1, \boldsymbol{y})$ we defined

$$p(s) = \int p(\boldsymbol{x}_1)\,\delta\left(s - \text{sim}(\boldsymbol{x}_1,\boldsymbol{y})\right)\,\mathrm{d}\boldsymbol{x}_1\,. \tag{A90}$$

Eq. (A76) can be written as

$$\text{DE} \leqslant \text{E}_{p(\boldsymbol{y})}\left[\ln \text{E}_{p(s)}\left[\exp(\tau^{-1}\,s)\right] - \tau^{-1}\,\text{E}_{p(s)}\left[s\right]\right] = \text{E}_{p(\boldsymbol{y})}\left[\ln \text{E}_{p(s)}\left[\exp(\tau^{-1}\,(s - \bar{s}))\right]\right]\,, \tag{A91}$$

with $\bar{s} = \text{E}_{p(s)}\left[s\right]$.

We assume that the random variable $S$ with realization $s = \text{sim}(\boldsymbol{x}_1, \boldsymbol{y})$ is sub-Gaussian, where $\boldsymbol{y}$ is given and $\boldsymbol{x}_1$ is drawn independently from $\boldsymbol{y}$ according to $p(\boldsymbol{x}_1)$. Therefore, we assume that the similarity of a random matching is sub-Gaussian. This assumption is true for bounded similarities (like the cosine similarity) and for almost sure bounded similarities. The assumption is true if using vectors that are retrieved from a continuous modern Hopfield network since they are bounded by the largest stored vector. This assumption is true for a continuous similarity function if $\boldsymbol{x}$, $\boldsymbol{y}$, and parameters are bounded, since the bounded $\boldsymbol{x}$, $\boldsymbol{y}$, and parameters can be embedded in a compact set on which the similarity has a maximum. This assumption is true for learned similarities if the input is bounded.

For a random variable $S$ that is $\sigma^2$-sub-Gaussian (Definition 2.2 in Wainwright (2019)) the constant $\sigma^2$ is called a *proxy variance*. The minimal proxy variance $\sigma^2_{\text{opt}}$ is called the *optimal proxy variance* with $\text{Var}[S] \leqslant \sigma^2_{\text{opt}}$ (Arbel et al., 2019). $S$ is *strictly* sub-Gaussian, if $\sigma^2_{\text{opt}} = \text{Var}[S]$. Proposition 2.1 in Arbel et al. (2019) states

$$\sigma^2_{\text{opt}} = \sup_{\lambda \in \mathbb{R}} \frac{2}{\lambda^2}\ln\left(\text{E}\left[\exp(\lambda(S - \mu))\right]\right)\,, \tag{A92}$$

with $\mu = \text{E}[S]$. The supremum is attained for almost surely bounded random variables $S$. Eq. (4) in Arbel et al. (2019) states

$$\lim_{\lambda \to 0} \frac{2}{\lambda^2}\ln\left(\text{E}\left[\exp(\lambda(S - \mu))\right]\right) = \text{Var}[S]\,. \tag{A93}$$

Thus, for $S$ being sub-Gaussian, we have

$$\text{DE} \leqslant \tau^{-2}\,\text{E}_{p(\boldsymbol{y})}\left[\sigma^2_{\text{opt}}(S)\right]\,, \tag{A94}$$

where $\sigma^2_{\text{opt}}$ is the optimal proxy variance of $S$. For example, bounded random variables $S \in [a, b]$ are sub-Gaussian with $\sigma_{\text{opt}} \leqslant (b - a)/2$ (Exercise 2.4 in Wainwright (2019)).

KL is decreased by making the variation distribution $q(\boldsymbol{x}_1 \mid \boldsymbol{y})$ more similar to the posterior $p(\boldsymbol{x}_1 \mid \boldsymbol{y})$. The value DE only depends on the marginal distributions $p(\boldsymbol{y})$ and $p(\boldsymbol{x})$, since $p(\tilde{X}) = \prod_{i=2}^N p(\boldsymbol{x}_i)$. The value DE can be changed by adding an offset to $f(\boldsymbol{x}, \boldsymbol{y})$. However, scaling $f(\boldsymbol{x}, \boldsymbol{y})$ by a factor does not change DE. Consequently, DE is difficult to change.

Therefore, increasing InfoLOOB is most effective by making $q(\boldsymbol{x}_1 \mid \boldsymbol{y})$ more similar to the posterior $p(\boldsymbol{x}_1 \mid \boldsymbol{y})$.

**Gradient of InfoLOOB expressed by gradients of** KL **and** DE. Assume that the similarity is parametrized by $\boldsymbol{w}$ giving $\text{sim}(\boldsymbol{x}, \boldsymbol{y}; \boldsymbol{w})$.

$$\text{KL}(p(\boldsymbol{x}_1 \mid \boldsymbol{y}) \parallel q(\boldsymbol{x}_1 \mid \boldsymbol{y})) = \int p(\boldsymbol{x}_1 \mid \boldsymbol{y}) \ln \left( \frac{p(\boldsymbol{x}_1 \mid \boldsymbol{y})}{q(\boldsymbol{x}_1 \mid \boldsymbol{y})} \right) \, \mathrm{d}\boldsymbol{x} \qquad \text{(A95)}$$

$$= -\tau^{-1} \int p(\boldsymbol{x}_1 \mid \boldsymbol{y}) \, \text{sim}(\boldsymbol{x}_1, \boldsymbol{y}; \boldsymbol{w}) \, \mathrm{d}\boldsymbol{x}_1 + \ln Z + C \, ,$$

where $C$ is independent of $\boldsymbol{w}$.

Next, we compute the derivative of KL with respect to parameters $\boldsymbol{w}$.

$$\frac{\partial \text{KL}}{\partial \boldsymbol{w}} \qquad\qquad\qquad\qquad\qquad\qquad\qquad\qquad\qquad\qquad\qquad\qquad \text{(A96)}$$

$$= -\tau^{-1} \int p(\boldsymbol{x}_1 \mid \boldsymbol{y}) \frac{\partial \text{sim}(\boldsymbol{x}_1, \boldsymbol{y}; \boldsymbol{w})}{\partial \boldsymbol{w}} \, \mathrm{d}\boldsymbol{x}_1 + \frac{1}{Z} \int p(\boldsymbol{x}_1) \frac{\exp(\tau^{-1} \text{sim}(\boldsymbol{x}_1, \boldsymbol{y}; \boldsymbol{w}))}{\partial \text{sim}(\boldsymbol{x}_1, \boldsymbol{y}; \boldsymbol{w})} \frac{\partial \text{sim}(\boldsymbol{x}_1, \boldsymbol{y}; \boldsymbol{w})}{\partial \boldsymbol{w}} \, \mathrm{d}\boldsymbol{x}_1$$

$$= -\tau^{-1} \int p(\boldsymbol{x}_1 \mid \boldsymbol{y}) \frac{\partial \text{sim}(\boldsymbol{x}_1, \boldsymbol{y}; \boldsymbol{w})}{\partial \boldsymbol{w}} \, \mathrm{d}\boldsymbol{x}_1 + \tau^{-1} \int p(\boldsymbol{x}_1) \frac{\exp(\tau^{-1} \text{sim}(\boldsymbol{x}_1, \boldsymbol{y}; \boldsymbol{w}))}{Z} \frac{\partial \text{sim}(\boldsymbol{x}_1, \boldsymbol{y}; \boldsymbol{w})}{\partial \boldsymbol{w}} \, \mathrm{d}\boldsymbol{x}_1$$

$$= -\tau^{-1} \int p(\boldsymbol{x}_1 \mid \boldsymbol{y}) \frac{\partial \text{sim}(\boldsymbol{x}_1, \boldsymbol{y}; \boldsymbol{w})}{\partial \boldsymbol{w}} \, \mathrm{d}\boldsymbol{x}_1 + \tau^{-1} \int q(\boldsymbol{x}_1 \mid \boldsymbol{y}) \frac{\partial \text{sim}(\boldsymbol{x}_1, \boldsymbol{y}; \boldsymbol{w})}{\partial \boldsymbol{w}} \, \mathrm{d}\boldsymbol{x}_1$$

$$= \tau^{-1} \int (q(\boldsymbol{x}_1 \mid \boldsymbol{y}) - p(\boldsymbol{x}_1 \mid \boldsymbol{y})) \frac{\partial \text{sim}(\boldsymbol{x}_1, \boldsymbol{y}; \boldsymbol{w})}{\partial \boldsymbol{w}} \, \mathrm{d}\boldsymbol{x}_1 \, .$$

The derivative is the average difference between the posterior distribution $p(\boldsymbol{x}_1 \mid \boldsymbol{y})$ and the variational distribution $q(\boldsymbol{x}_1 \mid \boldsymbol{y})$ multiplied by the derivative of the similarity function. If both distribution match, then the derivative vanishes.

Next, we compute the derivative of DE with respect to parameters $\boldsymbol{w}$.

$$\frac{\partial \mathrm{DE}}{\partial \boldsymbol{w}} \tag{A97}$$

$$= \mathrm{E}_{p(\boldsymbol{y})}\left[\frac{\partial \ln Z}{\partial \boldsymbol{w}}\right] - \mathrm{E}_{p(\boldsymbol{y})}\left[\mathrm{E}_{p(\tilde{X})}\left[\frac{\frac{1}{N-1}\sum_{i=2}^{N}\tau^{-1}\exp(\tau^{-1}\mathrm{sim}(\boldsymbol{x}_i,\boldsymbol{y};\boldsymbol{w}))\frac{\partial \mathrm{sim}(\boldsymbol{x}_i,\boldsymbol{y};\boldsymbol{w})}{\partial \boldsymbol{w}}}{\frac{1}{N-1}\sum_{j=2}^{N}f(\boldsymbol{x}_j,\boldsymbol{y})}\right]\right]$$

$$= \mathrm{E}_{p(\boldsymbol{y})}\left[\tau^{-1}\int q(\boldsymbol{x}_1\mid\boldsymbol{y})\frac{\partial \mathrm{sim}(\boldsymbol{x}_1,\boldsymbol{y};\boldsymbol{w})}{\partial \boldsymbol{w}}\,\mathrm{d}\boldsymbol{x}_1\right]$$

$$- \mathrm{E}_{p(\boldsymbol{y})}\left[\mathrm{E}_{p(\tilde{X})}\left[\frac{\frac{1}{N-1}\sum_{i=2}^{N}\tau^{-1}\exp(\tau^{-1}\mathrm{sim}(\boldsymbol{x}_i,\boldsymbol{y};\boldsymbol{w}))\frac{\partial \mathrm{sim}(\boldsymbol{x}_i,\boldsymbol{y};\boldsymbol{w})}{\partial \boldsymbol{w}}}{\frac{1}{N-1}\sum_{j=2}^{N}f(\boldsymbol{x}_j,\boldsymbol{y})}\right]\right]$$

$$= \tau^{-1}\,\mathrm{E}_{p(\boldsymbol{y})}\left[\int q(\boldsymbol{x}_1\mid\boldsymbol{y})\frac{\partial \mathrm{sim}(\boldsymbol{x}_1,\boldsymbol{y};\boldsymbol{w})}{\partial \boldsymbol{w}}\,\mathrm{d}\boldsymbol{x}_1\right]$$

$$- \tau^{-1}\,\mathrm{E}_{p(\boldsymbol{y})}\left[\mathrm{E}_{p(\tilde{X})}\left[\frac{1}{N-1}\sum_{i=2}^{N}\frac{f(\boldsymbol{x}_i,\boldsymbol{y})}{\frac{1}{N-1}\sum_{j=2}^{N}f(\boldsymbol{x}_j,\boldsymbol{y})}\frac{\partial \mathrm{sim}(\boldsymbol{x}_i,\boldsymbol{y};\boldsymbol{w})}{\partial \boldsymbol{w}}\right]\right]$$

$$= \tau^{-1}\,\mathrm{E}_{p(\boldsymbol{y})}\left[\int\frac{p(\boldsymbol{x}_1)\,f(\boldsymbol{x}_1,\boldsymbol{y})}{\mathrm{E}_{p(\boldsymbol{x})}[f(\boldsymbol{x},\boldsymbol{y})]}\frac{\partial \mathrm{sim}(\boldsymbol{x}_1,\boldsymbol{y};\boldsymbol{w})}{\partial \boldsymbol{w}}\,\mathrm{d}\boldsymbol{x}_1\right]$$

$$- \tau^{-1}\,\mathrm{E}_{p(\boldsymbol{y})}\left[\mathrm{E}_{p(\tilde{X})}\left[\frac{1}{N-1}\sum_{i=2}^{N}\frac{f(\boldsymbol{x}_i,\boldsymbol{y})}{\frac{1}{N-1}\sum_{j=2}^{N}f(\boldsymbol{x}_j,\boldsymbol{y})}\frac{\partial \mathrm{sim}(\boldsymbol{x}_i,\boldsymbol{y};\boldsymbol{w})}{\partial \boldsymbol{w}}\right]\right]$$

$$= \tau^{-1}\,\mathrm{E}_{p(\boldsymbol{y})}\left[\mathrm{E}_{p(\boldsymbol{x}_1)}\left[\frac{f(\boldsymbol{x}_1,\boldsymbol{y})}{\mathrm{E}_{p(\boldsymbol{x})}[f(\boldsymbol{x},\boldsymbol{y})]}\frac{\partial \mathrm{sim}(\boldsymbol{x}_1,\boldsymbol{y};\boldsymbol{w})}{\partial \boldsymbol{w}}\right]\right]$$

$$- \tau^{-1}\,\mathrm{E}_{p(\boldsymbol{y})}\left[\mathrm{E}_{p(\tilde{X})}\left[\frac{1}{N-1}\sum_{i=2}^{N}\frac{f(\boldsymbol{x}_i,\boldsymbol{y})}{\frac{1}{N-1}\sum_{j=2}^{N}f(\boldsymbol{x}_j,\boldsymbol{y})}\frac{\partial \mathrm{sim}(\boldsymbol{x}_i,\boldsymbol{y};\boldsymbol{w})}{\partial \boldsymbol{w}}\right]\right]$$

$$= \tau^{-1}\,\mathrm{E}_{p(\boldsymbol{y})}\left[\frac{1}{N-1}\sum_{i=2}^{N}\mathrm{E}_{p(\boldsymbol{x}_i)}\left[\frac{f(\boldsymbol{x}_i,\boldsymbol{y})}{\mathrm{E}_{p(\boldsymbol{x})}[f(\boldsymbol{x},\boldsymbol{y})]}\frac{\partial \mathrm{sim}(\boldsymbol{x}_i,\boldsymbol{y};\boldsymbol{w})}{\partial \boldsymbol{w}}\right]\right]$$

$$- \tau^{-1}\,\mathrm{E}_{p(\boldsymbol{y})}\left[\mathrm{E}_{p(\tilde{X})}\left[\frac{1}{N-1}\sum_{i=2}^{N}\frac{f(\boldsymbol{x}_i,\boldsymbol{y})}{\frac{1}{N-1}\sum_{j=2}^{N}f(\boldsymbol{x}_j,\boldsymbol{y})}\frac{\partial \mathrm{sim}(\boldsymbol{x}_i,\boldsymbol{y};\boldsymbol{w})}{\partial \boldsymbol{w}}\right]\right]$$

$$= \tau^{-1}\,\mathrm{E}_{p(\boldsymbol{y})}\left[\mathrm{E}_{p(\tilde{X})}\left[\frac{1}{N-1}\sum_{i=2}^{N}\frac{f(\boldsymbol{x}_i,\boldsymbol{y})}{\mathrm{E}_{p(\boldsymbol{x})}[f(\boldsymbol{x},\boldsymbol{y})]}\frac{\partial \mathrm{sim}(\boldsymbol{x}_i,\boldsymbol{y};\boldsymbol{w})}{\partial \boldsymbol{w}}\right]\right]$$

$$- \tau^{-1}\,\mathrm{E}_{p(\boldsymbol{y})}\left[\mathrm{E}_{p(\tilde{X})}\left[\frac{1}{N-1}\sum_{i=2}^{N}\frac{f(\boldsymbol{x}_i,\boldsymbol{y})}{\frac{1}{N-1}\sum_{j=2}^{N}f(\boldsymbol{x}_j,\boldsymbol{y})}\frac{\partial \mathrm{sim}(\boldsymbol{x}_i,\boldsymbol{y};\boldsymbol{w})}{\partial \boldsymbol{w}}\right]\right]$$

$$= \tau^{-1}\,\mathrm{E}_{p(\boldsymbol{y})}\left[\mathrm{E}_{p(\tilde{X})}\left[\frac{1}{N-1}\sum_{i=2}^{N}\left(\frac{1}{\mathrm{E}_{p(\boldsymbol{x})}[f(\boldsymbol{x},\boldsymbol{y})]}-\frac{1}{\frac{1}{N-1}\sum_{j=2}^{N}f(\boldsymbol{x}_j,\boldsymbol{y})}\right)f(\boldsymbol{x}_i,\boldsymbol{y})\frac{\partial \mathrm{sim}(\boldsymbol{x}_i,\boldsymbol{y};\boldsymbol{w})}{\partial \boldsymbol{w}}\right]\right]$$

$$= \tau^{-1}\,\mathrm{E}_{p(\boldsymbol{y})}\left[\mathrm{E}_{p(\tilde{X})}\left[\frac{1}{N-1}\sum_{i=2}^{N}\left(\frac{1}{Z}-\frac{1}{\frac{1}{N-1}\sum_{j=2}^{N}f(\boldsymbol{x}_j,\boldsymbol{y})}\right)f(\boldsymbol{x}_i,\boldsymbol{y})\frac{\partial \mathrm{sim}(\boldsymbol{x}_i,\boldsymbol{y};\boldsymbol{w})}{\partial \boldsymbol{w}}\right]\right]\;.$$

The derivative is the average of $\frac{1}{Z} - \frac{1}{\frac{1}{N-1}\sum_{j=2}^{N}f(\boldsymbol{x}_j,\boldsymbol{y})}$ multiplied by the score function and the derivative of the similarity function. The average is over $\boldsymbol{y}$ and $\tilde{X}$, therefore the whole derivative becomes even smaller. Consequently, for small $b-a$ and large $N$, the derivative of DE is small.

Note that for

$$\left|\mathrm{E}_{p(\boldsymbol{x}_1)}[f(\boldsymbol{x}_1,\boldsymbol{y})] - \frac{1}{N-1}\sum_{i=2}^{N}f(\boldsymbol{x}_i,\boldsymbol{y})\right| \leqslant \epsilon \tag{A98}$$

we have

$$\frac{1}{Z} - \frac{1}{\frac{1}{N-1}\sum_{j=2}^{N} f(\boldsymbol{x}_j, \boldsymbol{y})} \;\leqslant\; \frac{1}{Z} - \frac{1}{Z+\epsilon} \;=\; \frac{\epsilon}{Z(Z+\epsilon)}\;,\tag{A99}$$

$$\frac{1}{Z} - \frac{1}{\frac{1}{N-1}\sum_{j=2}^{N} f(\boldsymbol{x}_j, \boldsymbol{y})} \;\geq\; \frac{1}{Z} - \frac{1}{Z-\epsilon} \;=\; -\frac{\epsilon}{Z(Z-\epsilon)}\;,\tag{A100}$$

therefore

$$\left| \frac{1}{Z} - \frac{1}{\frac{1}{N-1}\sum_{j=2}^{N} f(\boldsymbol{x}_j, \boldsymbol{y})} \right| \;\leqslant\; \frac{\epsilon}{Z(Z-\epsilon)}\;.\tag{A101}$$

If the expectation $Z$ is well approximated by the average $\frac{1}{N-1}\sum_{j=2}^{N} f(\boldsymbol{x}_j, \boldsymbol{y})$, then both DE and its gradient are small.

Derivative of InfoLOOB via KL and DE:

$$\frac{\partial \mathrm{I}_{\mathrm{InfoLOOB}}(X_1\;;\;Y)}{\partial \boldsymbol{w}} \;=\; \frac{\partial \mathrm{DE}}{\partial \boldsymbol{w}} - \frac{\partial \mathrm{KL}}{\partial \boldsymbol{w}}\;.\tag{A102}$$

In this gradient, the KL term is dominating, therefore $f(\boldsymbol{x}, \boldsymbol{y})$ is pushed to approximate the conditional probability $p(\boldsymbol{y} \mid \boldsymbol{x})$. Modern Hopfield networks lead to larger values of $p(\boldsymbol{y} \mid \boldsymbol{x})$ as the mutual information becomes larger, therefore modern Hopfield networks help to push $f(\boldsymbol{x}, \boldsymbol{y})$ to large values. Furthermore, modern Hopfield networks increase $Z$, which is in the denominator of the bound on DE and its derivative.

### A.1.4  InfoNCE and InfoLOOB: Gradients

We consider the InfoNCE and the InfoLOOB loss function. For computing the loss function, we sample $N$ pairs independently from $p(\boldsymbol{x}, \boldsymbol{y})$, which gives the training set $\{(\boldsymbol{x}_1, \boldsymbol{y}_1), (\boldsymbol{x}_2, \boldsymbol{y}_2), \dots, (\boldsymbol{x}_N, \boldsymbol{y}_N)\}$. InfoNCE and InfoLOOB only differ in using the positive example in the negatives. More precisely, for the sample $\boldsymbol{y}_1$, InfoNCE uses for the matrix of negative samples $\boldsymbol{X} = (\boldsymbol{x}_1, \dots, \boldsymbol{x}_N)$, while InfoLOOB uses $\tilde{\boldsymbol{X}} = (\boldsymbol{x}_2, \dots, \boldsymbol{x}_N)$.

**InfoNCE.** The InfoNCE loss is

$$\mathrm{L}_{\mathrm{InfoNCE}} \;=\; -\frac{1}{N}\sum_{i=1}^{N} \ln\left( \frac{f(\boldsymbol{x}_i, \boldsymbol{y}_i)}{\frac{1}{N}\sum_{j=1}^{N} f(\boldsymbol{x}_j, \boldsymbol{y}_i)} \right) \;=\; \frac{1}{N}\sum_{i=1}^{N} \mathrm{L}_{\mathrm{InfoNCE}}(\boldsymbol{y}_i)\;,\tag{A103}$$

where we used

$$\mathrm{L}_{\mathrm{InfoNCE}}(\boldsymbol{y}_i) \;=\; -\ln\left( \frac{f(\boldsymbol{x}_i, \boldsymbol{y}_i)}{\frac{1}{N}\sum_{j=1}^{N} f(\boldsymbol{x}_j, \boldsymbol{y}_i)} \right)\;.\tag{A104}$$

For the score function $f(\boldsymbol{x}, \boldsymbol{y})$, we use

$$f(\boldsymbol{x}, \boldsymbol{y}) \;=\; \exp(\tau^{-1}\,\mathrm{sim}(\boldsymbol{x}, \boldsymbol{y}))\;,\tag{A105}$$

$$\mathrm{sim}(\boldsymbol{x}, \boldsymbol{y}) \;=\; \boldsymbol{y}^T \boldsymbol{x}\tag{A106}$$

with $\tau$ as the temperature.

The loss function for this score function is

$$\mathrm{L}_{\mathrm{InfoNCE}}(\boldsymbol{y}) \;=\; -\tau^{-1}\,\boldsymbol{y}^T\boldsymbol{x}_1 + \tau^{-1}\,\mathrm{lse}\left(\tau^{-1}, \boldsymbol{X}^T\boldsymbol{y}\right)\;,\tag{A107}$$

where lse is the *log-sum-exp function* (lse):

$$\mathrm{lse}(\beta, \boldsymbol{a}) \;=\; \beta^{-1}\log\left( \sum_{i=1}^{N} \exp(\beta a_i) \right)\;,\tag{A108}$$

for $\beta > 0$ and vector $\boldsymbol{a} = (a_1, \dots, a_N)$.

The gradient with respect to $\boldsymbol{y}$ is

$$\frac{\partial \text{L}_{\text{InfoNCE}}(\boldsymbol{y})}{\partial \boldsymbol{y}} = -\tau^{-1}\,\boldsymbol{x}_1 + \tau^{-1}\,\boldsymbol{X}\,\text{softmax}\left(\tau^{-1}\boldsymbol{X}^T\boldsymbol{y}\right), \tag{A109}$$

which is the positive example $\boldsymbol{x}_1$ that fits to the anchor example $\boldsymbol{y}$ minus the Hopfield network update with state pattern $\boldsymbol{y}$ and stored patterns $\boldsymbol{X}$ and then this difference multiplied by $\tau^{-1}$.

This gradient can be simplified, since the positive example $\boldsymbol{x}_1$ is also in the negative examples. Using $\boldsymbol{p} = (p_1, \ldots, p_N)^T = \text{softmax}\left(\tau^{-1}\boldsymbol{X}^T\boldsymbol{y}\right)$, we obtain

$$\frac{\partial \text{L}_{\text{InfoNCE}}(\boldsymbol{y})}{\partial \boldsymbol{y}} \tag{A110}$$

$$= -\tau^{-1}\,(1 - p_1)\,\left(\boldsymbol{x}_1 - \frac{1}{1 - p_1}\,\boldsymbol{X}\,\left(\text{softmax}\left(\tau^{-1}\boldsymbol{X}^T\boldsymbol{y}\right) - (p_1, 0, \ldots, 0)^T\right)\right)$$

$$= -\tau^{-1}\,(1 - p_1)\,\left(\boldsymbol{x}_1 - \tilde{\boldsymbol{X}}\,\text{softmax}\left(\tau^{-1}\tilde{\boldsymbol{X}}^T\boldsymbol{y}\right)\right) = (1 - p_1)\,\frac{\partial \text{L}_{\text{InfoLOOB}}(\boldsymbol{y})}{\partial \boldsymbol{y}}\,.$$

where

$$\frac{1}{1 - p_1}\,\boldsymbol{X}\,\left(\text{softmax}\left(\tau^{-1}\boldsymbol{X}^T\boldsymbol{y}\right) - (p_1, 0, \ldots, 0)^T\right) \tag{A111}$$

$$= \frac{1}{1 - p_1}\boldsymbol{X}\,\left((p_1, p_2, \ldots, p_N)^T - (p_1, 0, \ldots, 0)^T\right)$$

$$= \frac{1}{1 - p_1}\boldsymbol{X}(0, p_2, \ldots, p_N)^T = \frac{1}{1 - p_1}\sum_{i=2}^{N} p_i\,\boldsymbol{x}_i$$

is the softmax average over the negatives $\boldsymbol{x}_i$ for $2 \leqslant i \leqslant N$ without $\boldsymbol{x}_1$. It can be easily seen that $\frac{1}{1-p_1}\sum_{i=2}^{N} p_i = \frac{1-p_1}{1-p_1} = 1$. For the derivative of the InfoLOOB see below.

The gradient with respect to $\boldsymbol{x}_1$ is

$$\frac{\partial \text{L}_{\text{InfoNCE}}(\boldsymbol{y})}{\partial \boldsymbol{x}_1} = -\tau^{-1}\,\boldsymbol{y} + \tau^{-1}\,\frac{\exp(\tau^{-1}\,\boldsymbol{x}_1^T\boldsymbol{y})}{\sum_{i=1}^{N}\exp(\tau^{-1}\boldsymbol{x}_i^T\boldsymbol{y})}\,\boldsymbol{y} \tag{A112}$$

$$= -\tau^{-1}\,(1 - p_1)\,\boldsymbol{y}\,. \tag{A113}$$

Consequently, the learning rate is scaled by $(1 - p_1)$.

The sum of gradients with respect to $\boldsymbol{x}_1$ and $\boldsymbol{x}_i$ is

$$\frac{\partial \text{L}_{\text{InfoNCE}}(\boldsymbol{y})}{\partial \boldsymbol{x}_1} + \sum_{i=1}^{N}\frac{\partial \text{L}_{\text{InfoNCE}}(\boldsymbol{y})}{\partial \boldsymbol{x}_i} = -\tau^{-1}\,\boldsymbol{y} + \tau^{-1}\,\boldsymbol{y}\,\mathbf{1}^T\text{softmax}\left(\tau^{-1}\boldsymbol{X}^T\boldsymbol{y}\right) \tag{A114}$$

$$= -\tau^{-1}\,\boldsymbol{y} + \tau^{-1}\,\boldsymbol{y} = 0\,,$$

where $\mathbf{1}$ is the vector with ones. However, the derivatives with respect to the weights are not zero since the $\boldsymbol{x}_i$ are differently computed.

**InfoLOOB.** The InfoLOOB loss is

$$\text{L}_{\text{InfoLOOB}} = -\frac{1}{N}\sum_{i=1}^{N}\ln\left(\frac{f(\boldsymbol{x}_i, \boldsymbol{y}_i)}{\frac{1}{N-1}\sum_{j=1,j\neq i}^{N}f(\boldsymbol{x}_j, \boldsymbol{y}_i)}\right) = \frac{1}{N}\sum_{i=1}^{N}\text{L}_{\text{InfoLOOB}}(\boldsymbol{y}_i)\,, \tag{A115}$$

where we used

$$\text{L}_{\text{InfoLOOB}}(\boldsymbol{y}_i) = -\ln\left(\frac{f(\boldsymbol{x}_i, \boldsymbol{y}_i)}{\frac{1}{N-1}\sum_{j=1,j\neq i}^{N}f(\boldsymbol{x}_j, \boldsymbol{y}_i)}\right)\,. \tag{A116}$$

For the score function $f(\boldsymbol{x}, \boldsymbol{y})$, we use

$$f(\boldsymbol{x}, \boldsymbol{y}) = \exp(\tau^{-1}\,\text{sim}(\boldsymbol{x}, \boldsymbol{y}))\,, \tag{A117}$$

$$\text{sim}(\boldsymbol{x}, \boldsymbol{y}) = \boldsymbol{y}^T\boldsymbol{x} \tag{A118}$$

with $\tau$ as the temperature.

The loss function for this score function is

$$\mathrm{L}_{\mathrm{InfoLOOB}}(\boldsymbol{y}) \ = \ - \ \tau^{-1} \ \boldsymbol{y}^T \boldsymbol{x}_1 \ + \ \tau^{-1} \ \mathrm{lse}\left(\tau^{-1}, \tilde{\boldsymbol{X}}^T \boldsymbol{y}\right) \ , \tag{A119}$$

where lse is the log-sum-exponential function.

The gradient with respect to $\boldsymbol{y}$ is

$$\frac{\partial \mathrm{L}_{\mathrm{InfoLOOB}}(\boldsymbol{y})}{\partial \boldsymbol{y}} \ = \ - \ \tau^{-1} \ \boldsymbol{x}_1 \ + \ \tau^{-1} \ \tilde{\boldsymbol{X}} \ \mathrm{softmax}\left(\tau^{-1}\tilde{\boldsymbol{X}}^T \boldsymbol{y}\right) \ , \tag{A120}$$

which is the positive example $\boldsymbol{x}_1$ that fits to the anchor example $\boldsymbol{y}$ minus the Hopfield network update with state pattern $\boldsymbol{y}$ and stored patterns $\tilde{\boldsymbol{X}}$ and then this difference multiplied by $\tau^{-1}$.

The gradient with respect to $\boldsymbol{x}_1$ is

$$\frac{\partial \mathrm{L}_{\mathrm{InfoLOOB}}(\boldsymbol{y})}{\partial \boldsymbol{x}_1} \ = \ - \ \tau^{-1} \ \boldsymbol{y} \ . \tag{A121}$$

The sum of gradients with respect to $\boldsymbol{x}_1$ and $\boldsymbol{x}_i$ is

$$\frac{\partial \mathrm{L}_{\mathrm{InfoLOOB}}(\boldsymbol{y})}{\partial \boldsymbol{x}_1} \ + \ \sum_i \frac{\partial \mathrm{L}_{\mathrm{InfoLOOB}}(\boldsymbol{y})}{\partial \boldsymbol{x}_i} \ = \ - \ \tau^{-1} \ \boldsymbol{y} \ + \ \tau^{-1} \ \boldsymbol{y} \ \boldsymbol{1}^T \mathrm{softmax}\left(\tau^{-1}\tilde{\boldsymbol{X}}^T \boldsymbol{y}\right) \tag{A122}$$

$$= \ - \ \tau^{-1} \ \boldsymbol{y} \ + \ \tau^{-1} \ \boldsymbol{y} \ = \ 0 \ ,$$

where $\boldsymbol{1}$ is the vector with ones. However, the derivatives with respect to the weights are not zero since the $\boldsymbol{x}_i$ are differently computed.

**Gradients with respect to $\tau^{-1}$.** The gradient of the InfoNCE loss Eq. (A103) using the similarity Eq. (A105) with respect to $\tau^{-1}$ is

$$\frac{\partial \mathrm{L}_{\mathrm{InfoNCE}}(\boldsymbol{y})}{\partial \tau^{-1}} \ = \ - \ \boldsymbol{y}^T \ \boldsymbol{x}_1 \ + \ \boldsymbol{y}^T \ \boldsymbol{X} \ \mathrm{softmax}\left(\tau^{-1}\boldsymbol{X}^T \boldsymbol{y}\right) \tag{A123}$$

$$= \ - \ \boldsymbol{y}^T \ \left(\boldsymbol{x}_1 \ - \ \boldsymbol{X} \ \mathrm{softmax}\left(\tau^{-1}\boldsymbol{X}^T \boldsymbol{y}\right)\right) \ , \tag{A124}$$

which is the similarity of the anchor $\boldsymbol{y}$ with the difference of the positive example $\boldsymbol{x}_1$ and the Hopfield network update with state pattern $\boldsymbol{y}$ and stored patterns $\boldsymbol{X}$. The gradient of the InfoLOOB loss Eq. (A115) using the similarity Eq. (A117) with respect to $\tau^{-1}$ is

$$\frac{\partial \mathrm{L}_{\mathrm{InfoLOOB}}(\boldsymbol{y})}{\partial \tau^{-1}} \ = \ - \ \boldsymbol{y}^T \ \boldsymbol{x}_1 \ + \ \boldsymbol{y}^T \ \tilde{\boldsymbol{X}} \ \mathrm{softmax}\left(\tau^{-1}\tilde{\boldsymbol{X}}^T \boldsymbol{y}\right) \tag{A125}$$

$$= \ - \ \boldsymbol{y}^T \ \left(\boldsymbol{x}_1 \ - \ \tilde{\boldsymbol{X}} \ \mathrm{softmax}\left(\tau^{-1}\tilde{\boldsymbol{X}}^T \boldsymbol{y}\right)\right) \ , \tag{A126}$$

with the difference compared to Eq. (A123) that the Hopfield network update is done with stored patterns $\tilde{\boldsymbol{X}}$ instead of $\boldsymbol{X}$. Without the positive example $\boldsymbol{x}_1$ in the stored patterns $\tilde{\boldsymbol{X}}$, the term $\boldsymbol{x}_1 - \tilde{\boldsymbol{X}} \ \mathrm{softmax}\left(\tau^{-1}\tilde{\boldsymbol{X}}^T \boldsymbol{y}\right)$ in Eq. (A125) will not decrease like the term $\boldsymbol{x}_1 - \boldsymbol{X} \ \mathrm{softmax}\left(\tau^{-1}\boldsymbol{X}^T \boldsymbol{y}\right)$ in Eq. (A123) but grow even larger with better separation of the positive and negative examples.

### A.1.5  InfoLOOB and InfoNCE: Probability Estimators

In McAllester & Stratos (2018, 2020) it was shown that estimators of the mutual information by lower bounds have problems as they come with serious statistical limitations. Statistically more justified for representing the mutual information is a difference of entropies, which are estimated by minimizing the cross-entropy loss. Both InfoNCE and InfoLOOB losses can be viewed as cross-entropy losses.

We sample $N$ pairs independently from $p(\boldsymbol{x}, \boldsymbol{y})$, which gives $Z = \{(\boldsymbol{x}_1, \boldsymbol{y}_1), (\boldsymbol{x}_2, \boldsymbol{y}_2), \dots, (\boldsymbol{x}_N, \boldsymbol{y}_N)\}$. We set $X = \{\boldsymbol{x}_1, \boldsymbol{x}_2, \dots, \boldsymbol{x}_N\}$ and $Y = \{\boldsymbol{y}_1, \boldsymbol{y}_2, \dots, \boldsymbol{y}_N\}$, so that, $Z = X \times Y$. The score function $f(\boldsymbol{x}, \boldsymbol{y})$ is an estimator for $p(\boldsymbol{x}, \boldsymbol{y})$. Then we obtain

estimators $\hat{q}$ for the conditional probabilities. $\hat{q}(\boldsymbol{y}_i \mid \boldsymbol{x}_i, Y \setminus \{\boldsymbol{y}_i\})$ is an estimator for $p(\boldsymbol{y}_i \mid \boldsymbol{x}_i)$ and $\hat{q}(\boldsymbol{x}_i \mid \boldsymbol{y}_i, X \setminus \{\boldsymbol{x}_i\})$ an estimator for $p(\boldsymbol{x}_i \mid \boldsymbol{y}_i)$. Each estimator $\hat{q}$ uses beyond $(\boldsymbol{x}_i, \boldsymbol{y}_i)$ additional samples to estimate the normalizing constant. For InfoNCE these estimators are

$$\hat{q}^1(\boldsymbol{y}_i \mid \boldsymbol{x}_i, Y \setminus \{\boldsymbol{y}_i\}) \;=\; \frac{f(\boldsymbol{x}_i, \boldsymbol{y}_i)}{\frac{1}{N} \sum_{j=1}^{N} f(\boldsymbol{x}_i, \boldsymbol{y}_j)} \;\approx\; \frac{f(\boldsymbol{x}_i, \boldsymbol{y}_i)}{\mathrm{E}_{p(\boldsymbol{y})}\left[f(\boldsymbol{x}_i, \boldsymbol{y})\right]}, \tag{A127}$$

$$\hat{q}^2(\boldsymbol{x}_i \mid \boldsymbol{y}_i, X \setminus \{\boldsymbol{x}_i\}) \;=\; \frac{f(\boldsymbol{x}_i, \boldsymbol{y}_i)}{\frac{1}{N} \sum_{j=1}^{N} f(\boldsymbol{x}_j, \boldsymbol{y}_i)} \;\approx\; \frac{f(\boldsymbol{x}_i, \boldsymbol{y}_i)}{\mathrm{E}_{p(\boldsymbol{x})}\left[f(\boldsymbol{x}, \boldsymbol{y}_i)\right]} \,. \tag{A128}$$

The cross-entropy losses for the InfoNCE estimators are

$$\mathrm{L}_{\mathrm{InfoNCE}}^1 \;=\; -\frac{1}{N} \sum_{i=1}^{N} \ln\left( \frac{f(\boldsymbol{x}_i, \boldsymbol{y}_i)}{\frac{1}{N} \sum_{j=1}^{N} f(\boldsymbol{x}_i, \boldsymbol{y}_j)} \right) , \tag{A129}$$

$$\mathrm{L}_{\mathrm{InfoNCE}}^2 \;=\; -\frac{1}{N} \sum_{i=1}^{N} \ln\left( \frac{f(\boldsymbol{x}_i, \boldsymbol{y}_i)}{\frac{1}{N} \sum_{j=1}^{N} f(\boldsymbol{x}_j, \boldsymbol{y}_i)} \right) . \tag{A130}$$

For InfoLOOB these estimators are

$$\hat{q}^1(\boldsymbol{y}_i \mid \boldsymbol{x}_i, Y \setminus \{\boldsymbol{y}_i\}) \;=\; \frac{f(\boldsymbol{x}_i, \boldsymbol{y}_i)}{\frac{1}{N-1} \sum_{j=1, j \neq i}^{N} f(\boldsymbol{x}_i, \boldsymbol{y}_j)} \;\approx\; \frac{f(\boldsymbol{x}_i, \boldsymbol{y}_i)}{\mathrm{E}_{p(\boldsymbol{y})}\left[f(\boldsymbol{x}_i, \boldsymbol{y})\right]} , \tag{A131}$$

$$\hat{q}^2(\boldsymbol{x}_i \mid \boldsymbol{y}_i, X \setminus \{\boldsymbol{x}_i\}) \;=\; \frac{f(\boldsymbol{x}_i, \boldsymbol{y}_i)}{\frac{1}{N-1} \sum_{j=1, j \neq i}^{N} f(\boldsymbol{x}_j, \boldsymbol{y}_i)} \;\approx\; \frac{f(\boldsymbol{x}_i, \boldsymbol{y}_i)}{\mathrm{E}_{p(\boldsymbol{x})}\left[f(\boldsymbol{x}, \boldsymbol{y}_i)\right]} \,. \tag{A132}$$

The cross-entropy losses for the InfoLOOB estimators are

$$\mathrm{L}_{\mathrm{InfoLOOB}}^1 \;=\; -\frac{1}{N} \sum_{i=1}^{N} \ln\left( \frac{f(\boldsymbol{x}_i, \boldsymbol{y}_i)}{\frac{1}{N-1} \sum_{j=1, j \neq i}^{N} f(\boldsymbol{x}_i, \boldsymbol{y}_j)} \right) , \tag{A133}$$

$$\mathrm{L}_{\mathrm{InfoLOOB}}^2 \;=\; -\frac{1}{N} \sum_{i=1}^{N} \ln\left( \frac{f(\boldsymbol{x}_i, \boldsymbol{y}_i)}{\frac{1}{N-1} \sum_{j=1, j \neq i}^{N} f(\boldsymbol{x}_j, \boldsymbol{y}_i)} \right) . \tag{A134}$$

The InfoLOOB estimator uses for normalization

$$\mathrm{E}_{p(\boldsymbol{x})}\left[f(\boldsymbol{x}, \boldsymbol{y}_i)\right] \;\approx\; \frac{1}{N-1} \sum_{j=1, j \neq i}^{N} f(\boldsymbol{x}_j, \boldsymbol{y}_i) \,, \tag{A135}$$

$$\mathrm{E}_{p(\boldsymbol{y})}\left[f(\boldsymbol{x}_i, \boldsymbol{y})\right] \;\approx\; \frac{1}{N-1} \sum_{j=1, j \neq i}^{N} f(\boldsymbol{x}_i, \boldsymbol{y}_j) \,, \tag{A136}$$

in contrast to InfoNCE, which uses

$$\mathrm{E}_{p(\boldsymbol{x})}\left[f(\boldsymbol{x}, \boldsymbol{y}_i)\right] \;\approx\; \frac{1}{N} \sum_{j=1}^{N} f(\boldsymbol{x}_j, \boldsymbol{y}_i) \,, \tag{A137}$$

$$\mathrm{E}_{p(\boldsymbol{y})}\left[f(\boldsymbol{x}_i, \boldsymbol{y})\right] \;\approx\; \frac{1}{N} \sum_{j=1}^{N} f(\boldsymbol{x}_i, \boldsymbol{y}_j) \,. \tag{A138}$$

If InfoNCE estimates the normalizing constant separately, then it would be biased. $(\boldsymbol{x}_i, \boldsymbol{y}_i)$ is drawn according to $p(\boldsymbol{x}_i, \boldsymbol{y}_i)$ instead of $p(\boldsymbol{x}_i)p(\boldsymbol{y}_i)$. In contrast, if InfoLOOB estimated the normalizing constant separately, then it would be unbiased.

### A.1.6 InfoLOOB and InfoNCE: Losses

We have $N$ pairs drawn iid from $p(\boldsymbol{x}, \boldsymbol{y})$, where we assume that a pair $(\boldsymbol{x}_i, \boldsymbol{y}_i)$ is already an embedding of the original drawn pair. These build up the embedding training set $Z =$

$\{(\boldsymbol{x}_1, \boldsymbol{y}_1), (\boldsymbol{x}_2, \boldsymbol{y}_2), \dots, (\boldsymbol{x}_N, \boldsymbol{y}_N)\}$ that allows to construct the matrices $\boldsymbol{X} = (\boldsymbol{x}_1, \boldsymbol{x}_2, \dots, \boldsymbol{x}_N)$ of $N$ embedding samples $\boldsymbol{x}_i$ and $\boldsymbol{Y} = (\boldsymbol{y}_1, \boldsymbol{y}_2, \dots, \boldsymbol{y}_N)$ of $N$ embedding samples $\boldsymbol{y}_i$. We also have $M$ stored patterns $\boldsymbol{U} = (\boldsymbol{u}_1, \dots, \boldsymbol{u}_M)$ and $K$ stored patterns $\boldsymbol{V} = (\boldsymbol{v}_1, \dots, \boldsymbol{v}_K)$.

The state vectors $\boldsymbol{x}_i$ and $\boldsymbol{y}_i$ are the queries for the Hopfield networks, which retrieve some vectors from $\boldsymbol{U}$ or $\boldsymbol{V}$. We normalize vectors $\|\boldsymbol{x}_i\| = \|\boldsymbol{y}_i\| = \|\boldsymbol{u}_i\| = \|\boldsymbol{v}_i\| = 1$. The following vectors are retrieved from modern Hopfield networks (Ramsauer et al., 2021):

$$\boldsymbol{U}_{\boldsymbol{x}_i} = \boldsymbol{U}\,\mathrm{softmax}(\beta\,\boldsymbol{U}^T\boldsymbol{x}_i)\,, \quad \text{(A139)} \qquad \boldsymbol{V}_{\boldsymbol{x}_i} = \boldsymbol{V}\,\mathrm{softmax}(\beta\,\boldsymbol{V}^T\boldsymbol{x}_i)\,, \quad \text{(A141)}$$

$$\boldsymbol{U}_{\boldsymbol{y}_i} = \boldsymbol{U}\,\mathrm{softmax}(\beta\,\boldsymbol{U}^T\boldsymbol{y}_i)\,, \quad \text{(A140)} \qquad \boldsymbol{V}_{\boldsymbol{y}_i} = \boldsymbol{V}\,\mathrm{softmax}(\beta\,\boldsymbol{V}^T\boldsymbol{y}_i)\,. \quad \text{(A142)}$$

where $\boldsymbol{U}_{\boldsymbol{x}_i}$ denotes an image-retrieved image embedding, $\boldsymbol{U}_{\boldsymbol{y}_i}$ a text-retrieved image embedding, $\boldsymbol{V}_{\boldsymbol{x}_i}$ an image-retrieved text embedding and $\boldsymbol{V}_{\boldsymbol{y}_i}$ a text-retrieved text embedding. The hyperparameter $\beta$ corresponds to the inverse temperature: $\beta = 0$ retrieves the average of the stored pattern, while large $\beta$ retrieve the stored pattern that is most similar to the state pattern (query).

We consider the loss functions

$$\mathrm{L}_{\mathrm{InfoNCE}} = -\frac{1}{N}\sum_{i=1}^{N}\log\frac{\exp(\tau^{-1}\boldsymbol{x}_i^T\boldsymbol{y}_i)}{\sum_{j=1}^{N}\exp(\tau^{-1}\boldsymbol{x}_i^T\boldsymbol{y}_j)} - \frac{1}{N}\sum_{i=1}^{N}\log\frac{\exp(\tau^{-1}\boldsymbol{x}_i^T\boldsymbol{y}_i)}{\sum_{j=1}^{N}\exp(\tau^{-1}\boldsymbol{x}_j^T\boldsymbol{y}_i)}\,, \tag{A143}$$

$$\mathrm{L}_{\mathrm{InfoLOOB}} = -\frac{1}{N}\sum_{i=1}^{N}\log\frac{\exp(\tau^{-1}\boldsymbol{x}_i^T\boldsymbol{y}_i)}{\sum_{j\neq i}^{N}\exp(\tau^{-1}\boldsymbol{x}_i^T\boldsymbol{y}_j)} - \frac{1}{N}\sum_{i=1}^{N}\log\frac{\exp(\tau^{-1}\boldsymbol{x}_i^T\boldsymbol{y}_i)}{\sum_{j\neq i}^{N}\exp(\tau^{-1}\boldsymbol{x}_j^T\boldsymbol{y}_i)}\,, \tag{A144}$$

$$\mathrm{L}_{\mathrm{InfoLOOB}}^{\mathrm{H-UVUV}} = -\frac{1}{N}\sum_{i=1}^{N}\log\frac{\exp(\tau^{-1}\boldsymbol{U}_{\boldsymbol{x}_i}^T\boldsymbol{V}_{\boldsymbol{y}_i})}{\sum_{j\neq i}^{N}\exp(\tau^{-1}\boldsymbol{U}_{\boldsymbol{x}_i}^T\boldsymbol{V}_{\boldsymbol{y}_j})} - \frac{1}{N}\sum_{i=1}^{N}\log\frac{\exp(\tau^{-1}\boldsymbol{U}_{\boldsymbol{x}_i}^T\boldsymbol{V}_{\boldsymbol{y}_i})}{\sum_{j\neq i}^{N}\exp(\tau^{-1}\boldsymbol{U}_{\boldsymbol{x}_j}^T\boldsymbol{V}_{\boldsymbol{y}_i})}\,, \tag{A145}$$

$$\mathrm{L}_{\mathrm{InfoLOOB}}^{\mathrm{H-UUVV}} = -\frac{1}{N}\sum_{i=1}^{N}\log\frac{\exp(\tau^{-1}\boldsymbol{U}_{\boldsymbol{x}_i}^T\boldsymbol{U}_{\boldsymbol{y}_i})}{\sum_{j\neq i}^{N}\exp(\tau^{-1}\boldsymbol{U}_{\boldsymbol{x}_i}^T\boldsymbol{U}_{\boldsymbol{y}_j})} - \frac{1}{N}\sum_{i=1}^{N}\log\frac{\exp(\tau^{-1}\boldsymbol{V}_{\boldsymbol{x}_i}^T\boldsymbol{V}_{\boldsymbol{y}_i})}{\sum_{j\neq i}^{N}\exp(\tau^{-1}\boldsymbol{V}_{\boldsymbol{x}_j}^T\boldsymbol{V}_{\boldsymbol{y}_i})}\,, \tag{A146}$$

where for InfoLOOB the sum $\sum_{j\neq i}$ in the denominator contains only negative examples $j$. We do not consider the loss function $\mathrm{L}_{\mathrm{InfoLOOB}}^{\mathrm{H-UVUV}}$ because of the high variance in the dot product $\boldsymbol{U}_{\boldsymbol{x}_i}^T\boldsymbol{V}_{\boldsymbol{y}_i}$ as elaborated in the following.

Let us consider the dot product between the anchor retrieval with the positive pattern retrieval for the loss functions with Hopfield. In the first term of the loss function Eq. (A145), $\boldsymbol{U}_{\boldsymbol{x}_i}$ is the anchor with $\boldsymbol{V}_{\boldsymbol{y}_i}$ as the positive sample and $\boldsymbol{V}_{\boldsymbol{y}_i}$ with $\boldsymbol{U}_{\boldsymbol{x}_i}$ as the positive sample for the second term, since the anchor also appears in each term of the denominator. Equivalently the same is valid for Eq. (A146), but with positive samples $\boldsymbol{V}_{\boldsymbol{x}_i}$ and $\boldsymbol{U}_{\boldsymbol{y}_i}$ respectively. These dot products can be written as

$$\boldsymbol{U}_{\boldsymbol{x}_i}^T\boldsymbol{V}_{\boldsymbol{y}_i} = \mathrm{softmax}(\beta\,\boldsymbol{U}^T\boldsymbol{x}_i)^T\,\boldsymbol{U}^T\boldsymbol{V}\,\mathrm{softmax}(\beta\,\boldsymbol{V}^T\boldsymbol{y}_i)\,, \tag{A147}$$

$$\boldsymbol{U}_{\boldsymbol{x}_i}^T\boldsymbol{U}_{\boldsymbol{y}_i} = \mathrm{softmax}(\beta\,\boldsymbol{U}^T\boldsymbol{x}_i)^T\,\boldsymbol{U}^T\boldsymbol{U}\,\mathrm{softmax}(\beta\,\boldsymbol{U}^T\boldsymbol{y}_i)\,, \tag{A148}$$

$$\boldsymbol{V}_{\boldsymbol{x}_i}^T\boldsymbol{V}_{\boldsymbol{y}_i} = \mathrm{softmax}(\beta\,\boldsymbol{V}^T\boldsymbol{x}_i)^T\,\boldsymbol{V}^T\boldsymbol{V}\,\mathrm{softmax}(\beta\,\boldsymbol{V}^T\boldsymbol{y}_i)\,. \tag{A149}$$

**High variance of $\boldsymbol{U}_{\boldsymbol{x}_i}^T\boldsymbol{V}_{\boldsymbol{y}_i}$.** To compute the dot product $\boldsymbol{U}_{\boldsymbol{x}_i}^T\boldsymbol{V}_{\boldsymbol{y}_i}$, $M + K$ stored patterns are required ($M$ of the $\boldsymbol{u}_j$ and $K$ of the $\boldsymbol{v}_j$). In contrast, the dot products $\boldsymbol{U}_{\boldsymbol{x}_i}^T\boldsymbol{U}_{\boldsymbol{y}_i}$ and $\boldsymbol{V}_{\boldsymbol{x}_i}^T\boldsymbol{V}_{\boldsymbol{y}_i}$ require only $M$ or respectively $K$ stored patterns. Therefore, $\boldsymbol{U}_{\boldsymbol{x}_i}^T\boldsymbol{V}_{\boldsymbol{y}_i}$ has higher variance than both $\boldsymbol{U}_{\boldsymbol{x}_i}^T\boldsymbol{U}_{\boldsymbol{y}_i}$ and $\boldsymbol{V}_{\boldsymbol{x}_i}^T\boldsymbol{V}_{\boldsymbol{y}_i}$.

**Covariance structure extracted by $\boldsymbol{U}_{\boldsymbol{x}_i}^T\boldsymbol{U}_{\boldsymbol{y}_i}$ and $\boldsymbol{V}_{\boldsymbol{x}_i}^T\boldsymbol{V}_{\boldsymbol{y}_i}$.** The Jacobian J of the softmax $\boldsymbol{p} = \mathrm{softmax}(\beta\boldsymbol{a})$ is

$$\mathrm{J}(\beta\boldsymbol{a}) = \frac{\partial\mathrm{softmax}(\beta\boldsymbol{a})}{\partial\boldsymbol{a}} = \beta\left(\mathrm{diag}(\boldsymbol{p}) - \boldsymbol{p}\boldsymbol{p}^T\right)\,, \tag{A150}$$

which is a symmetric, positive semi-definite matrix with one eigenvalue of zero for eigenvector $\mathbf{1}$. $J(\beta \boldsymbol{a})$ is diagonally dominant since $|p_i(1 - p_i)| - \sum_{j \neq i} |p_i p_j| = p_i - \sum_j p_i p_j = p_i - p_i = 0$.

Next we give upper bounds on the norm of J.

**Lemma A1.** *For a softmax $\boldsymbol{p} = \mathrm{softmax}(\beta \boldsymbol{x})$ with $m = \max_i p_i(1 - p_i)$, the spectral norm of the Jacobian J of the softmax is bounded:*

$$\|J\|_2 \leqslant 2\, m\, \beta \,, \tag{A151}$$

$$\|J\|_1 \leqslant 2\, m\, \beta \,, \tag{A152}$$

$$\|J\|_\infty \leqslant 2\, m\, \beta \,. \tag{A153}$$

*In particular everywhere holds*

$$\|J\|_2 \leqslant \frac{1}{2}\, \beta \,. \tag{A154}$$

*If $p_{\max} = \max_i p_i \geq 1 - \epsilon \geq 0.5$, then for the spectral norm of the Jacobian holds*

$$\|J\|_2 \leqslant 2\, \epsilon\, \beta \; - \; 2\, \epsilon^2\, \beta \; < \; 2\, \epsilon\, \beta \,. \tag{A155}$$

*Proof.* We consider the maximum absolute column sum norm

$$\|\boldsymbol{A}\|_1 \; = \; \max_j \sum_i |a_{ij}| \tag{A156}$$

and the maximum absolute row sum norm

$$\|\boldsymbol{A}\|_\infty \; = \; \max_i \sum_j |a_{ij}| \,. \tag{A157}$$

We have for $\boldsymbol{A} = J = \beta \left( \mathrm{diag}(\boldsymbol{p}) - \boldsymbol{p}\boldsymbol{p}^T \right)$

$$\sum_j |a_{ij}| \; = \; \beta \left( p_i(1 - p_i) + \sum_{j, j \neq i} p_i p_j \right) \; = \; \beta\, p_i \left( 1 - 2p_i + \sum_j p_j \right) \tag{A158}$$

$$= \; 2\, \beta\, p_i\, (1 - p_i) \; \leqslant \; 2\, m\, \beta \,,$$

$$\sum_i |a_{ij}| \; = \; \beta \left( p_j(1 - p_j) + \sum_{i, i \neq j} p_j p_i \right) \; = \; \beta\, p_j \left( 1 - 2p_j + \sum_i p_i \right) \tag{A159}$$

$$= \; 2\, \beta\, p_j\, (1 - p_j) \; \leqslant \; 2\, m\, \beta \,.$$

Therefore, we have

$$\|J\|_1 \leqslant 2\, m\, \beta \,, \tag{A160}$$

$$\|J\|_\infty \leqslant 2\, m\, \beta \,, \tag{A161}$$

$$\|J\|_2 \leqslant \sqrt{\|J\|_1 \|J\|_\infty} \leqslant 2\, m\, \beta \,. \tag{A162}$$

The last inequality is a direct consequence of Hölder's inequality.

For $0 \leqslant p_i \leqslant 1$, we have $p_i(1 - p_i) \leqslant 0.25$. Therefore, $m \leqslant 0.25$ for all values of $p_i$.

If $p_{\max} \geq 1 - \epsilon \geq 0.5$ ($\epsilon \leqslant 0.5$), then $1 - p_{\max} \leqslant \epsilon$ and for $p_i \neq p_{\max}$ $p_i \leqslant \epsilon$. The derivative $\partial x(1 - x)/\partial x = 1 - 2x > 0$ for $x < 0.5$, therefore $x(1 - x)$ increases with $x$ for $x < 0.5$. Using $x = 1 - p_{\max}$ and for $p_i \neq p_{\max}$ $x = p_i$, we obtain $p_i(1 - p_i) \leqslant \epsilon(1 - \epsilon)$ for all $i$. Consequently, we have $m \leqslant \epsilon(1 - \epsilon)$. $\qquad \square$

For the softmax $\boldsymbol{p} = \mathrm{softmax}(\beta \boldsymbol{a})$ with Jacobian $\partial J/\partial \boldsymbol{a} = J(\beta \boldsymbol{a}) = \beta \left( \mathrm{diag}(\boldsymbol{p}) - \boldsymbol{p}\boldsymbol{p}^T \right)$ and for arbitrary $N$-dimensional vectors $\boldsymbol{b}$ and $\boldsymbol{c}$, we have

$$\boldsymbol{b}^T J(\beta \boldsymbol{a})\, \boldsymbol{c} \; = \; \beta\, \boldsymbol{b}^T \left( \mathrm{diag}(\boldsymbol{p}) - \boldsymbol{p}\,\boldsymbol{p}^T \right)\, \boldsymbol{c} \; = \; \beta \left( \sum_i p_i\, b_i\, c_i - \left( \sum_i p_i\, b_i \right) \left( \sum_i p_i\, c_i \right) \right) \,. \tag{A163}$$

Therefore, $\boldsymbol{b}^T \mathrm{J}(\beta \boldsymbol{a}) \boldsymbol{c}$ is $\beta$ times the covariance between $\boldsymbol{b}$ and $\boldsymbol{c}$ if component $i$ is drawn with probability $p_i$ of the multinomial distribution $\boldsymbol{p}$. In our case the component $i$ is sample $i$.

Using the mean $\hat{\boldsymbol{u}} = 1/M \sum_{i=1}^{M} \boldsymbol{u}_i$, the empirical covariance of data $\boldsymbol{U}$ is

$$\mathrm{Cov}(\boldsymbol{U}) = 1/M \, \boldsymbol{U} \, \boldsymbol{U}^T - \hat{\boldsymbol{u}} \, \hat{\boldsymbol{u}}^T \,, \tag{A164}$$

$$[\mathrm{Cov}(\boldsymbol{U})]_{kl} = \sum_{i=1}^{M} 1/M \, u_{ik} \, u_{il} - \left( \sum_{i=1}^{M} 1/M \, u_{ik} \right) \left( \sum_{i=1}^{M} 1/M \, u_{il} \right) \,. \tag{A165}$$

The weighted covariance (samples $\boldsymbol{u}_i$ are drawn according to $p_i$)

$$\mathrm{Cov}(\boldsymbol{U}) = \boldsymbol{U} \, \mathrm{J}(\beta \, \boldsymbol{a}) \, \boldsymbol{U}^T \,, \tag{A166}$$

$$[\mathrm{Cov}(\boldsymbol{U})]_{kl} = \beta \left( \sum_{i=1}^{M} p_i \, u_{ik} \, u_{il} - \left( \sum_{i=1}^{M} p_i \, u_{ik} \right) \left( \sum_{i=1}^{M} p_i \, u_{il} \right) \right) \,, \tag{A167}$$

which replaces $1/M$ from equal sampling by the $p_i$, that is, $\boldsymbol{u}_i$ is sampled with probability $p_i$.

The next theorem states how to express the dot product $\boldsymbol{U}_{\boldsymbol{x}_i}^T \boldsymbol{U}_{\boldsymbol{y}_i}$ by weighted covariances of the data $\boldsymbol{U}$.

**Theorem A3** (Weighted Covariances). *Using the weighted covariances*

$$\mathrm{Cov}(\boldsymbol{U}, \boldsymbol{y}_i) = \boldsymbol{U} \, \mathrm{J}^{\mathrm{m}}(\beta \, \boldsymbol{U}^T \boldsymbol{y}_i) \, \boldsymbol{U}^T \,, \quad \mathrm{Cov}(\boldsymbol{U}, \boldsymbol{x}_i) = \boldsymbol{U} \, \mathrm{J}^{\mathrm{m}}(\beta \, \boldsymbol{U}^T \boldsymbol{x}_i) \, \boldsymbol{U}^T \,, \tag{A168}$$

$$\mathrm{J}^{\mathrm{m}}(\beta \, \boldsymbol{a}) = \int_0^1 \mathrm{J}(\lambda \, \beta \, \boldsymbol{a}) \, \mathrm{d}\lambda \,, \tag{A169}$$

*where the mean Jacobian $\mathrm{J}^{\mathrm{m}}$ is symmetric, diagonally dominant, and positive semi-definite with spectral norm bounded by $\|\mathrm{J}^{\mathrm{m}}\|_2 \leqslant 0.5\beta$.*

*The dot product $\boldsymbol{U}_{\boldsymbol{x}_i}^T \boldsymbol{U}_{\boldsymbol{y}_i}$ can be expressed by the weighted covariances*

$$\boldsymbol{U}_{\boldsymbol{x}_i}^T \boldsymbol{U}_{\boldsymbol{y}_i} = \left( \bar{\boldsymbol{u}} + \mathrm{Cov}(\boldsymbol{U}, \boldsymbol{x}_i) \, \boldsymbol{x}_i \right)^T \left( \bar{\boldsymbol{u}} + \mathrm{Cov}(\boldsymbol{U}, \boldsymbol{y}_i) \, \boldsymbol{y}_i \right) \,, \tag{A170}$$

*where the mean is $\bar{\boldsymbol{u}} = 1/M \boldsymbol{U} \boldsymbol{1}$.*

*Proof.* We apply the mean value theorem to the softmax with the symmetric, diagonally dominant, positive semi-definite Jacobian matrix $\mathrm{J}^{\mathrm{m}} = \int_0^1 \mathrm{J}(\lambda \boldsymbol{a} + (1 - \lambda)\boldsymbol{a}') \, \mathrm{d}\lambda$:

$$\mathrm{softmax}(\boldsymbol{a}) - \mathrm{softmax}(\boldsymbol{a}') = \mathrm{J}^{\mathrm{m}} \, (\boldsymbol{a} - \boldsymbol{a}') \,. \tag{A171}$$

We set $\boldsymbol{a}' = \boldsymbol{0}$ and use $\beta \boldsymbol{a}$ instead of $\boldsymbol{a}$, which gives:

$$\mathrm{softmax}(\beta \, \boldsymbol{a}) = 1/M \, \boldsymbol{1} + \mathrm{J}^{\mathrm{m}}(\beta \, \boldsymbol{a}) \, \boldsymbol{a} \,, \quad \mathrm{J}^{\mathrm{m}}(\beta \, \boldsymbol{a}) = \int_0^1 \mathrm{J}(\lambda \, \beta \, \boldsymbol{a}) \, \mathrm{d}\lambda \,, \tag{A172}$$

which is exact. We obtain

$$\mathrm{softmax}(\beta \, \boldsymbol{U}^T \boldsymbol{x}_i) = 1/M \, \boldsymbol{1} + \mathrm{J}^{\mathrm{m}}(\beta \, \boldsymbol{U}^T \boldsymbol{x}_i) \, \boldsymbol{U}^T \boldsymbol{x}_i \,, \tag{A173}$$

$$\mathrm{softmax}(\beta \, \boldsymbol{U}^T \boldsymbol{y}_i) = 1/M \, \boldsymbol{1} + \mathrm{J}^{\mathrm{m}}(\beta \, \boldsymbol{U}^T \boldsymbol{y}_i) \, \boldsymbol{U}^T \boldsymbol{y}_i \,. \tag{A174}$$

The spectral norm of $\mathrm{J}^{\mathrm{m}}$ is bounded by $\|\mathrm{J}^{\mathrm{m}}\|_2 \leqslant 0.5\beta$, since this bound holds for every $\mathrm{J}(\lambda\beta\boldsymbol{a})$ in $\mathrm{J}^{\mathrm{m}}(\beta \, \boldsymbol{a}) = \int_0^1 \mathrm{J}(\lambda\beta\boldsymbol{a}) \, \mathrm{d}\lambda$ according to Lemma A1.

The dot product between the anchor retrieval and the positive sample is:

$$\begin{aligned}
\boldsymbol{U}_{\boldsymbol{x}_i}^T \boldsymbol{U}_{\boldsymbol{y}_i} &= \mathrm{softmax}(\beta \, \boldsymbol{U}^T \boldsymbol{x}_i)^T \, \boldsymbol{U}^T \boldsymbol{U} \, \mathrm{softmax}(\beta \, \boldsymbol{U}^T \boldsymbol{y}_i) \\
&= \left( 1/M \, \boldsymbol{1} + \mathrm{J}^{\mathrm{m}}(\beta \, \boldsymbol{U}^T \boldsymbol{x}_i) \, \boldsymbol{U}^T \boldsymbol{x}_i \right)^T \, \boldsymbol{U}^T \boldsymbol{U} \, \left( 1/M \, \boldsymbol{1} + \mathrm{J}^{\mathrm{m}}(\beta \, \boldsymbol{U}^T \boldsymbol{y}_i) \, \boldsymbol{U}^T \boldsymbol{y}_i \right) \\
&= \left( 1/M \, \boldsymbol{U} \, \boldsymbol{1} + \boldsymbol{U} \, \mathrm{J}^{\mathrm{m}}(\beta \, \boldsymbol{U}^T \boldsymbol{x}_i) \, \boldsymbol{U}^T \boldsymbol{x}_i \right)^T \, \left( 1/M \, \boldsymbol{U} \boldsymbol{1} + \boldsymbol{U} \, \mathrm{J}^{\mathrm{m}}(\beta \, \boldsymbol{U}^T \boldsymbol{y}_i) \, \boldsymbol{U}^T \boldsymbol{y}_i \right) \\
&= \left( \bar{\boldsymbol{u}} + \mathrm{Cov}(\boldsymbol{U}, \boldsymbol{x}_i) \, \boldsymbol{x}_i \right)^T \, \left( \bar{\boldsymbol{u}} + \mathrm{Cov}(\boldsymbol{U}, \boldsymbol{y}_i) \, \boldsymbol{y}_i \right) \,,
\end{aligned} \tag{A175}$$

where we used the mean $\bar{\boldsymbol{u}} = 1/M \boldsymbol{U} \boldsymbol{1}$ and the weighted covariances

$$\mathrm{Cov}(\boldsymbol{U}, \boldsymbol{y}_i) = \boldsymbol{U} \, \mathrm{J}^{\mathrm{m}}(\beta \, \boldsymbol{U}^T \boldsymbol{y}_i) \, \boldsymbol{U}^T \,, \quad \mathrm{Cov}(\boldsymbol{U}, \boldsymbol{x}_i) = \boldsymbol{U} \, \mathrm{J}^{\mathrm{m}}(\beta \, \boldsymbol{U}^T \boldsymbol{x}_i) \, \boldsymbol{U}^T \,. \tag{A176}$$

$\square$

The Jacobian $J^m$ is symmetric, diagonally dominant, and positive semi-definite. The weighted covariance $\text{Cov}(\boldsymbol{U}, .)$ is the covariance if the stored pattern $\boldsymbol{u}_i$ is drawn according to an averaged $p_i$ given by $J^m(.)$. Analog for weighted covariance $\text{Cov}(\boldsymbol{V}, .)$. When maximizing the dot product $\boldsymbol{U}_{\boldsymbol{x}_i}^T \boldsymbol{U}_{\boldsymbol{y}_i}$, the normalized vectors $\boldsymbol{x}_i$ and $\boldsymbol{y}_i$ are encouraged to agree on drawing the patterns $\boldsymbol{u}_i$ with the same probability $p_i$ to generate similar weighted covariance matrices $\text{Cov}(\boldsymbol{U}, .)$. If subsets of $\boldsymbol{U}$ have a strong covariance structure, then it can be exploited to produce large weighted covariances and, in turn, large dot products of $\boldsymbol{U}_{\boldsymbol{x}_i}^T \boldsymbol{U}_{\boldsymbol{y}_i}$. Furthermore, for a large dot product $\boldsymbol{U}_{\boldsymbol{x}_i}^T \boldsymbol{U}_{\boldsymbol{y}_i}$, $\boldsymbol{x}_i$ and $\boldsymbol{y}_i$ have to be similar to one another to extract the same direction from the covariance matrices. All considerations are analog for $\boldsymbol{V}_{\boldsymbol{x}_i}^T \boldsymbol{V}_{\boldsymbol{y}_i}$.

## A.2 Experiments

### A.2.1 Ablation studies

As detailed in the main part of the paper, CLOOB has two new main components compared to CLIP: (1) the modern Hopfield networks and (2) the InfoLOOB objective instead of the InfoNCE objective. To assess effects of the new major components of CLOOB, we performed ablation studies on the CC and YFCC datasets. The results are reported in Table A1 for models pre-trained on CC for 31 and 128 epochs and in Table A2 for models pre-trained on YFCC for 28 epochs.

Table A1: Influence of loss functions and Hopfield retrieval for models pre-trained on CC for 31 epochs (left) and 128 epochs (right, indicated by *). Both InfoLOOB and InfoNCE with Hopfield decrease the performance compared to InfoNCE in most of the tasks. InfoLOOB with Hopfield has a strong synergistic effect and therefore considerably improves the performance in 5 out of 8 datasets (epoch 31) and 7 out of 8 datasets (epoch 128) compared to all other models.

| Dataset | InfoNCE | InfoLOOB | InfoNCE Hopfield | InfoLOOB Hopfield | InfoNCE* | InfoLOOB* | InfoNCE Hopfield* | InfoLOOB Hopfield* |
|---|---|---|---|---|---|---|---|---|
| Birdsnap | **2.58** | 2.37 | 1.67 | 2.53 | 2.15 | 1.89 | 2.15 | **3.39** |
| Country211 | 0.53 | 0.63 | 0.54 | **0.76** | 0.62 | 0.62 | 0.66 | **0.79** |
| Flowers102 | 13.16 | 13.03 | 11.53 | **14.24** | 11.79 | 11.57 | 10.86 | **14.24** |
| GTSRB | 4.47 | 4.39 | 5.76 | **5.86** | **9.25** | 6.93 | 6.24 | 8.67 |
| UCF101 | **23.68** | 19.14 | 20.56 | 22.29 | 21.33 | 20.56 | 21.40 | **24.05** |
| Stanford Cars | **1.38** | 1.33 | 1.24 | 1.37 | 1.26 | 1.19 | 1.24 | **1.62** |
| ImageNet | 21.74 | 22.13 | 19.04 | **24.21** | 22.80 | 22.69 | 20.29 | **25.59** |
| ImageNetV2 | 21.45 | 21.65 | 18.97 | **23.80** | 22.44 | 22.13 | 20.22 | **25.50** |

Table A2: Influence of loss functions and Hopfield retrieval for models pre-trained on YFCC for 28 epochs. Both InfoLOOB and InfoNCE with Hopfield decrease the performance compared to InfoNCE in most of the tasks. InfoLOOB with Hopfield has a strong synergistic effect and therefore considerably improves the performance in 6 out of 8 datasets compared to all other models.

| Dataset | InfoNCE | InfoLOOB | InfoNCE Hopfield | InfoLOOB Hopfield |
|---|---|---|---|---|
| Birdsnap | 22.1 | 19.6 | 19.1 | **28.9** |
| Country211 | 7.8 | 7.5 | 6.4 | **7.9** |
| Flowers102 | 48.2 | 50.4 | 43.5 | **55.1** |
| GTSRB | **8.9** | 3.7 | 5.9 | 8.1 |
| UCF101 | **26.7** | 24.0 | 25.4 | 25.3 |
| Stanford Cars | 3.1 | 2.4 | 2.8 | **4.1** |
| ImageNet | 34.0 | 32.2 | 30.4 | **35.7** |
| ImageNetV2 | 32.8 | 30.9 | 29.4 | **34.6** |

For the ablation studies above we use a fixed inverse temperature parameter $\tau^{-1}$ of 30 for all compared models. The value of $\tau^{-1}$ was determined via hyperparameter search (see Section A.2.2).

In contrast to CLIP, we use a learning rate scheduler with restarts (Loshchilov & Hutter, 2017) to be more flexible regarding the number of total training epochs and enable training up to a plateau. To investigate the influence of the learning rate scheduler, we performed experiments with and without

restarts. Table A3 shows the zero-shot performance for the different downstream tasks for CLIP and CLOOB respectively. For both CLIP and CLOOB, the performance at the majority of the tasks either increases or remains roughly the same with restarts.

Table A3: Influence of learning rate scheduler. For most of the tasks the performance either increases or remains roughly the same with restarts for both CLIP and CLOOB.

| Dataset | CLIP | | CLOOB | |
|---|---|---|---|---|
| | w/o restarts | w/ restarts | w/o restarts | w/ restarts |
| Birdsnap | **2.10** | 1.94 | **2.64** | 2.53 |
| Country211 | **0.71** | 0.62 | 0.63 | **0.76** |
| Flowers102 | 11.00 | **13.04** | 11.50 | **14.24** |
| GTSRB | 6.16 | **7.28** | 5.05 | **5.86** |
| UCF101 | 19.05 | **21.00** | 21.97 | **22.29** |
| Stanford Cars | **1.29** | 0.90 | 1.22 | **1.37** |
| ImageNet | 20.19 | **20.31** | 23.29 | **24.21** |
| ImageNet V2 | 20.53 | **20.63** | 22.97 | **23.80** |

### A.2.2 Hyperparameters

The hyperparameter search was done on a validation split of CC with about 15,000 samples. For the hyperparameter $\tau^{-1}$ several values were considered (14.3, 30, 50, 70), where 30 leads to the best results for both YFCC and CC. Analogously to CLIP, we use the Adam optimizer (Kingma et al., 2014) with decoupled weight decay regularization (Loshchilov & Hutter, 2019). The weight decay is only applied to weights that are not gains or biases. As proposed in OpenCLIP (Ilharco et al., 2021) weight decay was set to 0.1. Different choices of weight decay (0.2 or 0.05), did not lead to a relevant performance change. We use the same learning rate of $1 \times 10^{-3}$ for CC and $5 \times 10^{-4}$ for YFCC as used in OpenCLIP. For the hyperparameter $\beta$ we considered values in the range of 5 to 20. A value of 8 resulted in the best performance for CC and 14.3 for YFCC. The batch size for CC was reduced to 512 due to computational restraints which did not result in performance losses. The batch size for YFCC was kept at 1024 as reported by OpenCLIP since a reduction resulted in a significant drop in performance. The learning rate scheduler for all experiments is cosine annealing with warmup and hard restarts (Loshchilov & Hutter, 2017) with a cycle length of 7 epochs. For models trained on YFCC the warmup was set to 10000 steps and for models trained on CC to 20000 steps.

### A.2.3 Datasets

For pre-training we considered two datasets, Conceptual Captions (CC) (Sharma et al., 2018) and YFCC100M (Thomee et al., 2016). The **CC** dataset consists of 2.9 million images and corresponding high-quality captions. Images and their corresponding notations for CC have been gathered via an automated process from the web and therefore represent a wide variety of styles. Raw descriptions of images are collected from the *alt-text* HTML attribute. Both images and texts were filtered such that only image-text pairs above a certain quality threshold are part of this dataset. The dataset we refer to as **YFCC** is a subset of the Yahoo Flickr Creative Commons 100 Million (YFCC100M) dataset. It was created by filtering for images which contain natural language descriptions and/or titles in English resulting in 15 million image-caption pairs. The textual descriptions contain less useful information than CC because they are not filtered by quality. Occasionally they also contain metadata like camera settings or web addresses.

We evaluate and compare our method on several downstream classification tasks. We evaluate on the same set of datasets as CLIP reported for a model trained on YFCC. This set contains Birdsnap (Berg et al., 2014), Country211 (Radford et al., 2021), Flowers102 (Nilsback & Zisserman, 2008), GTSRB (Stallkamp et al., 2011), UCF101 (Soomro et al., 2012), Stanford Cars (Krause et al., 2013) and ImageNet (Deng et al., 2009). We also include ImageNet V2 in our analysis (Recht et al., 2019). Additionally we added zero-shot results for Caltech101 (Fei-Fei et al., 2004), CIFAR10 (Krizhevsky, 2009), CIFAR100 (Krizhevsky, 2009), DTD (Cimpoi et al., 2014), Eurosat (Helber et al., 2018, 2019), FER2013 (Goodfellow et al., 2013), FGVC-Aircraft (Maji et al., 2013), Food101 (Bossard et al.,

Table A4: Datasets used for downstream evaluation. In the case of several train or test sets per dataset we report the total number of samples. It should be noted that at the time of this work some images from the Birdsnap dataset were not accessible anymore.

| Dataset | Classes | Train size | Test size | Evaluation metric |
|---|---|---|---|---|
| Birdsnap | 500 | 38,411 | 1,855 | accuracy |
| Country211 | 211 | 42,200 | 21,100 | accuracy |
| Flowers102 | 102 | 2,040 | 6,149 | class-weighted accuracy |
| GTSRB | 43 | 26,640 | 12,630 | accuracy |
| ImageNet | 1,000 | 1,281,167 | 50,000 | accuracy |
| ImageNet V2 | 1,000 | 1,281,167 | 30,000 | accuracy |
| Stanford Cars | 196 | 8,144 | 8,041 | accuracy |
| UCF101 | 101 | 28,747 | 11,213 | accuracy |
| Caltech101 | 102 | 3,120 | 6,024 | class-weighted accuracy |
| CIFAR10 | 10 | 50,000 | 10,000 | accuracy |
| CIFAR100 | 100 | 50,000 | 10,000 | accuracy |
| DTD | 47 | 3,807 | 1,833 | accuracy |
| Eurosat | 10 | 10,000 | 5,000 | accuracy |
| FER2013 | 7 | 28,709 | 7,178 | accuracy |
| FGVC-Aircraft | 100 | 10,000 | 3,333 | class-weighted accuracy |
| Food101 | 101 | 75,750 | 25,250 | accuracy |
| Pets | 37 | 3,696 | 3,694 | class-weighted accuracy |
| RESISC45 | 45 | 6,300 | 25,200 | accuracy |
| STL10 | 10 | 1,000 | 8,000 | accuracy |
| SUN397 | 397 | 72,763 | 35,991 | accuracy |

2014), Pets (Parkhi et al., 2012), RESISC45 (Cheng et al., 2017), STL10 (Coates et al., 2011) and SUN397 (Xiao et al., 2010).

Table A4 shows an overview of training and test set sizes, number of classes and the applied evaluation metric. In the case of several test sets per dataset the metric is calculated for every set individually and the average performance is reported. The set size in Table A4 corresponds to the total number of samples across all test and training sets of a dataset respectively.

**Birdsnap** contains images of North American bird species, however our dataset is smaller than reported in CLIP as some samples are no longer available. The **Country211** dataset was published in CLIP and is a small subset of the YFCC100m dataset. It consists of photos that can be assigned to 211 countries via GPS coordinates. For each country 200 photos are sampled for the training set and 100 for testing. For the **Flowers102** images of 102 flower categories commonly occuring in the United Kingdom were collected. Several classes are very similar and there is a large variation in scale, pose and lighting. The German Traffic Sign Recognition Benchmark (**GTSRB**) was a challenge held at the IJCNN 2011. The dataset contains images of german traffic signs from more than 40 classes. Note that two versions of this dataset exist, one used for the challenge and an official dataset released after the competition. For CLIP the linear probing classifiers were trained using the competition training set but tested on the official test set. **Stanford Cars** contains images of 196 car models at the level of make, model and year (e.g. Tesla Model S Sedan 2012). **UCF101** (Soomro et al., 2012) is a video dataset with short clips for action recognition consisting of three training sets and three test sets. We follow the procedure reported in CLIP and extract the middle frame of every video to assemble the dataset. The **ImageNet** Large Scale Visual Recognition Challenge was held from 2012 through 2017 and is one of the most widely used benchmarks for object detection and localization. Several years later **ImageNet V2** assembled three new test sets with images from the same 1,000 classes to test for generalization of models optimized for the original ImageNet benchmark. Every test set comprises 10,000 samples.

### A.2.4 Zero-shot evaluation

Class names for all downstream tasks were adopted from CLIP, that is, among other changes special characters like hyphens or apostrophes were removed. Furthermore, some class names of the datasets

Table A5: Zero-shot results for models trained on CC with ResNet-50 vision encoders for two different checkpoints over 20 datasets. Results are given as mean accuracy over 5 runs. Statistically significant results are shown in bold. CLIP and CLOOB were trained for 31 epochs while CLIP* and CLOOB* were trained for 128 epochs.

| Dataset | CLIP RN-50 | CLOOB RN-50 | CLIP* RN-50 | CLOOB* RN-50 |
|---|---|---|---|---|
| Birdsnap | 2.26 ± 0.20 | **3.06 ± 0.30** | 2.8 ± 0.16 | **3.24 ± 0.31** |
| Country211 | 0.67 ± 0.11 | 0.67 ± 0.05 | 0.7 ± 0.04 | 0.73 ± 0.05 |
| Flowers102 | 12.56 ± 0.38 | 13.45 ± 1.19 | 13.32 ± 0.43 | 14.36 ± 1.17 |
| GTSRB | 7.66 ± 1.07 | 6.38 ± 2.11 | 8.96 ± 1.70 | 7.03 ± 1.22 |
| UCF101 | 20.98 ± 1.55 | 22.26 ± 0.72 | 21.63 ± 0.65 | **23.03 ± 0.85** |
| Stanford Cars | 0.91 ± 0.10 | **1.23 ± 0.10** | 0.99 ± 0.16 | **1.41 ± 0.32** |
| ImageNet | 20.33 ± 0.28 | **23.97 ± 0.15** | 21.3 ± 0.42 | **25.67 ± 0.22** |
| ImageNet V2 | 20.24 ± 0.50 | **23.59 ± 0.15** | 21.24 ± 0.22 | **25.49 ± 0.11** |
| Caltech101 | 45.59 ± 0.44 | **48.73 ± 0.94** | 46.39 ± 1.58 | **50.62 ± 0.84** |
| CIFAR10 | **50.18 ± 1.52** | 40.95 ± 2.24 | **53.75 ± 1.49** | 43.48 ± 2.84 |
| CIFAR100 | 20.82 ± 1.45 | 21.59 ± 0.87 | 23.45 ± 1.99 | 24.41 ± 1.27 |
| DTD | 14.7 ± 1.32 | **17.96 ± 2.04** | 16.29 ± 1.30 | 16.51 ± 0.98 |
| Eurosat | 14.86 ± 5.98 | 21.47 ± 4.66 | 16.84 ± 2.28 | 19.56 ± 6.19 |
| FER2013 | **24.67 ± 1.34** | 18.50 ± 1.74 | 22.70 ± 3.99 | 23.52 ± 2.73 |
| FGVC-Aircraft | 1.40 ± 0.27 | 1.31 ± 0.13 | 1.53 ± 0.19 | 1.30 ± 0.37 |
| Food101 | 13.08 ± 0.36 | **16.20 ± 0.38** | 14.88 ± 0.51 | **16.57 ± 0.39** |
| Pets | 12.13 ± 1.87 | 12.93 ± 1.00 | 12.68 ± 0.86 | 13.45 ± 0.68 |
| RESISC45 | 25.85 ± 2.01 | **28.01 ± 1.02** | 25.97 ± 1.56 | **30.54 ± 1.21** |
| STL10 | **82.97 ± 1.82** | 79.11 ± 1.69 | **84.02 ± 0.71** | 82.28 ± 1.22 |
| SUN397 | 38.96 ± 0.35 | **42.29 ± 0.54** | 39.86 ± 0.55 | **44.15 ± 0.27** |

were slightly changed (e.g. "kite" to "kite (bird of prey)" in ImageNet). For zero-shot evaluation, we use the same prompt as published in CLIP. Depending on the dataset the number of prompts can vary from one prompt (e.g. "a photo of a {label}, a type of bird." for Birdsnap) up to 80 prompts for ImageNet covering various settings (e.g. "a cropped photo of a {label}.", "a origami {label}."). In case of several prompts an average embedding over all prompt embeddings is calculated. Figure A4 shows the zero-shot results for all evaluation tasks with the ResNet-50x4 model reported in Table 3.

In addition to the results of the main paper, the zeroshot performance of the models was tested on additional datasets. For details about the additional datasets we refer the reader to Section A.1 of Radford et al. (2021). Table A5 shows the results for models trained on CC. Table A6 shows the results for models trained on YFCC.

### A.2.5 Linear probing

We tried to follow the evaluation procedure in Radford et al. (2021) as closely as possible. We note one difference with respect to the implementation: Instead of scikit-learn's logistic regression using the L-BFGS solver, we use cuML's logistic regression classifier with L-BFGS algorithm to utilize GPUs for efficiency. All hyperparameters are the same as described in Radford et al. (2021), the maximum number of iterations was set to 1000, and the L2 regularization strength $\lambda$ was determined by using a parametric binary search.

We tried to reproduce the CLIP results with the correspondingly published models, however, failed to produce the exact numbers. This could be due to several factors:

- The train and validation split. Same as in Radford et al. (2021) , we use the provided validation set to perform the hyperparameter search. When there is none provided, we use a random half of the training dataset for validation.
- In case of a tie in the validation score, we use the maximal $\lambda$ for the strongest regularization. We note though that we came closer to reproducing the results published in CLIP when using the mean $\lambda$ over all ties when these exist.

Table A6: Zero-shot results for the CLIP reimplementation and CLOOB using different ResNet architectures trained on YFCC over 20 datasets.

| Dataset | RN-50 | | RN-101 | | RN-50x4 | |
| --- | --- | --- | --- | --- | --- | --- |
| | CLIP | CLOOB | CLIP | CLOOB | CLIP | CLOOB |
| Birdsnap | 21.8 | **28.9** | 22.6 | **30.3** | 20.8 | **32.0** |
| Country211 | 6.9 | **7.9** | 7.8 | **8.5** | 8.1 | **9.3** |
| Flowers102 | 48.0 | **55.1** | 48.0 | **55.3** | 50.1 | **54.3** |
| GTSRB | 7.9 | **8.1** | 7.4 | **11.6** | 9.4 | **11.8** |
| UCF101 | **27.2** | 25.3 | 28.6 | **28.8** | 31.0 | **31.9** |
| Stanford Cars | 3.7 | **4.1** | 3.8 | **5.5** | 3.5 | **6.1** |
| ImageNet | 34.6 | **35.7** | 35.3 | **37.1** | 37.7 | **39.0** |
| ImageNet V2 | 33.4 | **34.6** | 34.1 | **35.6** | 35.9 | **37.3** |
| Caltech101 | **55.8** | 53.5 | **57.7** | 56.4 | 57.8 | **58.7** |
| CIFAR10 | **44.3** | 42.4 | **53.9** | 51.4 | **57.0** | 47.4 |
| CIFAR100 | **21.9** | 18.8 | 22.8 | **23.1** | **23.1** | 21.9 |
| DTD | 19.6 | **20.3** | **22.5** | 18.1 | **22.4** | 21.3 |
| Eurosat | 25.0 | **25.9** | **24.9** | 23.0 | 22.1 | **28.5** |
| FER2013 | **14.4** | 11.0 | **29.5** | 17.2 | **31.4** | 16.3 |
| FGVC-Aircraft | 3.5 | **5.6** | 3.0 | **5.8** | 4.5 | **6.4** |
| Food101 | 47.8 | **50.5** | 47.8 | **54.4** | 50.1 | **57.9** |
| Pets | 28.9 | **30.4** | 28.5 | **31.4** | 32.1 | **32.6** |
| RESISC45 | **23.2** | 22.1 | 22.4 | **22.9** | 22.1 | **26.6** |
| STL10 | **85.8** | 81.9 | **88.2** | 81.7 | **88.9** | 83.2 |
| SUN397 | 46.2 | **47.3** | 47.7 | **47.9** | 47.6 | **47.8** |

- For the Birdsnap dataset, the resources that we have got online at the time of this writing could be different from the resources that CLIP's authors obtained at the time.

Linear probing evaluation of YFCC pre-trained models is shown in Table A7. Comparing our reimplementation of CLIP and CLOOB with different ResNet encoders, we observe mixed results. The reason for this effect might be attributed to the observed task-dependence of multimodal models (Devillers et al., 2021). Another potential reason is that the benefit of the restrictions to more reliable patterns that occur in both modalities does not directly translate to an evaluation of just the encoding part of one modality. Again, as expected in self-supervised training, increasing the capacity of the CLOOB models benefits accuracy.

Table A7: Linear probing results for the reimplementation of CLIP and CLOOB using different ResNet architectures trained on YFCC for 28 epochs. The performance of CLOOB scales with increased encoder size.

| Dataset | RN-50 | | RN-101 | | RN-50x4 | |
| --- | --- | --- | --- | --- | --- | --- |
| | CLIP | CLOOB | CLIP | CLOOB | CLIP | CLOOB |
| Birdsnap | 50.9 | **56.2** | 51.6 | **58.1** | 57.6 | **62.2** |
| Country211 | 19.5 | **20.6** | 20.8 | **21.8** | 22.5 | **24.2** |
| Flowers102 | 94.8 | **96.1** | 94.5 | **96.1** | 95.1 | **96.2** |
| GTSRB | **82.5** | 78.9 | **80.3** | 77.9 | **84.6** | 80.6 |
| UCF101 | **75.2** | 72.3 | **76.0** | 72.8 | **77.3** | 75.3 |
| Stanford Cars | 36.2 | **37.7** | 34.9 | **39.0** | 38.5 | **44.3** |
| ImageNet | **66.9** | 65.7 | **67.9** | 67.0 | **70.0** | 69.7 |
| ImageNet V2 | **60.2** | 58.7 | **61.0** | 60.3 | **62.8** | 62.2 |

Table A8: Results for image-to-text and text-to-image retrieval on the CC validation set containing 13,330 samples. Results are given as mean accuracy over 5 runs. Statistically significant results are shown in bold. CLIP and CLOOB were trained for 31 epochs while CLIP* and CLOOB* were trained for 128 epochs.

| Task | CLIP RN-50 | CLOOB RN-50 | CLIP* RN-50 | CLOOB* RN-50 |
|---|---|---|---|---|
| image-to-text R@1 | $0.297 \pm 0.001$ | $\mathbf{0.319 \pm 0.002}$ | $0.316 \pm 0.002$ | $\mathbf{0.342 \pm 0.002}$ |
| image-to-text R@5 | $0.540 \pm 0.003$ | $\mathbf{0.557 \pm 0.001}$ | $0.563 \pm 0.003$ | $\mathbf{0.586 \pm 0.002}$ |
| image-to-text R@10 | $0.638 \pm 0.003$ | $\mathbf{0.651 \pm 0.002}$ | $0.660 \pm 0.002$ | $\mathbf{0.678 \pm 0.001}$ |
| text-to-image R@1 | $0.300 \pm 0.003$ | $\mathbf{0.324 \pm 0.001}$ | $0.316 \pm 0.002$ | $\mathbf{0.348 \pm 0.001}$ |
| text-to-image R@5 | $0.542 \pm 0.003$ | $\mathbf{0.566 \pm 0.001}$ | $0.565 \pm 0.002$ | $\mathbf{0.593 \pm 0.002}$ |
| text-to-image R@10 | $0.638 \pm 0.002$ | $\mathbf{0.655 \pm 0.001}$ | $0.661 \pm 0.001$ | $\mathbf{0.679 \pm 0.001}$ |

### A.2.6  Image-Text retrieval

In addition to zero-shot and linear probing, we tested the models trained on CC in image-to-text retrieval and text-to-image retrieval. The task is to find the matching image to a given text (image-to-text) or, respectively, finding the matching text to a given image (text-to-image). The dataset used for this task is the validation set of CC, which contains 13,330 image-text pairs. We report the results in Table A8. CLOOB significantly outperforms CLIP in both image-to-text and text-to-image retrieval.

### A.2.7  Analysis of the image and text embeddings

Following our ablation studies, we use models trained on CC to disentangle the effects of InfoLOOB and modern Hopfield networks. To track the behaviour during learning, we calculate the embeddings of the validation set of CC (consisting of 13,330 image-caption pairs) for all epochs before a restart of the learning rate scheduler.

We apply the extended uniformity test $A_n$ of Ajne (Ajne, 1968; Prentice, 1978) to the respective embeddings of the image and text encoders. Let $\boldsymbol{X} = (\boldsymbol{x}_1, \dots, \boldsymbol{x}_n)$ be the embeddings of one modality (text or image), consisting of $n$ samples of dimension $d$. The samples are normalized: $\boldsymbol{x}_i = 1$. As specified in Eq. (A177), $A_n$ calculates the difference between a uniform distribution, where all samples on the hypersphere are orthogonal to each other, and the actual distribution of the embedding $\boldsymbol{X}$. Consequently, an embedding with low uniformity results in a high Ajne $A_n$ test statistic:

$$A_n = \frac{n}{4} - \frac{1}{\pi n} \sum_{i=1}^{n} \sum_{j>i}^{n} \cos^{-1}(\boldsymbol{x}_j^T \boldsymbol{x}_i) . \qquad (A177)$$

To understand the influence of modern Hopfield networks, we analyzed the covariance structure of the image and caption embeddings. Similar to Jing et al. (2022), we calculated the sorted eigenvalues of the covariance matrices of image embeddings. Figure A1 shows the sorted eigenvalues of InfoNCE and InfoLOOB with and without Hopfield retrievals during training. Both models without modern Hopfield networks struggle to increase the number of effective eigenvalues which contribute to the variance of the embeddings. InfoNCE with modern Hopfield networks starts with a small number of effective eigenvalues. We attribute this to the saturating effect of InfoNCE, which impedes the modern Hopfield network to extract more covariance. During training the effective eigenvalues steadily increase at a consistent rate. InfoLOOB with Hopfield starts with a high number of effective eigenvalues and strongly improves them during training.

Additionally, we looked at the distribution of similarity scores (dot product) of matched pairs and unmatched pairs over the validation set of CC of embeddings from models trained for 128 epochs. In Figure A2, we contrast the distributions of similarity scores of matched pairs with the distributions of the similarity score for the 1,000 unmatched pairs that have the highest similarity score with the anchor. InfoNCE with Hopfield results in a moderate increase of the similarity scores of matched pairs, as well as in an increase of the similarity score of unmatched pairs. The latter is an undesired side effect of Hopfield networks as unmatched pairs get also more similar to one another. Compared to InfoNCE, InfoLOOB does not saturate. Therefore, it considerably increases the similarity between

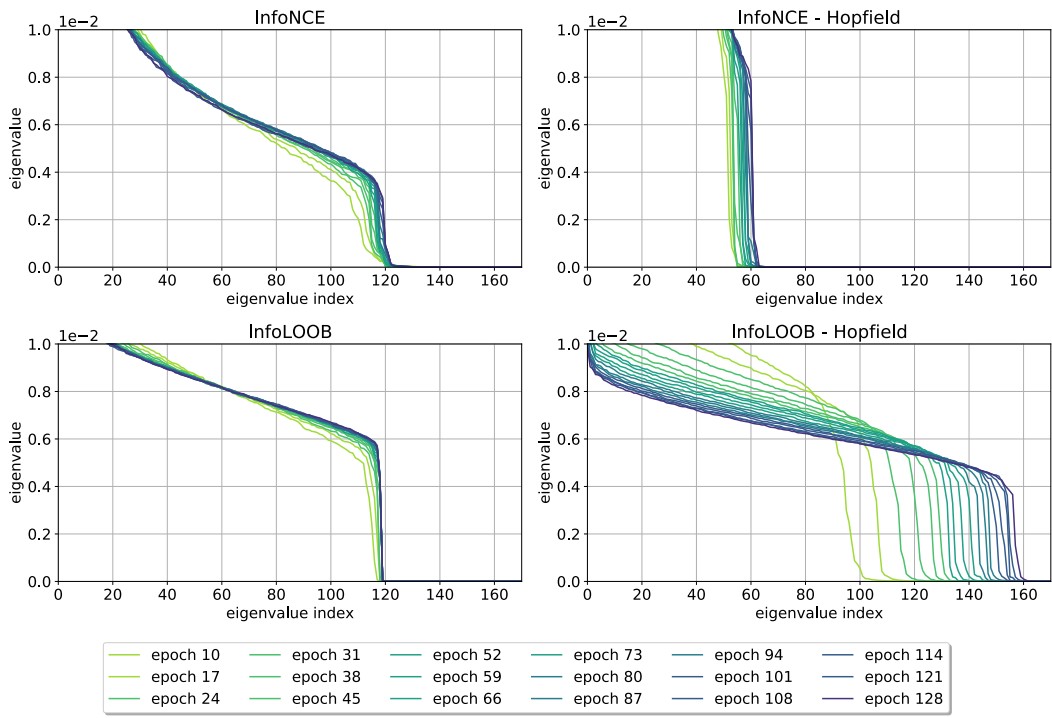

Figure A1: Sorted eigenvalues of InfoNCE and InfoLOOB with and without Hopfield retrievals during training.

matched pairs and also reduces the average similarity of the top-1000 unmatched pairs. InfoLOOB attributes a cosine similarity of one to many pairs, which is not plausible for multi-modal pairs. Clearly, this is an overfitting problem of InfoLOOB. Noteworthy, an observed increase in alignment between epoch 31 and epoch 128 does not benefit the downstream performance. The combination of Hopfield and InfoLOOB increases the similarity score of matched pairs and simultaneously reduces the average similarity of unmatched pairs compared to InfoNCE without Hopfield (the CLIP setting).

Figure A3 illustrates the overfitting problem of InfoLOOB. The distributions of unmatched pairs takes into account the ten unmatched pairs per anchor with the highest similarity score. In particular in the case of InfoLOOB without Hopfield, high similarity scores of the matched pairs correlate with high similarity scores of the top-10 unmatched pairs. In contrast, InfoLOOB with Hopfield does not suffer from this overfitting problem.

### A.2.8   Training time and memory consumption

To elaborate on the time and memory consumption of CLOOB, Table A9 compares the experiments done in Section 5.1. The memory consumption of CLOOB is the same as CLIP since only embeddings from the mini-batch are used both in the objective and in the Hopfield memories. The time consumption is approximately $5\%$ higher, which is because of the retrieval of the modern Hopfield networks. The added complexity of modern Hopfield networks is $\mathcal{O}(N)$ per sample, where $N$ denotes the batch size.

Table A9: Memory and time consumption of CLIP and CLOOB when trained for 31 epochs on CC.

| Model | Batch size (per GPU) | Memory (per GPU) | GPU hours | ImageNet zero-shot |
|---|---|---|---|---|
| CLIP | 128 | 13.7GB | 141 | 20.3 |
| CLOOB | 128 | 13.7GB | 148 | 24.0 |

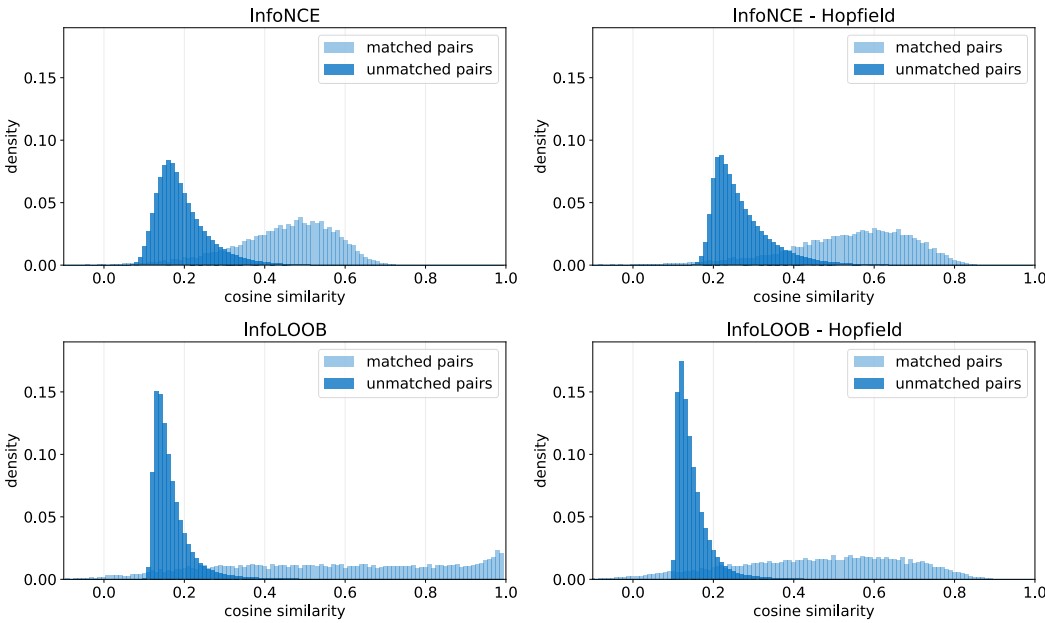

Figure A2: Distribution of the cosine similarity of matched pairs and the cosine similarity of the 1,000 unmatched pairs that have the highest similarity score with the anchor.

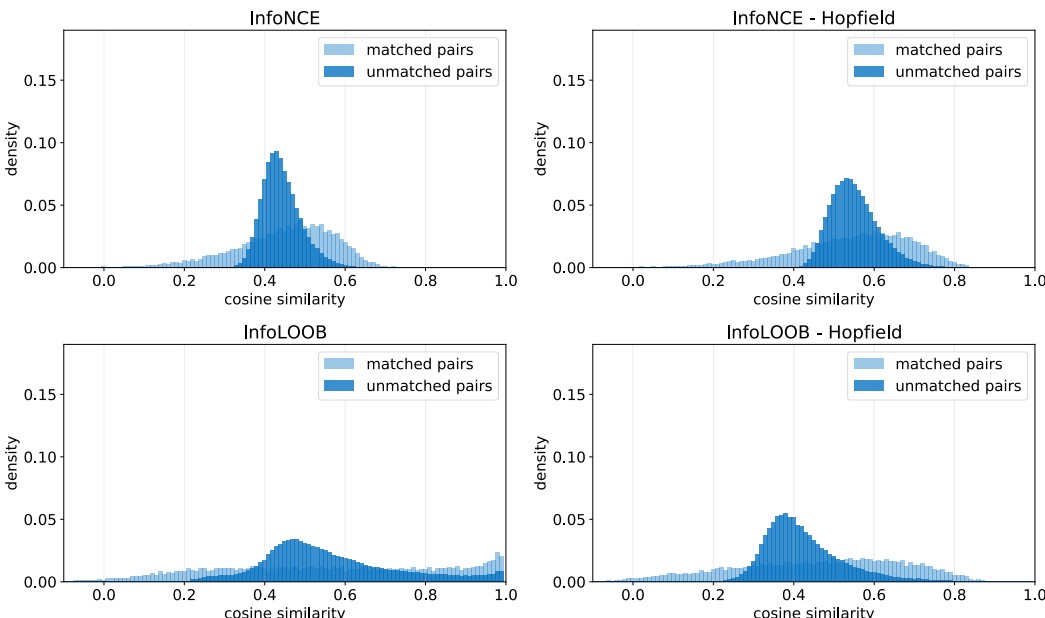

Figure A3: Distribution of the cosine similarity of matched pairs and the cosine similarity of the 10 unmatched pairs that have the highest similarity score with the anchor.

## A.3 Review of Modern Hopfield Networks

We briefly review continuous modern Hopfield networks that are used for deep learning architectures. They are continuous and differentiable, therefore they a work with gradient descent in deep architectures. They retrieve with one update only, therefore they can be activated like other deep learning layers. They have exponential storage capacity, therefore they can tackle large problems. Hopfield networks are energy-based, binary associative memories, which popularized artificial neural networks in the 1980s (Hopfield, 1982, 1984). Associative memory networks have been designed to store and

retrieve samples. Their storage capacity can be considerably increased by polynomial terms in the energy function (Chen et al., 1986; Psaltis & Cheol, 1986; Baldi & Venkatesh, 1987; Gardner, 1987; Abbott & Arian, 1987; Horn & Usher, 1988; Caputo & Niemann, 2002; Krotov & Hopfield, 2016). In contrast to these binary memory networks, we use continuous associative memory networks with very high storage capacity. These modern Hopfield networks for deep learning architectures have an energy function with continuous states and can retrieve samples with only one update (Ramsauer et al., 2021). Modern Hopfield Networks have been successfully applied to immune repertoire classification (Widrich et al., 2020) and chemical reaction prediction (Seidl et al., 2021).

We assume a set of patterns $\{\boldsymbol{u}_1, \ldots, \boldsymbol{u}_N\} \subset \mathbb{R}^d$ that are stacked as columns to the matrix $\boldsymbol{U} = (\boldsymbol{u}_1, \ldots, \boldsymbol{u}_N)$ and a state pattern (query) $\boldsymbol{\xi} \in \mathbb{R}^d$ that represents the current state. The largest norm of a stored pattern is $M = \max_i \|\boldsymbol{u}_i\|$. Continuous modern Hopfield networks with state $\boldsymbol{\xi}$ have the energy

$$\mathrm{E} = -\beta^{-1} \log \left( \sum_{i=1}^{N} \exp(\beta \boldsymbol{u}_i^T \boldsymbol{\xi}) \right) + \beta^{-1} \log N + \frac{1}{2} \boldsymbol{\xi}^T \boldsymbol{\xi} + \frac{1}{2} M^2 . \tag{A178}$$

For energy E and state $\boldsymbol{\xi}$, the update rule

$$\boldsymbol{\xi}^{\mathrm{new}} = f(\boldsymbol{\xi}; \boldsymbol{U}, \beta) = \boldsymbol{U} \, \boldsymbol{p} = \boldsymbol{U} \, \mathrm{softmax}(\beta \boldsymbol{U}^T \boldsymbol{\xi}) \tag{A179}$$

has been proven to converge globally to stationary points of the energy E, which are almost always local minima (Ramsauer et al., 2021). The update rule Eq. (A179) is also the formula of the well-known transformer attention mechanism (Ramsauer et al., 2021), therefore Hopfield retrieval and transformer attention coincide.

The *separation* $\Delta_i$ of a pattern $\boldsymbol{u}_i$ is defined as its minimal dot product difference to any of the other patterns: $\Delta_i = \min_{j, j \neq i} \left( \boldsymbol{u}_i^T \boldsymbol{u}_i - \boldsymbol{u}_i^T \boldsymbol{u}_j \right)$. A pattern is *well-separated* from the data if $\Delta_i \geq \frac{2}{\beta N} + \frac{1}{\beta} \log \left( 2(N-1) N \beta M^2 \right)$. If the patterns $\boldsymbol{u}_i$ are well separated, the iterate Eq. (A179) converges to a fixed point close to a stored pattern. If some patterns are similar to one another and, therefore, not well separated, the update rule Eq. (A179) converges to a fixed point close to the mean of the similar patterns. This fixed point is a *metastable state* of the energy function and averages over similar patterns.

The next theorem states that the update rule Eq. (A179) typically converges after one update if the patterns are well separated. Furthermore, it states that the retrieval error is exponentially small in the separation $\Delta_i$.

**Theorem A4** (Modern Hopfield Networks: Retrieval with One Update). *With query $\boldsymbol{\xi}$, after one update the distance of the new point $f(\boldsymbol{\xi})$ to the fixed point $\boldsymbol{u}_i^*$ is exponentially small in the separation $\Delta_i$. The precise bounds using the Jacobian $\mathrm{J} = \frac{\partial f(\boldsymbol{\xi})}{\partial \boldsymbol{\xi}}$ and its value $\mathrm{J}^m$ in the mean value theorem are:*

$$\|f(\boldsymbol{\xi}) - \boldsymbol{u}_i^*\| \leqslant \|\mathrm{J}^m\|_2 \|\boldsymbol{\xi} - \boldsymbol{u}_i^*\| , \tag{A180}$$

$$\|\mathrm{J}^m\|_2 \leqslant 2 \beta N M^2 (N-1) \exp(-\beta (\Delta_i - 2 \max\{\|\boldsymbol{\xi} - \boldsymbol{u}_i\|, \|\boldsymbol{u}_i^* - \boldsymbol{u}_i\|\} M)) . \tag{A181}$$

*For given $\epsilon$ and sufficient large $\Delta_i$, we have $\|f(\boldsymbol{\xi}) - \boldsymbol{u}_i^*\| < \epsilon$, that is, retrieval with one update. The retrieval error $\|f(\boldsymbol{\xi}) - \boldsymbol{u}_i\|$ of pattern $\boldsymbol{u}_i$ is bounded by*

$$\|f(\boldsymbol{\xi}) - \boldsymbol{u}_i\| \leqslant 2 (N-1) \exp(-\beta (\Delta_i - 2 \max\{\|\boldsymbol{\xi} - \boldsymbol{u}_i\|, \|\boldsymbol{u}_i^* - \boldsymbol{u}_i\|\} M)) M . \tag{A182}$$

For a proof see (Ramsauer et al., 2021).

The main requirement of modern Hopfield networks to be suited for contrastive learning is that they can store and retrieve enough embeddings if the batch size is large. We want to store a potentially large set of embeddings. We first define what we mean by storing and retrieving patterns from a modern Hopfield network.

**Definition A1** (Pattern Stored and Retrieved). *We assume that around every pattern $\boldsymbol{u}_i$ a sphere $\mathrm{S}_i$ is given. We say $\boldsymbol{u}_i$ is stored if there is a single fixed point $\boldsymbol{u}_i^* \in \mathrm{S}_i$ to which all points $\boldsymbol{\xi} \in \mathrm{S}_i$ converge, and $\mathrm{S}_i \cap \mathrm{S}_j = \emptyset$ for $i \neq j$. We say $\boldsymbol{u}_i$ is retrieved for a given $\epsilon$ if iteration (update rule) Eq. (A179) gives a point $\tilde{\boldsymbol{x}}_i$ that is at least $\epsilon$-close to the single fixed point $\boldsymbol{u}_i^* \in \mathrm{S}_i$. The retrieval error is $\|\tilde{\boldsymbol{x}}_i - \boldsymbol{u}_i\|$.*

As with classical Hopfield networks, we consider patterns on the sphere, i.e. patterns with a fixed norm. For randomly chosen patterns, the number of patterns that can be stored is exponential in the dimension $d$ of the space of the patterns ($\boldsymbol{u}_i \in \mathbb{R}^d$).

**Theorem A5** (Modern Hopfield Networks: Exponential Storage Capacity). *We assume a failure probability $0 < p \leqslant 1$ and randomly chosen patterns on the sphere with radius $M := K\sqrt{d-1}$. We define $a := \frac{2}{d-1}(1 + \ln(2\beta K^2 p(d-1)))$, $b := \frac{2K^2\beta}{5}$, and $c := \frac{b}{W_0(\exp(a+\ln(b)))}$, where $W_0$ is the upper branch of the Lambert W function (Olver et al., 2010, (4.13)) , and ensure $c \geq \left(\frac{2}{\sqrt{p}}\right)^{\frac{4}{d-1}}$. Then with probability $1 - p$, the number of random patterns that can be stored is*

$$N \geq \sqrt{p}\, c^{\frac{d-1}{4}} . \tag{A183}$$

*Therefore it is proven for $c \geq 3.1546$ with $\beta = 1$, $K = 3$, $d = 20$ and $p = 0.001$ ($a + \ln(b) > 1.27$) and proven for $c \geq 1.3718$ with $\beta = 1$, $K = 1$, $d = 75$, and $p = 0.001$ ($a + \ln(b) < -0.94$).*

For a proof see (Ramsauer et al., 2021).

This theorem justifies to use continuous modern Hopfield networks for using retrieved embeddings instead of the original embeddings for large batch sizes. Even for hundreds of thousands of embeddings, the continuous modern Hopfield network is able to retrieve the embeddings if the dimension of the embeddings is large enough.

## A.4   Further Related Work

With the advent of large corpora of unlabeled data in vision and language, self-supervised learning via contrastive learning has become highly successful. Some contrastive learning objectives, such as those of BYOL (Grill et al., 2020) and SimSiam (Chen & He, 2021), do not require negative samples. However, the most popular objective for contrastive learning is InfoNCE (van den Oord et al., 2018), in which for an anchor sample, a positive sample is contrasted with negative samples.

The idea to use objectives with negative samples is well known in deep learning (Gutmann & Hyväri-nen, 2010; Chen et al., 2017; Mikolov et al., 2013). For contrastive learning, the most successful objective is InfoNCE, which has been introduced as Contrastive Predictive Coding (CPC) (van den Oord et al., 2018). InfoNCE has been applied to transfer learning (Hénaff et al., 2019), to natural language response suggestion (Henderson et al., 2017), to learning sentence representations from unlabelled data (Logeswaran & Lee, 2018), and to unsupervised feature learning by maximizing distinctions between instances (Wu et al., 2018). InfoNCE has been used for learning visual representations in Pretext-Invariant Representation Learning (PIRL) (Misra & vanDerMaaten, 2020), in Momentum Contrast (MoCo) (He et al., 2020), and in SimCLR (Chen et al., 2020). SimCLR became well known as is was highly effective for transfer learning. Zero-shot transfer learning (Lampert et al., 2009) is one of the most ambitious goals in vision, since it would improve various real-world downstream applications. Current models in natural language processing and vision perform very well on standard benchmarks, but they fail at new data, new applications, deployments in the wild, and stress tests (D'Amour et al., 2020; Recht et al., 2019; Taori et al., 2020; Lapuschkin et al., 2019; Geirhos et al., 2020). A model with high zero-shot transfer learning performance will not fail on such data, therefore will be trusted by practitioners.

Multiple works have proposed improvements to InfoNCE. Joint Contrastive Learning (JCL) studies the effect of sampling multiple positives for each anchor. (Cai et al., 2020). Sampling negatives around each positive leads to higher bias but lower variance than InfoNCE (Wu et al., 2021). InfoNCE has been generalized to C-InfoNCE and WeaC-InfoNCE, which are conditional contrastive learning approaches to remove undesirable information in self-supervised representations (Tsai et al., 2021). ProtoNCE is a generalized version of the InfoNCE, which pushes representations to be closer to their assigned prototypes (Li et al., 2021). ProtoNCE combines contrastive learning with clustering. SimCSE employs InfoNCE for contrastive learning to learn sentence embeddings (Gao et al., 2021). InfoNCE has been extended to video representation learning (Han et al., 2020).

CLOOB uses InfoLOOB, which is an upper bound on the mutual information. An alternative upper bound on the mutual information would be Contrastive Log-ratio Upper Bound (CLUB), which was used for minimizing the mutual information (Cheng et al., 2020). So far CLUB was only used for minimizing the mutual information, except for the analysis in (Wang & Liu, 2021). In our

experiments, maximizing CLUB failed as confirmed in (Wang & Liu, 2021). The reason is that the embedding distribution is not uniform as required for successful contrastive learning (Wang & Isola, 2020; Wang & Liu, 2021).

Many follow up works have been based on the CLIP model. The CLIP model is used in Vision-and-Language tasks (Shen et al., 2021). The CLIP model guided generative models via an additional training objective (Bau et al., 2021; Galatolo et al., 2021; Frans et al., 2021) and improved clustering of latent representations (Pakhomov et al., 2021). It is used in studies of out of distribution performance (Devillers et al., 2021; Milbich et al., 2021; Miller et al., 2021), of fine-tuning robustness (Wortsman et al., 2021), of zero-shot prompts (Zhou et al., 2021) and of adversarial attacks to uncurated datasets (Carlini & Terzis, 2021). It stirred discussions about more holistic evaluation schemes in computer vision (Agarwal et al., 2021). Multiple methods utilize the CLIP model in a straightforward way to perform text-to-video retrieval (Fang et al., 2021; Luo et al., 2021; Narasimhan et al., 2021).

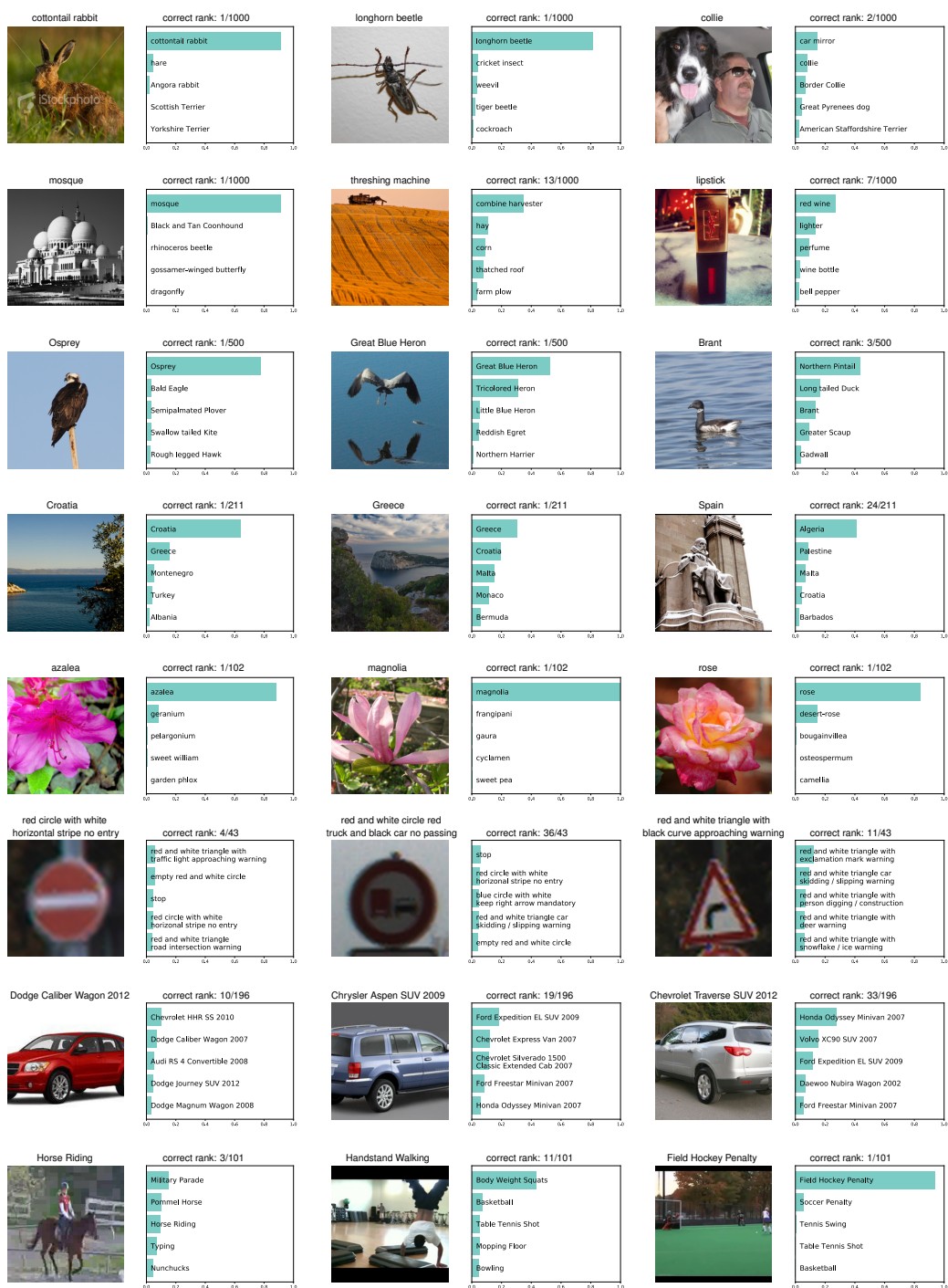

Figure A4: Visualization of zero-shot classification of three examples from each dataset. The following datasets are used (top to bottom): ImageNet, ImageNet V2, Birdsnap, Country211, Flowers102, GTSRB, Stanford Cars and UCF101. The ground truth label is displayed above the picture. The bar plots show the softmax values of the top 5 classes.