# OpenReview forum: "CLOOB: Modern Hopfield Networks with InfoLOOB Outperform CLIP"
_NeurIPS.cc/2022/Conference — NeurIPS 2022 Accept_

### Official Review · Reviewer_pFNt · 2022-07-10

**Rating:** 4
**Confidence:** 4
**Soundness:** 3 good
**Presentation:** 3 good
**Contribution:** 2 fair

**Summary:**

The paper proposes a new contrastive learning methods which leverages Modern Hopfield Networks combined with  InfoLOOB objective. The experiments show that the proposed method has promising results when compared with the baseline method on several benchmark datasets.

**Questions:**

what's the time complexity of the proposed model, and how does it compared with the baseline method?

**Limitations:**

see above

**Strengths And Weaknesses:**

strength:
1. The paper is well written with comprehensive details.
2. The proposed method outperforms the baseline methods by a large margin

weakness:
1. The novelty of the paper is below the bar.  It combines the InfoLOOB and modern Hopfield networks, which are all existing works even though the application scenario is new

---

> ### Author Response · Authors · 2022-08-02
> **Response to Reviewer pFNt**
>
> Thank you for your comments which helped to improve the paper. We addressed all your comments and followed all your suggestions.
>
> **Novelty**
>
> We disagree with the reviewer concerning the novelty. So far, Hopfield networks have been used as associative memories to store patterns. In contrast, we use modern Hopfield networks to amplify the covariance structure in the data. We utilize modern Hopfield networks in a completely novel way beyond storing patterns by reinforcing the covariance structure of the data. In classical Hopfield networks the memory matrix constructed by Hebb learning is a covariance matrix, where the retrieval procedure is reinforcing the query covariance structure. Thus, there is a close connection between associative memories and the enhancement of the data covariance structure.
>
> The theoretical analysis at the beginning of Section 3 shows that modern Hopfield networks amplify the covariance structure in the retrieved embedding compared to the actual embedding. According to this analysis, the actual embedding is multiplied by the covariance matrix of stored embeddings that are similar to the actual embedding. Hence, co-occurrences in the actual embedding that also appear in stored embeddings are amplified while other co-occurrences diminish.
>
> In a recent work [1] it was shown that contrastive loss functions like InfoNCE are governed by the empirical covariance matrix as demonstrated for the methods TiCo, MoCo, or Barlow Twins. This fact explains why amplifying the covariance structure by modern Hopfield networks boosts the performance of contrastive loss functions.
>
> We explain the novelty of our work more clearly in the new version of the paper.
>
> **Training time**
>
> When training CLOOB and CLIP on CC with ResNet-50 vision encoders, CLOOB requires approximately 5% more training time than CLIP, i.e. CLOOB is a factor of 1.05 slower than CLIP. Both methods have a complexity of O(N) when N is the batch size.
>
> In the new manuscript we report the training time and memory consumption in Section A.2.8.
>
>
> [1] J. Zhu et al. 'TiCo: Transformation Invariance and Covariance Contrast for Self-Supervised Visual Representation Learning', arXiv, 2022

---

### Official Review · Reviewer_63eh · 2022-07-11

**Rating:** 5
**Confidence:** 5
**Soundness:** 4 excellent
**Presentation:** 4 excellent
**Contribution:** 3 good

**Summary:**

This paper propose CLOOB, which utilizes modern Hopfield networks to enchancing the learning of covariance structure in the original multi-modal data. In addition, in order to alleviate the impact of Hopfield networks on InfoNCE, the author proposes to use the InfoLOOB objective. Experiments on several classification datasets demenstrated the effectiveness of CLOOB.

**Questions:**

1. How much additional computational overhead will the Hopfield Network bring?

2. Besides CLIP, can CLOOB be used in other pre-training methods like DECLIP, FILIP?

3. CLOOB is only transferred to the downstream classification task, how is the result on the image-text retrieval task?

4. A little worried about the results of CLOOB, the results of RN50 pre-trained on CC3M did not exceed the results of the DECLIP report.

**Limitations:**

The authors have adequately addressed the limitations and potential negative societal impact of their work.

**Strengths And Weaknesses:**

1. The motivation is clear, i.e. enchancing the learning of covariance structure in the original multi-modal data.

2. It is proved on the several downstream classification dataset that CLOOB improves the pre-training on both CC3M and YFCC subsets.

---

> ### Author Response · Authors · 2022-08-02
> **Response to Reviewer 63eh**
>
> Thank you for your review and helpful comments which were very useful to improve the paper. We address your points individually.
>
> **Training time**
>
> When training CLOOB and CLIP on CC with ResNet-50 vision encoders, CLOOB requires approximately 5% more training time than CLIP, i.e. CLOOB is a factor of 1.05 slower than CLIP. Both methods have a complexity of O(N) when N is the batch size.
>
> In the new manuscript we report the training time and memory consumption in Section A.2.8.
>
> **CLOOB as replacement for CLIP**
>
> Yes. CLOOB can readily substitute CLIP in every method that improves CLIP.
>
> We now mention this in the new manuscript.
>
> **Image-text retrieval**
>
> The reviewer raises an important point. We now report results on the image-text retrieval tasks. CLOOB clearly outperforms CLIP.
>
> We now report results on image-text retrieval tasks in the new manuscript in Section A.2.6 and Table A8.
>
> **DeCLIP performance**
>
> The results of CLOOB and DeCLIP cannot be compared. DeCLIP uses self-supervision within each modality, multi-view supervision across modalities, and nearest-neighbor supervision from other similar pairs. DeCLIP considerably augments the data and uses self-supervision. Of course, CLOOB can use the same techniques. However, our goal was to show that CLOOB performs better than CLIP without enhancements or data augmentation. We were interested to compare the basic methods CLOOB and CLIP and not their extensions.

---

### Official Review · Reviewer_3XnZ · 2022-07-11

**Rating:** 5
**Confidence:** 4
**Soundness:** 3 good
**Presentation:** 2 fair
**Contribution:** 1 poor

**Summary:**

This work proposes the CLOOB objective for contrastive image-text joint training and performs studies comparing against CLIP objective. Results show improvement in performance on multiple downstream tasks on zero-shot performance. However, there is a concern about novelty of this work, as both modern Hopfield networks and InfoLOOB are proposed previously. In addition, zero shot performance improvements do not consistently reflect on to linear probe performance, which is concerning and hence some experimental setups need more detailed analysis.

**Questions:**

- How does the training time gets affected with CLOOB framework compared to CLIP framework?
- Did the authors also try with other backbones like VIT? Larger models?
- For trying out larger models, datasets larger than YFCC can be used. Example open sourced datasets are https://huggingface.co/datasets/facebook/pmd.

**Limitations:**

Yes authors have adequately addressed the limitations and potential negative societal impact of their work.

**Strengths And Weaknesses:**

### Strengths

- The method CLOOB improves performance compared to reimplemented CLIP baselines on Resnet architectures.
- Authors show some interesting analysis of the benefits of both Hopfield networks and InfoLOOB

### Weaknesses

- Given both InfoLOOB and modern Hopfield networks are previous works, the proposed method is not novel and just incremental.
- The improvements from using InfoLOOB as the contrastive objective are already discussed in previous work of Decoupled Contrastive Learning by Yeh et.al, which also shows benefits of applying InfoLOOB against SimCLR, MoCo frameworks.
- Given the sole method that the paper compares against is CLIP, the work only compares against 8 tasks as opposed to 30+ vision tasks that CLIP evaluates on. All the datasets are open sourced and since the work compares against a reimplementation of CLIP, showing results on more variety of evaluation datasets would be helpful to understand the improvements from CLOOB on a larger variety of tasks.
- One big difference I observed is that the batch sizes used in this work are very small (512, 1024) compared to CLIP (16K, 32K). Since the benefits of CLIP loss improves with significant large batch sizes due to large number of negative samples in the contrastive setup, it is not clear to me how the CLOOB objective compares with CLIP on larger batch sizes.
- Linear probing results in appendix are not very encouraging as CLOOB has mixed results compared to CLIP in Table A5. This is also concerning as the main paper Table 2 shows comparison with CLIP(OpenAI) that shows improvements on most tasks, but with similar experimentation setup, the results are mixed.
- Overall the paper's story has lot of repetition of the similar sentences about why Hopfield networks and InfoLOOB are better. I would encourage the authors to restructure the paper and bring more analysis from the Appendix section to the main paper instead of repetitive sentences explaining the benefits.

---

> ### Author Response · Authors · 2022-08-02
> **Response to Reviewer 3XnZ (Part 2)**
>
> **Large batch sizes**
>
> We agree with the reviewer that larger mini-batches reduce the bias of the InfoNCE loss. Due to hardware restrictions, we were not able to use these extremely large batch sizes. We believe that CLOOB profits as well from larger batch sizes, since the Hopfield networks with more samples contain a richer covariance structure.
>
> **Linear probing**
>
> The reviewer is right. CLOOB and the reimplemented CLIP show comparable performances for linear probing, which are considerably larger than the corresponding zero-shot performances. This indicates that linear probing can extract information from embeddings as long as the information is available. Linear probing achieves comparable results as logistic regression on public checkpoints with similar ImageNet performance [2]. Thus, linear probing is less indicative of the quality of the embeddings but indicates whether the embeddings have lost information from the raw input.
>
> **Repetition in the paper**
>
> We follow the suggestions of the reviewer. We removed the repetitions. We moved the analysis on cosine similarity of matched and unmatched embeddings of CLIP, CLIP+InfoLOOB, CLIP+Hopfield, and CLOOB (CLIP+InfoLOOB+Hopfield) from the appendix Section A.2.7 to the main paper. This analysis shows how InfoLOOB and modern Hopfield networks change the cosine similarity of matched and unmatched embeddings.
>
> We changed the manuscript accordingly.
>
> **Training time**
>
> When training CLOOB and CLIP on CC with ResNet-50 vision encoders, CLOOB requires approximately 5% more training time than CLIP, i.e. CLOOB is a factor of 1.05 slower than CLIP. Both methods have a complexity of O(N) when N is the batch size.
>
> In the new manuscript, we report the training time and memory consumption in Section A.2.8.
>
> **ViT and larger models**
>
> Yes, we made some initial experiments but saw that larger datasets than YFCC are needed to train ViT architectures.
> However, we were limited by our hardware constraints.
>
> **Larger datasets**
>
> Thank you for the hint. We will run experiments with these data sets but they will not be completed within the rebuttal phase.
>
> CLOOB excels on data sets with rich covariance structure. For example, CC has a strong covariance structure in particular for the text, therefore CLOOB clearly outperforms CLIP on CC.
>
>
>
> [1] J. Zhu et al. 'TiCo: Transformation Invariance and Covariance Contrast for Self-Supervised Visual Representation Learning', arXiv, 2022
>
> [2] S. Kornblith et al. 'Do Better ImageNet Models Transfer Better?', CVPR, 2019

---

> ### Author Response · Authors · 2022-08-02
> **Response to Reviewer 3XnZ (Part 1)**
>
> We thank you for your thorough review and your suggestion to expand our work which thereby improved the paper. We addressed all your comments and followed all your suggestions.
>
> **Novelty**
>
> We disagree with the reviewer concerning the novelty. So far, Hopfield networks have been used as associative memories to store patterns. In contrast, we use modern Hopfield networks to amplify the covariance structure in the data. We utilize modern Hopfield networks in a completely novel way beyond storing patterns by reinforcing the covariance structure of the data. In classical Hopfield networks the memory matrix constructed by Hebb learning is a covariance matrix, where the retrieval procedure is reinforcing the query covariance structure. Thus, there is a close connection between associative memories and the enhancement of the data covariance structure.
>
> The theoretical analysis at the beginning of Section 3 shows that modern Hopfield networks amplify the covariance structure in the retrieved embedding compared to the actual embedding. According to this analysis, the actual embedding is multiplied by the covariance matrix of stored embeddings that are similar to the actual embedding. Hence, co-occurrences in the actual embedding that also appear in stored embeddings are amplified while other co-occurrences diminish.
>
> In a recent work [1] it was shown that contrastive loss functions like InfoNCE are governed by the empirical covariance matrix as demonstrated for the methods TiCo, MoCo, or Barlow Twins. This fact explains why amplifying the covariance structure by modern Hopfield networks boosts the performance of contrastive loss functions.
>
> We explain the novelty of our work more clearly in the new version of the paper.
>
> **InfoLOOB already discussed in Yeh et al.**
>
> The reviewer is right. The advantage of InfoLOOB was simultaneously and independently discovered in Yeh et al. and our work. We cite Yeh et al.. In contrast to Yeh et al. we give a theoretical justification for using InfoLOOB as an objective in Appendix A.1. In particular in A.1.3, we show why InfoLOOB is a proper objective even if it is not a bound on the mutual information. We think this analysis is essential for justifying InfoLOOB as an objective.
>
> **More evaluation datasets**
>
> We agree with the reviewer. We have chosen these 8 tasks to compare to the original OpenAI CLIP, for which only results for these tasks on CC and YFCC have been reported.
> We now added zero-shot results on Caltech101, Describable Textures, Eurosat, Facial Emotion Recognition 2013, FGVC-Aircraft, Food101, Oxford Pets, RESISC45, STL10, SUN397, CIFAR10 and CIFAR100. This results in an evaluation on 12 additional zero-shot datasets, which shows the improvements of CLOOB on a large variety of tasks.
> In total we evaluate on datasets in alignment with other papers like DeCLIP and FILIP.
>
> We changed the manuscript accordingly. The results for models trained on CC are presented in Table A5. The results for models trained on YFCC are presented in Table A6.

---

### Author Response · Authors · 2022-08-02
**General response**

We thank all reviewers for their positive feedback and for their constructive comments and suggestions.
The reviewers appreciated our experiments which show that CLOOB improves over CLIP.

The feedback of reviewers helped a lot to improve our paper. We answered all questions and comments of the reviewers and changed the paper accordingly. Furthermore we followed all their suggestions in the new version of the paper.

We provide answers, clarifications, and improvements of the paper in the individual responses to the respective reviewers. Further, we uploaded a revision of our paper in OpenReview.

**Concerning the novelty:**

So far, Hopfield networks have been used as associative memories to store patterns. In contrast, we use modern Hopfield networks to amplify the covariance structure in the data. We utilize modern Hopfield networks in a completely novel way beyond storing patterns by reinforcing the covariance structure of the data. In a recent work [1] it was shown that contrastive loss functions like InfoNCE are governed by the empirical covariance matrix as demonstrated for the methods TiCo, MoCo, or Barlow Twins.

**Training time**

When training CLOOB and CLIP on CC with ResNet-50 vision encoders, CLOOB requires approximately 5% more training time than CLIP.


[1] J. Zhu et al. 'TiCo: Transformation Invariance and Covariance Contrast for Self-Supervised Visual Representation Learning', arXiv, 2022

---

### Meta-Review · Area_Chair_7QNH · 2022-08-31

**Recommendation:** Accept
**Confidence:** Less certain

**Metareview:**

The submission proposes a new objective for contrastive training with images+text. The comparison is made with CLIP. The method, CLOOB, improves upon a reimplementation of CLIP. The main concern here is that InfoLOOB and Hopfield nets are known already and whether this submission is thus incremental. Reviewer 3XnZ points out that InfoLOOB as a contrastive objective was previously discussed, for example. Like the reviewer, I am concerned that the comparisons against CLIP are not quite done properly: fewer tasks (this is improved in the rebuttal) and much smaller batches. The authors *claim* that larger batches sizes would benefit CLOOB too, but it's not something really shown in the work.

I believe the rebuttal addresses many of the reviewers' concerns. I can understand the concerns regarding novelty, but I believe the authors when they say that they use Hopfield nets in a way that wasn't used before. Given the promising results, careful ablations/studies, and relatively small (5%) time penalty, I think this could be of general interest, so I vote for this to be accepted.


**Award:**

No

---

### Decision · Program_Chairs · 2022-09-14

Accept